# HUGAGENT: BENCHMARKING LLMS FOR SIMULATION OF INDIVIDUALIZED HUMAN REASONING

## ABSTRACT

Simulating human reasoning in open-ended tasks has long been a central aspiration in AI and cognitive science. While large language models now approximate human responses at scale, they remain tuned to population-level consensus, often erasing the individuality of reasoning styles and belief trajectories. To advance the vision of more human-like reasoning in machines, we introduce **HugAgent** (**HU**man-**G**rounded **AGENT** Benchmark), which rethinks human reasoning simulation along three dimensions: (*i*) from **averaged** to **individualized** reasoning, (*ii*) from **behavioral mimicry** to **cognitive alignment**, and (*iii*) from **vignette-based** to **open-ended**[1] data. The benchmark evaluates whether a model can predict a *specific person's* **behavioral responses** and the underlying **reasoning dynamics** in out-of-distribution scenarios, given partial evidence of their prior views. HugAgent adopts a dual-track design: a *human track* that automates and scales the think-aloud method to collect ecologically valid human reasoning data and a *synthetic track* for further scalability and systematic stress testing. This architecture enables low-cost, extensible expansion to new tasks and populations. Experiments with state-of-the-art LLMs reveal persistent adaptation gaps, positioning HugAgent as the first extensible benchmark for aligning machine reasoning with the individuality of human thought. The benchmark, along with its complete data collection pipeline and companion chatbot, is open-sourced as *HugAgent* and *TraceYourThinking*.

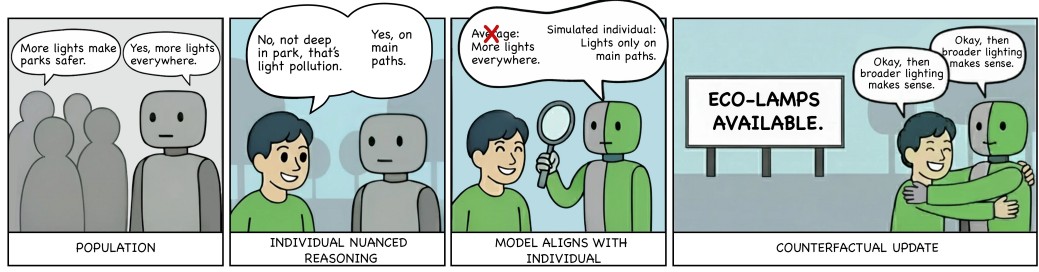

Figure 1: Illustration of **HugAgent** operationalizing "average-to-individual" reasoning adaptation: a gray robot repeats population consensus, then observes an individual's nuanced reasoning, gradually aligns with that individual (turning colorful), and finally adapts under counterfactual updates (e.g., with eco-lamps). This illustrates the shift from consensus mimicry to individualized reasoning.

## 1 INTRODUCTION

**Background.** Large language models (LLMs) are increasingly used as social simulators—to role-play individuals, build digital twins, and generate synthetic ('silicon') samples for testing social and policy ideas(Park et al., 2024; Argyle et al., 2023; Xie et al., 2024a; Jiang et al., 2022). These systems promise scalability and accessibility: instead of recruiting thousands of people, researchers and practitioners can use LLMs to approximate human perspectives at scale. Yet because LLMs are pretrained on population-level corpora, they tend to collapse into an "average voice," capturing consensus patterns while erasing the individuality of personal histories, beliefs, and reasoning styles(Wang et al., 2025; Santurkar et al., 2023; Durmus et al., 2023).

> **This paper asks a core question:** *can LLMs move from simulating the **average** to simulating the **individual**?*

In other words, can they predict how a specific person would think, believe, and reason in new scenarios, given evidence of their past views? We formalize this broad challenge as **average-to-**

---

[1]Here, "open-ended" refers to the *unconstrained nature of the human reasoning protocols* (think-aloud traces) from which our data is derived (Ericsson & Simon, 1993), as opposed to artificial, researcher-authored vignettes (Martinez, 1999). While our evaluation uses structured formats (e.g., MCQ) for rigor and scalability, the content anchors in rich, generative human cognition (Guilford, 1967).

**individual reasoning adaptation**, a measurable task that targets *intra-individual fidelity* in human simulation(formally defined in Section 2).

**Motivation.**    Current benchmarks fail to capture this ability, across three key dimensions. ❶ **Intra-agent vs. inter-agent fidelity.** Existing pluralistic alignment benchmarks probe group dynamics and social influence(Sorensen et al., 2024), but neglect whether models can faithfully reproduce reasoning *within* a single agent, which is crucial for identity-consistent modeling. ❷ **Reasoning traces vs. behavioral outcomes.** Large-scale "digital twin" datasets such as Agent Bank (Park et al., 2024) and Twin-2K-500 (Toubia et al., 2025) primarily assess static behavioral outcomes, but not the evolving reasoning trajectories of a single individual, which are essential for credible social simulation (Li et al., 2025). ❸ **Open-ended vs. vignettes.** Commonsense and social reasoning benchmarks (e.g., SocialIQA, ATOMIC) often reduce diverse answers to a single ground truth (Sap et al., 2019a;b). Opinion-oriented datasets likewise emphasize aggregate patterns over individual variation (Argyle et al., 2023; Santurkar et al., 2023). Theory-of-Mind style tests typically rely on short vignettes with designer labels (Wimmer & Perner, 1983; Chen et al., 2024), which limits ecological validity and overlooks first-person reasoning traces that could serve as a richer gold standard (Ying et al., 2025).

**Methodology.**    Motivated by these gaps, we introduce **HugAgent**, a benchmark that targets *intra-agent fidelity* by operationalizing average-to-individual reasoning adaptation as a measurable task. For Dimension ❶, HugAgent shifts the granularity from *inter-agent* to *intra-agent* fidelity: given a person's profile and reasoning history, a model must predict both their current belief state and how it would evolve when presented with new counterfactual evidence. In Dimension ❷, HugAgent advances beyond static outcomes toward reasoning trajectories. It collects *first-person, out-loud self-reports* as gold-standard reasoning traces. These traces offer a deeper target for prediction than the choice outcomes or survey responses typically captured in lab experiments. To address Dimension ❸, instead of relying on vignette-style benchmarks, HugAgent builds evaluation around open-ended contexts. We curated real-world topics, beginning with *socially and politically controversial issues that introduce inherent conflicts*. Through sustained follow-up questions, the benchmark probes participants' *deliberate, System 2 style* reasoning(Kahneman, 2011; Evans & Stanovich, 2013), transforming the dataset from toy settings into complex, open-ended domains. For a broader discussion of prior work on personalization, social reasoning, and user modeling in LLMs, we refer readers to Appendix A.

To build such a benchmark at *scale* while retaining *ecological validity*(i.e., the extent to which findings reflect reasoning as it occurs in real-world contexts), **HugAgent** combines two complementary tracks. A *synthetic track* provides large, controlled datasets where belief shifts and reasoning paths can be systematically manipulated(Yukhymenko et al., 2024; Xie et al., 2024b). A *human-grounded track* applies the same protocol to *real human participants*, yielding data anchored in individual variation(Srivastava et al., 2023).

**Contributions.**    Our contributions are fourfold:

- **What does it mean to adapt from the average to the individual?** We formalize *average-to-individual reasoning adaptation* as a measurable task: predicting an individual's beliefs and reasoning trajectory from partial self-reported data, rather than collapsing variation into an "average" label.

- **How well do today's models perform?** We introduce **HugAgent**, a dual-track benchmark (synthetic + human) that evaluates both *Belief State Inference* and *Belief Dynamics Update*. Initial experiments with state-of-the-art LLMs provide baseline results and reveal adaptation gaps. `https://anonymous.4open.science/r/HugAgent`

- **Where do they fail, and what can improve?** Building on these evaluations, we conduct detailed error analyses across synthetic agents, human participants, and state-of-the-art LLMs. This uncovers recurring failure modes and points to concrete avenues for alignment.

- **How can such evaluation scale and persist?** We release the entire pipeline as *open source*, including a semi-structured interview chatbot that elicits fine-grained, "out-loud" reasoning data on arbitrary topics. This provides the community with previously lacking resources for capturing not only static answers but also the reasoning processes behind them, ensuring HugAgent is reproducible, extensible, and sustainable. `https://anonymous.4open.science/r/trace-your-thinking`

By making "average-to-individual" reasoning adaptation measurable, **HugAgent** takes a first step toward a reproducible framework for studying human simulation at the level of individual reasoning.

## 2 PROBLEM SETUP AND THEORETICAL FRAMING

We operationalize individual reasoning through *belief states* (snapshots) and *belief dynamics* (updates under interventions). This framing allows measurable comparison while respecting the diversity of human reasoning paths.

### 2.1 FORMALIZATION

We formalize *average-to-individual reasoning adaptation* by modeling an individual $i$'s belief state as a distribution over $d$ factors

$$b_i \equiv P_{\phi_i}(\mathbf{s} \mid \mathcal{C}_i), \quad \mathbf{s} \in \mathbb{R}^d,$$

with context $\mathcal{C}_i$ (e.g., demographics, transcripts). Under an intervention $\mathcal{I}_t$, beliefs evolve via

$$b_i^{t+1} = \mathcal{U}(b_i^t, \mathcal{I}_t), \quad \Delta b_i^t = \mathbb{E}_{b_i^{t+1}}[\mathbf{s}] - \mathbb{E}_{b_i^t}[\mathbf{s}].$$

Here, $\mathcal{U}$ formalizes the *reasoning dynamics*, the mechanism by which an agent updates its internal state when receiving new information. We use $\mathcal{U}$ as a broad abstraction that covers both idealized normative updates (Section 2.2) and the heuristic transitions seen in human and LLM reasoning.

**Tasks.** (i) *Belief State Inference:* infer stance/factor polarity from $\mathcal{C}_i$. (ii) *Belief Dynamics Update:* predict stance shifts $\widehat{\Delta}\mathbf{s}_i$ given $(\mathcal{C}_i, \mathcal{I})$. Metrics include accuracy, mean absolute error (MAE), and rank correlation.

### 2.2 THEORETICAL ANCHORS: PROBABILISTIC AND CAUSAL PERSPECTIVES

We use normative models only as conceptual anchors rather than methodological assumptions. (1) **Bayesian / PLoT.** Idealized revision follows Bayesian conditioning $b_i'(\mathbf{s}) \propto b_i(\mathbf{s}) \, p(\mathcal{I} \mid \mathbf{s})$, interpreting language as probabilistic evidence over latent stances. We also draw on the **Probabilistic Language of Thought (PLoT)** framework(Goodman et al., 2015), which extends Bayesian inference to compositional linguistic structures. (2) **Structural Causal Models (SCM).** Interventions can be framed as $do(\mathcal{I})$ on a causal graph of values/reasons, yielding counterfactual belief shifts $\mathbb{E}[\mathbf{s} \mid do(\mathcal{I})]$. We additionally represent a person's value–reason structure as a signed directed graph $G_i$; we hypothesize that similarity between such graphs (e.g., via graph edit distance or learned embeddings) may predict cross-domain transfer, motivating Hypothesis H2 at a conceptual level; empirical tests of graph similarity remain future work. Human reasoning deviates from these ideals; the anchors provide principled baselines for analysis.

### 2.3 GUIDING HYPOTHESES

Grounding HugAgent in theory enables us to frame four *guiding hypotheses* that serve as lenses for interpreting empirical results, rather than assumptions to be fully verified:

- **H1 (Intra-individual consistency)**: With sufficient context (e.g., demographic features or prior transcripts), LLMs can stably capture an individual's belief state.
- **H2 (Cross-domain transfer bound)**: Reasoning patterns transfer partially across domains, and accuracy under domain transfer is significantly lower than in-domain performance.
- **H3 (Population prior reliance)**: Without individual context, LLMs default to global population priors rather than individual-specific cues.
- **H4 (Context information gain)**: Prediction accuracy increases monotonically with context length, until saturation.

These hypotheses move the benchmark beyond performance reporting: they test structural claims about how LLMs approximate, or fail to approximate the individuality of human reasoning.

**Validation roadmap.** To substantiate these hypotheses, we highlight four key *control experiments* that serve as evidence of individuality; later sections return to each in detail.

---

**What counts as evidence of individuality? (control experiments)**

- **Population Prior Baseline** – predict only from aggregate distributions (see Sec. 6.1).
- **Identity Shuffle Control** – shuffle person–context pairs (see Sec. 6.2).
- **Per-Person Leave-One-Out** – use partial history to predict held-out responses (see Sec. 4.2).
- **Context-Length Ablation** – vary context size to test information gain (see Sec. 5).

---

## 3 HUGAGENT BENCHMARK

Grounded in the theoretical setup in Section 2, we now introduce **HugAgent**, which translates these principles into concrete tasks (Sec. 3.2), a scalable data collection pipeline (Sec. 3.3), and evaluation protocols (Sec. 3.5).

**Task 1 – Belief State Inference**

**Context** (QA pairs from transcript)

**Q:** Technological reliability?
**A:** If the system can't be trusted, people will oppose it

**Q:** Main cause of support?
**A:** Safety is the priority… people want to feel secure

**Q:** Community trust?
**A:** Depends on past experience… takes time to build confidence

**Q:** Equity & bias?
**A:** Important to show fairness… treat all groups the same

**Question**

Based on this person's responses, what do they think about the effect of **greater accuracy and reliability** in technology on **support for surveillance cameras**?
**A. POSITIVE effect** ✅
B. NEGSTIVE effect

**Task 2 – Belief Dynamic Update**

**Context** (QA pairs from transcript)

**Q:** Technological reliability?
**A:** If the system can't be trusted, people will oppose it

**Q:** Main cause of support?
**A:** Safety is the priority… people want to feel secure

**Q:** Community trust?
**A:** Depends on past experience… takes time to build confidence

**Q:** Equity & bias?
**A:** Important to show fairness… treat all groups the same

**Question**

If surveillance footage was kept only 2 days and checked only after problems, how would this affect your opinion?
**Scale [1–10] Answer: 8**

How much do these reasons matter for your opinion?
**Scale [1–5]**
-Privacy **Answer: 4**
-Freedom / Autonomy **Answer: 1**
-Policy Persistence **Answer: 1**

Figure 2: **Two benchmark tasks.** Task 1 (Belief State Inference) infers stance and reasons from prior context; Task 2 (Belief Dynamics Update) predicts stance shifts and its reasoning under new evidence.

### 3.1 DESIGN PRINCIPLES

- **Open-ended but deeper.** Emphasize depth over breadth: semi-structured dialogue with targeted follow-ups surfaces individuality while avoiding over-scaffolding; ground truth comes from self-reports (Kvale, 1996; Srivastava et al., 2023; Park et al., 2024).

- **Two observable proxies.** We evaluate (i) belief state inference and (ii) belief dynamics update—tractable targets that avoid requiring exact trace imitation; cf. proxy-label benchmarks (Geva et al., 2021; Ho et al., 2022; Guerdan et al., 2023).

- **Dual track.** Human interviews provide ecological realism; synthetic agents provide controlled stress tests via scripted causal belief graphs—preventing circularity and enabling scale.

- **Human ceiling.** Test–retest reliability defines the upper bound, aligning with psychology standards (Nunnally & Bernstein, 1994; Cronbach, 1970) and recent large-scale simulations (Toubia et al., 2025; Park et al., 2024).

See Appendix L for extended discussion.

### 3.2 TASK DEFINITION

We formalize reasoning adaptation as predicting how an individual's belief state changes under new evidence.

**Belief representation.** A belief at time $t$ is $b_t = (s_t, \mathbf{w}_t)$, where $s_t \in \{1, \dots, 10\}$ is a stance score and $\mathbf{w}_t$ is a distribution over $K$ reason weights.

**Belief update.** Given evidence $e$, an update operator $U$ produces $b_{t+1} = U(b_t, e)$. In humans, $b_{t+1}$ comes from self-reports; in synthetic agents, from scripted rules with known ground truth.

**Tasks.** We instantiate two tasks: (i) **Belief State Inference**: predict $(s_t, \mathbf{w}_t)$ from prior responses. (ii) **Belief Dynamics Update**: predict $(s_{t+1}, \mathbf{w}_{t+1})$ given $(b_t, e)$. Figure 2 shows concrete examples of both tasks in HugAgent.

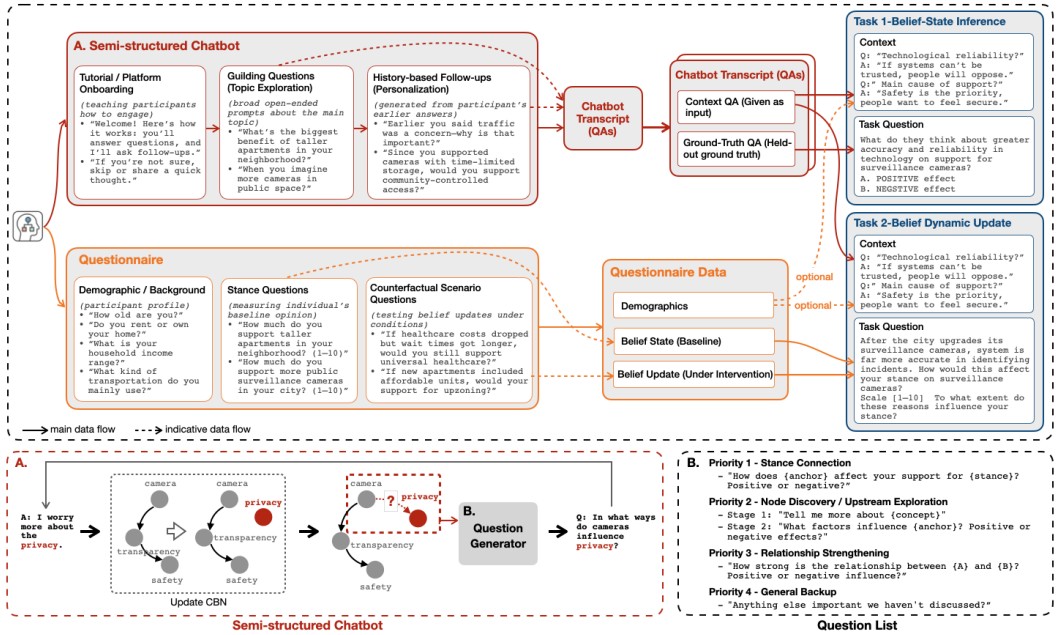

Figure 3: **HugAgent** benchmark pipeline. Inputs (demographics, questionnaires, and transcripts) flow through two components: a questionnaire that provides demographic anchors, stance baselines, and counterfactual updates, and a semi-structured chatbot that elicits individualized reasoning. Together these elements define two benchmark tasks: (i) *belief state inference*—recovering stance and factor polarity from context, and (ii) *belief dynamics update*—predicting stance shifts and reweighting under new evidence. (A) The chatbot maintains a causal belief network of factors, used to identify the most critical nodes and edges for follow-up. (B) A question generator derives targeted, context-specific probes from this network to structure the dialogue.

### 3.3 BUILDING HUGAGENT: SCALABLE ELICITATION OF INDIVIDUAL REASONING

HugAgent is built through a two–stage pipeline (Figure 3).

**1. The *questionnaire* stage** collects demographics, a baseline stance ($s_t$, 1–10), reason weights ($\mathbf{w}_t$, 1–5), and counterfactual interventions with updated stances and reasons. These structured responses provide gold labels for *Belief Dynamics Update* and serve as anchors for aligning free text to factors.

**2. The *chatbot* stage** elicits 8–20 question–answer pairs through semi-structured interview, combining open-ended elaborations (*Context QAs*) with concise polarity judgments (*GT QAs*). This setup captures both participants' belief, reasoning styles and explicit preferences on each decision factor. Each transcript thus supports both benchmark tasks: *Belief State Inference* (using Context and GT QAs) and *Belief Dynamics Update* (using Context QAs, questionnaire responses upon interventions). Survey-provided updates are never revealed in dialogue, preventing leakage.

Finally, we establish a human reliability ceiling through test–retest elicitation of a subset of items, reporting intra-individual consistency with Intraclass Correlation (ICC) and quadratic-weighted kappa (QWK) with 95% confidence intervals. Further design choices, intervention phrasing, prompt templates, and quality-control rules are detailed in Appendix L.

### 3.4 DATASET STATISTICS

Applying this pipeline yields HugAgent, a dataset spanning three socially salient domains: *healthcare*, *surveillance*, and *zoning*. These domains were chosen for their ecological validity, diversity of viewpoints, and internally rich trade-offs (e.g., affordability vs. neighborhood character, privacy vs. safety).

**Human track.** From over 120 participants, we retained 54 whose survey and interview data met predefined quality-control criteria (see Appendix O). Task 1 (*belief state inference*) contains 356

labeled questions, and Task 2 (*belief dynamics update*) contains 1,386 items, distributed as shown in Table 1.

**Synthetic track.** We construct a parallel synthetic track by assigning each agent a *scripted causal belief network* (CBN) that defines its reasoning structure and deterministic update rules. All synthetic agents follow the same survey and interview protocol as in the human track, allowing direct comparison under identical tasks. In total, the benchmark includes 500 synthetic agents, from which we use a stratified subset of 50 agents for computation and analysis. Unless otherwise specified, all experiments reported in this paper are conducted on this subset. Both the full and subset data are included in the benchmark release. Detailed construction procedures are provided in Appendix P.

| Task / Domain | Human Track (N=54) | | | | Synthetic Track (N=500) | | | |
|---|---|---|---|---|---|---|---|---|
| | Health | Surv. | Zoning | Total | Health | Surv. | Zoning | Total |
| Belief State Inference | 108 | 122 | 126 | 356 | 1,303 | 1,297 | 1,302 | 3,902 |
| Belief Dynamics Update | 472 | 364 | 550 | 1,386 | 3,888 | 2,610 | 3,818 | 10,316 |
| **Total Items** | 580 | 486 | 676 | 1,742 | 5,191 | 3,907 | 5,120 | 14,218 |

Table 1: Dataset statistics by task and domain. Human track (N=54 participants) provides ecological validity; synthetic track (N=500 scripted agents) scales coverage.

### 3.5 EVALUATION PROTOCOLS

Finally, we highlight the evaluation protocols that ensure HugAgent's utility as a scientific benchmark.

**Evaluation metrics.** We evaluate *belief state inference* using **accuracy**, the proportion of exact matches with ground-truth labels. For *belief dynamics update*, we report four metrics: (i) **accuracy**, the proportion of predictions within a tolerance band of the true response ($\pm 1$ for 5-point, $\pm 2$ for 10-point scales); (ii) **mean absolute error (MAE)**, the average magnitude of deviation from the ground truth, with all responses normalized to a 5-point scale for consistency; (iii) **directional accuracy** measures whether the predicted belief update matches the ground truth in direction(increase, decrease, or no change); and (iv) **average to individual (ATI)** score, following the GLUE (Wang et al., 2018) and SuperGLUE (Wang et al., 2019) scoring paradigm, we derive an overall score via hierarchical aggregation, where normalized task-specific metrics (accuracy, MAE, directional accuracy) are combined into belief dynamics update and averaged with belief state inference, with human and random guess performance used as upper and lower bounds for normalization. Formal definitions and computation details are given in Appendix S.1.

**Leakage control.** We masked attribution targets, drew interventions from external surveys, and presented each item independently with minimal-overlap prompts (see Appendix K.1 for templates).

**Human baselines.** We re-contacted a subset of participants for a short-interval (14-day) test–retest study. Across all sessions, 54 participants contributed data. Of these, 18 completed the retest, and 13 were retained following a demographic consistency check. *Belief State Inference* yielded an accuracy of **84.84%** (SD = 8.90, 95% CI: [80.00, 89.68]). *Belief Dynamics Update* achieved an accuracy of **85.66%** (SD = 7.66, 95% CI: [80.91, 90.40]) and a mean absolute error of **0.68** (SD = 0.20, 95% CI: [0.55, 0.80]). The directional accuracy was **88.92%** (SD = 9.89, 95% CI: [82.09, 96.24]). These scores establish a human consistency ceiling, as outlined in Section 3.1, against which model performance can be benchmarked.

## 4 MAIN RESULTS

### 4.1 BASELINES

We compare models against clear anchors: (i) an **upper bound** defined by **real human performance**, measured through test–retest consistency (see Section 3.1); (ii) a **lower bound** defined by **random guessing**, reflecting chance performance; and (iii) a set of strong **pretrained language models**, including `GPT`, `Gemini`, `LLaMA`, and `Qwen`, which serve as high-performing but non-agentic baselines without explicit memory, personalization, or retrieval components.

To evaluate the role of agent-like structure in modeling belief reasoning, we further include two **agent-style LLM baselines** that incorporate memory or retrieval. The first is the **Generative Agents** baseline, reproduced following the setup of Park et al. (2023), using `Qwen2.5-32B-instruct` as the base model. The second group includes two variants of **retrieval-augmented generation (RAG)**. The first, **RAG**, follows the standard setup (Lewis et al., 2020), replacing the original full QA

context with the top-$k$ retrieved QA pairs ($k = 5$). The second, **RAG with Full Context**, appends the retrieved QA pairs to the original input, allowing the model to jointly condition on both. This variant evaluates whether retrieval can serve as an auxiliary signal rather than a substitute for agent-specific context. Detailed settings and full results are provided in Appendix C.

## 4.2 OVERALL PERFORMANCE

Tables 2 summarize performance. We evaluate using the metrics introduced in Section 3.5. For **belief state inference**, best-performing LLMs approach but do not match human accuracy, trailing by 4–6 points. Open-source LLaMA and Qwen rival GPT-4o, while smaller or less aligned models lag significantly. For **belief dynamics update**, gaps are larger: models frequently mispredict the direction of stance change or fail to adjust reason weights, yielding higher error than human baseline.

| Model | Belief State Inference | | Belief Dynamics Update | | | | | | ATI (% ↑) |
| | Acc. (% ↑) | | Acc. (% ↑) | | MAE (↓) | | Dir. Acc. (% ↑) | | |
| | Real | Synth. | Real | Synth. | Real | Synth. | Real | Synth. | Real |
|---|---|---|---|---|---|---|---|---|---|
| Human | **84.84** | — | **85.66** | — | **0.68** | — | **88.92** | — | **100.00** |
| *OpenAI Models* | | | | | | | | | |
| GPT-4o | $74.66^{\pm0.24}$ | $67.93^{\pm0.32}$ | $63.11^{\pm0.14}$ | $58.61^{\pm0.28}$ | $1.29^{\pm0.00}$ | $1.44^{\pm0.00}$ | $82.27^{\pm1.02}$ | $76.83^{\pm0.46}$ | $67.29^{\pm0.74}$ |
| GPT-5-mini | $75.30^{\pm0.79}$ | $62.78^{\pm0.70}$ | $58.21^{\pm0.50}$ | $57.25^{\pm0.73}$ | $1.43^{\pm0.01}$ | $1.47^{\pm0.01}$ | $77.02^{\pm2.02}$ | $73.69^{\pm3.76}$ | $61.53^{\pm2.09}$ |
| o3-mini | $75.12^{\pm0.78}$ | $69.67^{\pm0.52}$ | $64.54^{\pm0.32}$ | $56.48^{\pm0.23}$ | $\mathbf{1.22}^{\pm0.01}$ | $1.45^{\pm0.01}$ | $71.29^{\pm2.87}$ | $64.34^{\pm4.24}$ | $60.92^{\pm1.68}$ |
| *Other Closed-Source Models* | | | | | | | | | |
| Gemini 2.0 Flash | $69.95^{\pm0.12}$ | $59.66^{\pm0.48}$ | $60.55^{\pm0.08}$ | $54.73^{\pm0.13}$ | $1.35^{\pm0.00}$ | $1.48^{\pm0.00}$ | $\mathbf{83.31}^{\pm0.19}$ | $68.58^{\pm0.00}$ | $59.76^{\pm0.18}$ |
| DeepSeek-R1 | $75.55^{\pm0.34}$ | $70.75^{\pm0.81}$ | $64.88^{\pm0.33}$ | $\mathbf{61.16}^{\pm0.26}$ | $1.29^{\pm0.01}$ | $\mathbf{1.38}^{\pm0.01}$ | $79.69^{\pm0.91}$ | $76.20^{\pm1.10}$ | $67.20^{\pm1.03}$ |
| Qwen-plus | $\mathbf{77.57}^{\pm0.44}$ | $67.81^{\pm1.06}$ | $58.93^{\pm0.22}$ | $55.29^{\pm0.14}$ | $1.40^{\pm0.00}$ | $1.49^{\pm0.00}$ | $77.17^{\pm0.61}$ | $75.82^{\pm4.48}$ | $65.48^{\pm0.87}$ |
| Qwen-max | $77.40^{\pm0.27}$ | $67.74^{\pm0.81}$ | $58.86^{\pm0.25}$ | $55.20^{\pm0.24}$ | $1.40^{\pm0.00}$ | $1.49^{\pm0.01}$ | $77.17^{\pm0.48}$ | $76.54^{\pm3.28}$ | $65.21^{\pm0.49}$ |
| *Open-Source Models* | | | | | | | | | |
| LLaMA 3.3 70B | $76.64^{\pm0.18}$ | $71.74^{\pm0.38}$ | $\mathbf{67.57}^{\pm0.20}$ | $58.35^{\pm0.22}$ | $1.24^{\pm0.00}$ | $1.49^{\pm0.00}$ | $79.56^{\pm0.36}$ | $74.93^{\pm2.48}$ | $\mathbf{69.84}^{\pm0.27}$ |
| Qwen2.5-32B-instr. | $77.17^{\pm0.32}$ | $68.21^{\pm0.30}$ | $58.96^{\pm0.05}$ | $55.37^{\pm0.24}$ | $1.40^{\pm0.00}$ | $1.49^{\pm0.01}$ | $76.88^{\pm0.29}$ | $75.32^{\pm4.62}$ | $64.71^{\pm0.23}$ |
| Qwen2.5-7B-instr. | $77.18^{\pm0.73}$ | $67.81^{\pm0.58}$ | $58.82^{\pm0.20}$ | $55.33^{\pm0.12}$ | $1.40^{\pm0.00}$ | $1.49^{\pm0.00}$ | $77.12^{\pm0.80}$ | $74.95^{\pm4.38}$ | $64.83^{\pm1.21}$ |
| *Memory-Augmented Baselines* | | | | | | | | | |
| RAG[1] | $75.46^{\pm0.47}$ | $63.06^{\pm0.86}$ | $51.90^{\pm0.39}$ | $51.85^{\pm0.20}$ | $1.57^{\pm0.05}$ | $1.56^{\pm0.00}$ | $72.25^{\pm0.57}$ | $66.70^{\pm2.71}$ | $54.96^{\pm0.87}$ |
| RAG-FC[1] | $77.56^{\pm0.38}$ | $72.82^{\pm0.91}$ | $59.97^{\pm0.23}$ | $59.34^{\pm0.40}$ | $1.39^{\pm0.00}$ | $1.45^{\pm0.01}$ | $76.80^{\pm0.80}$ | $76.81^{\pm1.53}$ | $65.65^{\pm0.61}$ |
| Generative Agents[1] | $76.19^{\pm0.36}$ | $\mathbf{73.48}^{\pm0.31}$ | $58.22^{\pm0.43}$ | $55.43^{\pm0.15}$ | $1.40^{\pm0.01}$ | $1.49^{\pm0.00}$ | $76.13^{\pm2.45}$ | $\mathbf{82.76}^{\pm0.36}$ | $62.43^{\pm1.74}$ |
| *Non-Learning Baselines* | | | | | | | | | |
| Global Majority | $65.77^{\pm0.00}$ | $64.30^{\pm0.00}$ | $58.18^{\pm0.00}$ | $54.60^{\pm0.00}$ | $2.54^{\pm0.00}$ | $2.37^{\pm0.00}$ | $17.93^{\pm0.00}$ | $21.95^{\pm0.00}$ | $4.44^{\pm0.00}$ |
| Random Guess | $51.89^{\pm3.96}$ | $50.62^{\pm3.16}$ | $43.12^{\pm0.78}$ | $50.36^{\pm1.16}$ | $1.88^{\pm0.05}$ | $1.62^{\pm0.03}$ | $46.74^{\pm3.28}$ | $29.78^{\pm13.18}$ | $0.00^{\pm5.80}$ |

Table 2: Average performance across all domains for *Belief State Inference* and *Belief Dynamics Update*. Results are reported as *mean±std* over 5 runs. Best-performing model (below human upper bound) per column is highlighted in **bold**.

## 5 MAIN FINDINGS

### FINDING 1: PRESERVING IDENTITY ACROSS DOMAINS IS HARDER THAN EXPECTED

| Task | Model | Health → Surv | Surv → Zone | Zone → Health | Avg |
|---|---|---|---|---|---|
| Belief State Inference (Accuracy % ↑) | Qwen2.5-32B-instr. | $53.52^{\pm1.52}$ | $58.69^{\pm1.10}$ | $55.08^{\pm1.20}$ | $55.76^{\pm1.09}$ |
| | GPT-4o | $58.52^{\pm0.77}$ | $65.41^{\pm1.58}$ | $51.75^{\pm1.03}$ | $58.56^{\pm0.82}$ |
| Belief Dynamics Update (Accuracy % ↑) | Qwen2.5-32B-instr. | $44.41^{\pm0.98}$ | $24.84^{\pm0.60}$ | $30.15^{\pm0.35}$ | $33.13^{\pm0.48}$ |
| | GPT-4o | $49.49^{\pm0.87}$ | $42.47^{\pm0.50}$ | $41.96^{\pm0.30}$ | $44.64^{\pm0.36}$ |
| Belief Dynamics Update (MAE ↓) | Qwen2.5-32B-instr. | $1.69^{\pm0.01}$ | $1.94^{\pm0.01}$ | $2.05^{\pm0.01}$ | $1.89^{\pm0.01}$ |
| | GPT-4o | $1.55^{\pm0.01}$ | $1.60^{\pm0.00}$ | $1.71^{\pm0.00}$ | $1.62^{\pm0.00}$ |

Table 3: Cross-domain swap test: models are trained with QA context from one domain and evaluated on another domain for the *same participant*. We report performance for *Healthcare → Surveillance*, *Surveillance → Zoning*, *Zoning → Healthcare*, and their average. Reported as *mean ± std* over 5 runs.

To evaluate whether models can generalize a person's contextual information across domains, we conduct a cross-domain swap test. Each model is given QA context from one domain and evaluated on another domain for the same participant (for example, using context from *Healthcare* and testing on *Surveillance*). Based on Table 2, we selected GPT-4o and Qwen2.5-32B-instruct as representative models from two categories: closed-source SOTA and strong open-source baseline. Under cross-domain transfer, model performance degrades substantially compared to within-domain

---

[1]RAG (Lewis et al., 2020), RAG-FC = RAG with Full Context (Lewis et al., 2020), Generative Agents (Park et al., 2023).

evaluation. `GPT-4o` achieves an average of **58.56**% on *belief state inference*, compared to its within-domain score of **74.66**%, while `Qwen2.5-32B-instruct` drops from **77.17**% to **55.76**%. In the *Belief Dynamics Update* task, `GPT-4o` declines from **63.11**% to **44.64**%. The degradation is most pronounced in *belief dynamics update*, where compounding errors accumulate across domains. These findings suggest that current models depend heavily on domain-specific linguistic and contextual cues, resulting in limited cross-domain reasoning transfer. This underscores the importance of evaluating *within-person, cross-domain consistency* as a key indicator of robust generalization, and suggests that improving transferability requires focusing on essential, domain-relevant context rather than surface-level correlations.

FINDING 2: MORE CONTEXT DOESN'T ALWAYS HELP

| Model | Context | Belief State Inference | Belief Dynamics Update | | | ATI (% ↑) |
|---|---|---|---|---|---|---|
| | | Acc. (% ↑) | Acc. (% ↑) | MAE (↓) | Dir. Acc. (% ↑) | |
| GPT-4o | 5 QAs | $71.68^{\pm 0.47}$ | $\mathbf{64.19}^{\pm 0.22}$ | $\mathbf{1.23}^{\pm 0.00}$ | $\mathbf{83.22}^{\pm 0.40}$ | $64.46^{\pm 0.81}$ |
| | 10 QAs | $70.65^{\pm 0.49}$ | $64.22^{\pm 0.20}$ | $1.25^{\pm 0.00}$ | $80.05^{\pm 0.47}$ | $60.46^{\pm 0.98}$ |
| | 20+ QAs | $\mathbf{74.66}^{\pm 0.24}$ | $63.11^{\pm 0.14}$ | $1.29^{\pm 0.00}$ | $82.27^{\pm 1.02}$ | $\mathbf{67.29}^{\pm 0.74}$ |
| Gemini 2.0 Flash | 5 QAs | $63.87^{\pm 0.13}$ | $\mathbf{62.56}^{\pm 0.18}$ | $\mathbf{1.31}^{\pm 0.00}$ | $76.54^{\pm 0.28}$ | $46.87^{\pm 0.35}$ |
| | 10 QAs | $65.40^{\pm 0.25}$ | $62.00^{\pm 0.13}$ | $1.31^{\pm 0.00}$ | $75.62^{\pm 0.70}$ | $48.24^{\pm 0.76}$ |
| | 20+ QAs | $\mathbf{69.95}^{\pm 0.12}$ | $60.55^{\pm 0.08}$ | $1.35^{\pm 0.00}$ | $\mathbf{83.31}^{\pm 0.19}$ | $\mathbf{59.76}^{\pm 0.18}$ |
| Qwen2.5-32B-instr. | 5 QAs | $68.84^{\pm 0.73}$ | $\mathbf{61.79}^{\pm 0.46}$ | $\mathbf{1.31}^{\pm 0.00}$ | $\mathbf{78.45}^{\pm 0.85}$ | $55.28^{\pm 1.46}$ |
| | 10 QAs | $73.02^{\pm 1.66}$ | $60.73^{\pm 0.46}$ | $1.35^{\pm 0.00}$ | $78.20^{\pm 0.70}$ | $60.58^{\pm 2.58}$ |
| | 20+ QAs | $\mathbf{77.17}^{\pm 0.32}$ | $58.96^{\pm 0.05}$ | $1.40^{\pm 0.00}$ | $76.88^{\pm 0.29}$ | $\mathbf{64.71}^{\pm 0.23}$ |

Table 4: Scaling context length for both *Belief State Inference* and *Belief Dynamics Update*. Reported as mean ± std over 5 runs. Best results per column within each model group are highlighted in **bold**.

To examine how context length influences model performance across tasks, we varied the number of Context QAs (5, 10, 20+). Based on Table 2, we selected `GPT-4o`, `Gemini2.0Flash`, and `Qwen2.5-32B-instruct` as representative models from three categories: closed-source SOTA, lightweight efficient, and strong open-source baseline.

The core result is that belief state inference accuracy rises monotonically with additional dialogue and saturates at 20+ questions, whereas belief dynamics update accuracy peaks at 5–10 questions before declining. Longer context provide richer cues for recovering belief states but also introduce noise that impairs belief updating. This asymmetry suggests that context length benefits *belief state inference* through scale, while *belief dynamics update* is sensitive to **contextual interference**, where **accumulated history acts as distractor noise** rather than causing a general cognitive overload. We next investigate whether these context-length effects stem from surface-level pattern matching rather than genuine identity modeling.

# 6 WHY IT HAPPENS: DIAGNOSTIC ABLATIONS

Section 5 revealed a **consistent failure to preserve identity across domains**. Here we ask *why*: is it because models never learned to use personal information, or because they use it in an *associative, non-generalizable* way? To answer this, we conduct two diagnostic ablations.

## 6.1 POPULATION PRIOR VS. INDIVIDUAL CONTEXT

| Method | Belief State Inference | Belief Dynamics Update | | | ATI (% ↑) |
|---|---|---|---|---|---|
| | Accuracy (% ↑) | Accuracy (% ↑) | MAE (↓) | Dir. Acc. (% ↑) | |
| GPT-4o (No-Context) | $58.49^{\pm 0.43}$ | $39.83^{\pm 0.45}$ | $1.70^{\pm 0.01}$ | $75.59^{\pm 0.81}$ | $26.98^{\pm 1.01}$ |
| GPT-4o (Full-Context) | $74.66^{\pm 0.24}$ | $63.11^{\pm 0.14}$ | $1.29^{\pm 0.00}$ | $82.27^{\pm 1.02}$ | $67.29^{\pm 0.74}$ |
| Qwen2.5-32B-instr. (No-Context) | $50.92^{\pm 0.55}$ | $32.12^{\pm 0.24}$ | $1.88^{\pm 0.01}$ | $69.00^{\pm 1.07}$ | $6.88^{\pm 0.81}$ |
| Qwen2.5-32B-instr. (Full-Context) | $77.17^{\pm 0.32}$ | $58.96^{\pm 0.05}$ | $1.40^{\pm 0.00}$ | $76.88^{\pm 0.29}$ | $64.71^{\pm 0.23}$ |

Table 5: Comparison of *No-Context* (population prior) and *Full-Context* (with individual transcripts) settings. Reported as *mean ± std* over 5 runs.

We first test whether providing individual context leads to better model performance than relying solely on population-level priors. The *No-Context* setting provides only demographic background, while the *Full-Context* setting additionally includes transcripts and survey answers. As shown in Table 5, `GPT-4o` achieves substantially higher performance when provided with individual context:

its belief-state inference accuracy increases from **58.49% with demographic priors** to **74.66% with full context**, and its belief-dynamics-update accuracy from **39.83%** to **63.11%**. The consistent gains in *Full-Context* indicate that models leverage individual cues rather than population priors, showing that the benchmark captures identity-sensitive reasoning rather than general demographic trends.

## 6.2 CROSS-PERSON GENERALIZATION

| Method | Belief State Inference | Belief Dynamics Update | | | ATI (% ↑) |
|---|---|---|---|---|---|
| | Accuracy (% ↑) | Accuracy (% ↑) | MAE (↓) | Dir. Acc. (% ↑) | |
| GPT-4o (Cross-Person) | $54.44^{\pm 2.30}$ | $39.30^{\pm 0.22}$ | $1.93^{\pm 0.01}$ | $63.56^{\pm 0.76}$ | $10.29^{\pm 3.88}$ |
| GPT-4o (Same-Person) | $74.66^{\pm 0.24}$ | $63.11^{\pm 0.14}$ | $1.29^{\pm 0.00}$ | $82.27^{\pm 1.02}$ | $67.29^{\pm 0.74}$ |
| Qwen2.5-32B-instr. (Cross-Person) | $60.44^{\pm 0.47}$ | $38.16^{\pm 0.44}$ | $2.02^{\pm 0.01}$ | $69.17^{\pm 1.06}$ | $22.15^{\pm 1.24}$ |
| Qwen2.5-32B-instr. (Same-Person) | $77.17^{\pm 0.32}$ | $58.96^{\pm 0.05}$ | $1.40^{\pm 0.00}$ | $76.88^{\pm 0.29}$ | $64.71^{\pm 0.23}$ |

Table 6: Cross-person swap test: QA context from one participant is used to predict another partici-pant's responses (Cross-Person) versus the same participant (Same-Person). Results are reported as *mean ± std* over 5 runs, averaged across domains.

We then test whether the observed gains reflect genuine identity modeling or simply the benefit of having richer, more fine-grained context. In this *Cross-Person* setting, QA context from one participant is used to predict another's responses within the same domain. As shown in Table 6, performance drops sharply, e.g., `GPT-4o` achieves only **39.30%** belief-dynamics-update accuracy with an MAE of **1.93**, showing that models fail to generalize when identity cues are mismatched. These results suggest that improvements in the *Full-Context* setting arise from learning identity-specific patterns rather than simply benefiting from additional contextual detail.

**Summary.** Together with the findings in Section 5, these ablations reveal that current models' failure to preserve identity across domains is not because they ignore individual context in inference, but rather implies a reliance on associative context matching instead of identity-consistent reasoning.

## 7 ERROR ANALYSIS: BIAS PATTERNS AND SOURCES OF FAILURE

**Domain dependence and cross-domain generalization.** Performance varies systematically across domains. *Within-domain*, *belief state inference* peaks in *Surveillance* for both models (`LLaMA3.3-70B` **80.66%**; `GPT-5-mini` **76.89%**). For *belief dynamics update*, `LLaMA3.3-70B` performs best on *Zoning* (**73.09%**, MAE **1.14**), whereas `GPT-5-mini` peaks on *Healthcare* (**59.92%**, MAE **1.37**).

*Cross-domain* transfer yields sharp degradation in performance. Using *Surveillance* context to predict *Zoning*, *belief dynamics updates* for `Qwen2.5-32B-instr` drop from in-domain *Zoning* (**62.25%**; MAE **1.33**) to **24.84%** (MAE **1.94**); `GPT-4o` falls from **64.04%** (MAE **1.26**) to **42.47%** (MAE **1.60**).

**Implication.** Models do not share structural regularities in belief updating across domains, suggesting that LLMs' *belief dynamics* rely more on corpus-specific semantic co-occurrence patterns than on abstract causal or cognitive mechanisms. Therefore, the degradation in cross-domain performance is not merely a domain shift issue, but rather reveals that current models **lack a domain-general inductive bias** for individualized human simulation.

**Directional error decomposition.** We decompose directional errors into two components: (1) **change-detection error**, reflecting the model's failure to detect whether a belief change has occurred, and (2) **direction-inference error**, reflecting the model's failure to predict the direction of that change (*increase*, *decrease*, or *no change*). Detailed definitions are provided in Appendix S.1.

Across systems, *change-detection* errors predominate. In the *Healthcare* domain, `GPT-4o` at-tains a change-detection accuracy of **49.36%**, compared to **88.89%** for direction-inference (Dir. Acc. 77.03%). Similarly, `Qwen2.5-32B-Instruct` records **33.19%** versus **84.29%** (Dir. Acc. 68.96%). This asymmetry persists across domains, yielding consistently high accuracy in *direction-inference* but notably lower accuracy in *change-detection*, resulting in only moderate *overall directional accuracy*.

The prevalence of change-detection errors suggests that models often preserve the *sign* of stance changes once detected but fail to capture *when* such updates should occur. Rather than inverting polarity, they tend to remain static despite contextual cues, reflecting an over-regularization or change-averse bias toward *no change*. This pattern also holds for strong models such as `Qwen-Max` and `GPT-4o`, which achieve only moderate directional accuracy (≈77–82%), lagging behind human performance (Dir. Acc. 88.92%) by 7–12 percentage points (Table 2). Notably, a low mean absolute

error (e.g., o3-mini, MAE 1.22) does not necessarily imply higher directional accuracy, confirming that magnitude alignment alone does not ensure correct update directionality.

**Implication.** LLMs tend to be *directionally accurate* yet *change-averse*, often preferring to preserve prior beliefs rather than initiating updates without strong contextual evidence. This pattern reflects an implicit bias toward stability over adaptability. It suggests that future models **require mechanisms for calibrated change detection**, potentially through more continuous output representations or confidence-aware update designs.

## 8 MITIGATION STRATEGIES: INSIGHTS AND GUIDING PRINCIPLES

Our error analysis revealed *structured failure modes* that recur across model families. Rather than proposing ad-hoc fixes, we position these as **diagnostic insights**, each grounded in preceding analyses (Sections 6–7), motivating guiding principles for future work.

**(i) Identity-consistent generalization gap.** Diagnosed in Section 6.2. Performance gains in *Full-Context* disappear under *Cross-Person* settings, suggesting associative matching rather than person-specific reasoning. *Principle:* Learn identity-conditioned representations and persistent persona states (e.g., per-person latent slots/adapters) with cross-domain consistency regularization.

**(ii) Change-aversion bias.** Evident from directional error decomposition in Section 7. Models often preserve prior beliefs even when contextual cues signal change, revealing weak mechanisms for deciding *when* to update beliefs. *Principle:* Incorporate meta-cognitive gating, such as confidence-weighted update triggers or explicit belief-state decay, to calibrate change sensitivity and mirror human mechanisms of gradual belief revision.

**(iii) Context-prioritization deficit.** Observed across context-length ablations and generalization tests (Sections 6.1). Extending context length alone does not enhance reasoning; models lack mechanisms for *context prioritization* and *signal compression*. *Principle:* Future systems should implement *adaptive span selection*, focusing computation on high-signal evidence regions to emulate human selective attention and working-memory limits.

**Synthesis.** Viewed as a whole, these principles underscore HugAgent's role not only as a benchmark but also as a *diagnostic tool*, revealing structured biases that standard metrics (accuracy, MAE) often obscure. Looking ahead, HugAgent also establishes a reproducible testbed for exploring and evaluating mitigation strategies, laying the groundwork for a broader research agenda.

## 9 DISCUSSION AND OPEN CHALLENGES

**Topic selection is not neutral.** Benchmarks not only measure performance, they define what counts as *understanding*. *What we ask* shapes *who* a model appears to simulate. Highly controversial topics elicit richer value trade-offs and sharper updates; homogeneous topics compress variation and inflate apparent accuracy. We propose treating topic choice as an **explicit experimental variable**: (i) curate pairs of domains with high vs. low opinion dispersion; (ii) report results stratified by a simple *controversy index* $C$ (e.g., stance variance + polarity entropy) so scores are comparable across corpora; (iii) add *dilemma framings* (e.g., fairness vs. safety) to probe value conflicts rather than single-axis opinions. Practically, this turns "topic selection" from a hidden confound into a **controlled factor** that the community can measure and replicate.

**From intuition to deliberation: eliciting System 2 without over-scaffolding.** Open-ended interviews risk collapsing into shallow System 1 reactions or over-guided narratives. Accuracy peaks at modest context length, hinting at overload beyond that point. We propose **lightweight controls** that foster deliberation while preserving individuality: (i) brief *tension probes* ("what would change your mind?"), (ii) contrasting *time-pressure* vs. *reflection*, (iii) tracking simple *deliberation proxies* (latency, self-corrections, counterfactual mentions). These signals do not enforce a single "correct" trace, but provide **anchors** for interpreting model alignment with human update logic.

**Comparative benchmarks, transfer, and ethics-by-design.** Out-of-domain generalization remains one of the most fundamental yet persistent challenges for models of human reasoning. We propose a **comparative benchmark protocol** that (i) pairs controversial vs. homogeneous topics, (ii) contrasts intuition vs. deliberation, and (iii) evaluates *within-person, cross-domain* transfer as a core metric. Preregistered topic panels and per-item metadata (controversy index $C$, tension flags, latency) will aid comparability. Ethically, tension induction must be *consentful and minimal*, with disclosed framings, capped length, and opt-out. The challenge is designing evaluation that **respects persons** while probing the hard cases where individuality matters most.

**Synthesis.** Ultimately, *moving reasoning from average to individual* requires not only better models, but also **better questions**: topics that surface trade-offs, protocols that invite careful thinking, and benchmarks that reward cross-domain fidelity rather than single-domain fit.

## 10 LIMITATIONS AND CONCLUSION

Our study is constrained by a modest human sample due to the resource demands of collecting deeper, higher-fidelity real human data, yet this is complemented by a synthetic track that provides controlled variation and an open, end-to-end pipeline for collecting and processing human subject data. This design enables the community to expand the dataset both in domain coverage and scale *at minimal cost*.

Even so, **HugAgent** establishes a benchmark for *average-to-individual reasoning adaptation*, uniting ecological validity with scalability. Across three key dimensions: (i) advancing from *averaged* to *individualized* reasoning, (ii) from *behavioral mimicry* to *cognitive alignment*, and (iii) from *vignette-based* to *open-ended* data, **HugAgent** redefines the methodological frontier of *human behavior simulation*. To our knowledge, it is the first benchmark to explicitly define and operationalize the anthropic question of how large language models simulate both human reasoning and behavior. Built as a fully open and extensible agent-simulation benchmark, **HugAgent** sets a new standard for evaluating how large language models simulate human-like reasoning and behavior under open-ended conditions.

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

## A    RELATED WORK

**Social Simulation, Digital Twins, and Population Panels.**    A growing line of work simulates societies with LLM agents. Early "silicon samples" use LMs to approximate human samples in stylized tasks (Argyle et al., 2023), while *Generative Agents* extend to rich daily-life environments with memory and social coordination (Park et al., 2023). This has scaled to population panels—e.g., simulations of *1,000 people* (Park et al., 2024) and digital-twin datasets such as *Twin-2K-500* (Toubia et al., 2025)—as well as community/role-play platforms for real-time social interaction (Kaiya et al., 2023) and personification benchmarks (Xie et al., 2024b;a). Complementing these persona-based evaluations, PersonalLLM introduces a benchmark and dataset for modeling fine-grained individual preferences, emphasizing heterogeneous user reward structures and personalization challenges(Zollo et al., 2025).These approaches provide breadth and largely *static* outcome measures; **HugAgent** adds *depth* via think-aloud transcripts that trace reasoning trajectories, counterfactual interventions to test belief updates, and a human test–retest reliability ceiling to anchor claims (Wurgaft et al., 2025; Nunnally & Bernstein, 1994; Cronbach, 1970).

**Social Reasoning and Theory of Mind.**    Work on Theory of Mind (ToM) in AI draws from developmental psychology tests such as the false-belief task (Wimmer & Perner, 1983), Sally-Anne (Baron-Cohen et al., 1985), and Strange Stories (Happé, 1994), later reformulated as computational tasks (Nematzadeh et al., 2018; Rabinowitz et al., 2018). Scaled language models brought ToM into broad benchmarks (Le et al., 2019; Hewitt & Cohen, 2021; Srivastava et al., 2023; Chen et al., 2024) and inspired synthetic testbeds such as BigToM (Gandhi et al., 2023), HI-TOM (He et al., 2023), FANToM (Kim et al., 2023), and MMToM-QA (Jin et al., 2024). UniToMBench (Thiyagarajan et al., 2025) advances synthetic ToM evaluation by integrating multi-interaction task structures and evolving narrative scenarios, offering a unified benchmark that highlights strengths and failures of current LLMs in belief and emotion reasoning. More recent directions ground ToM in dialogues and social contexts (Chan et al., 2024; Sap et al., 2019b; Strachan et al., 2024), or frame it through Bayesian belief attribution (Ying et al., 2024). Yet these benchmarks remain synthetic, vignette-based, and decontextualized, missing ecological and demographic variability (Wang et al., 2025; Stewart et al., 2017).

Parallel lines in AI reasoning emphasize world and agent models: causal world modeling (Wong et al., 2023; Lake et al., 2017; Ellis et al., 2020) and the LAW framework, which coordinates world, agent, and language models (Hu & Shu, 2023). Within this framing, HugAgent extends ToM evaluation by asking whether models can map natural language into personalized belief states and update them consistently under interventions, bridging synthetic ToM tasks and socially grounded reasoning.

## B  AUXILIARY DATASET

Our core benchmark is designed around interview transcripts, where each data point consists of demographic information, a context in the form of question–answer pairs, and ground-truth first-person self-reports. This setup defines a clean belief inference language task: given demographic cues and conversational context, models must infer individual beliefs and predict reactions.

To complement this benchmark, we also release an auxiliary demographics-only dataset (39 users, each with survey responses and demographic attributes). While these records do not support direct belief inference, they enable principled baselines and transfer settings. Concretely, we implemented a demographic–linear regression model based on survey responses, and further tested Qwen-Plus on the main belief dynamics update task with auxiliary supervision from a subset of 15 users. Results from both the demographic-linear baseline and the augmented Qwen-Plus setting are reported in Table 7, illustrating how population-level priors can inform personalized inference.

Table 7: Results on the auxiliary demographics-only setting. Comparison of a demographic-only prior baseline and Qwen-Plus with auxiliary supervision across three domains. Higher accuracy and lower MAE indicate better performance.

| Domain | Demographic-only prior | | Qwen-Plus w/ auxiliary data | |
|---|---|---|---|---|
| | Accuracy | MAE | Accuracy | MAE |
| Healthcare | 0.130 | 2.954 | 0.524 | 1.792 |
| Zoning | 0.202 | 2.954 | 0.474 | 1.765 |
| Surveillance | 0.081 | 2.797 | 0.589 | 1.819 |

## C  HugAgent Benchmark Details

We introduce the **HugAgent Benchmark** (*HUman-Grounded Theory of Mind*), a new evaluation suite for reasoning fidelity in generative agents. HugAgent formalizes the task of *causal BN reconstruction from human interviews*, and provides (1) a dataset of annotated causal belief graphs derived from natural language Q&A, and (2) multi-level evaluation metrics (node, edge, motif) to assess structural alignment between human and agent reasoning.

Unlike prior benchmarks focused on behavioral imitation or chain-of-thought generation, HugAgent directly evaluates whether agents can reconstruct the latent causal structures that underlie human judgments. This benchmark responds to recent concerns that LLM-based agents risk flattening individual identity representations by grounding evaluation in structured, human-annotated causal beliefs.

### Data schema and cognitive grounding

Inspired by cognitive science, we use causal BNs to represent the reasoning structures underlying human decision-making. Rather than modeling surface discourse, our schema captures latent causal dynamics by explicitly linking belief variables, affective states, and behavioral intentions in a directed graph. This design supports psychologically grounded and structurally coherent representations of human reasoning.

**Schema Design**  Each participant's causal BN is represented as a structured JSON object composed of three main components: `nodes`, `edges`, and `qa_history`. This schema is designed to encode causal beliefs extracted from interviews, with each element indexed by a unique ID to enable motif analysis, simulation, and evidence tracing.

NODES  Each node represents a belief concept and includes a `label`, a model-generated `confidence` score (ranging from 0.0 to 1.0), and a list of `source_qa` IDs that support the node's existence. Nodes also track their `incoming_edges` and `outgoing_edges` for efficient graph traversal.

EDGES  Edges capture directed causal links between nodes. Each edge includes a `source` node ID, a `target` node ID, and an `aggregate_confidence` score reflecting the model's overall belief in the causal connection. A `modifier` (in the range $[-1.0, 1.0]$) represents the direction and strength of influence: positive values indicate causal support, negative values indicate inhibition. Each edge is backed by a list of individual QA-based evidence entries with associated confidence scores.

QA HISTORY  The `qa_history` component stores raw interview responses, mapping each QA pair to its corresponding extracted causal relations. Each QA entry includes the original question and answer texts, as well as a list of `extracted_pairs`, where each pair links a source node to a target node with a confidence score.

This data structure supports fine-grained analysis of belief formation, causal reasoning, and evidence provenance across participants.

### Dataset Construction

We collected over 100 interviews from participants recruited through the Prolific platform. Topics—such as urban upzoning, surveillance cameras, and universal healthcare—were chosen to elicit reflective, ecologically valid reasoning.

### Transcript Collection

To construct structured causal BNs from qualitative interviews, we developed a **semi-structured, cognitively grounded elicitation framework**. This framework guides LLMs in extracting interpretable causal structures from natural language dialogue and generating follow-up questions that balance open-ended exploration with targeted inquiry (Chickering, 2002; Pohontsch & Meyer, 2015).

### Human Annotation Protocol

We asked annotators to label causal BNs using soft labels, capturing graded beliefs and allowing for variation across annotators. Our human-in-the-loop annotation tool supports annotators in assigning confidence scores to each node and edge(Zhang et al., 2023).

Annotators were also recruited from Prolific(Stewart et al., 2017). During selection, we followed three principles: (1) double-blind annotation, (2) matching annotators with similar backgrounds, and (3) using shared guidelines to maintain consistency. Annotation instructions were carefully designed to ensure reproducibility and interpretability.

## EVALUATION SETTINGS

All models are evaluated under a consistent inference setup. We fix the random seed to **42** and set the temperature to **0.1** for all experiments. Models from the `GPT` and `Gemini` series are executed in *batch inference* mode, while all other models use *real-time completion* inference. All outputs are constrained using **function calling** to ensure structured and valid responses. Prompts adopt a pure *in-context learning* format without any examples or reasoning demonstrations.

For the **Generative Agents** setting, following (Park et al., 2023), we employ `Qwen2.5-32B-instruct` to analyze the dialogue transcript and generate three high-level expert reflections that serve as auxiliary reasoning cues during inference. For the two retrieval-augmented variants of RAG(Lewis et al., 2020), we adopt a TF-IDF retriever to identify the top five most relevant QA pairs.

## HUMAN BASELINES

We re-contacted a subset of participants for a short-interval (14-day) test–retest study. Across all sessions, 54 participants contributed data. Of these, 18 completed the retest, and 13 were retained following a demographic consistency check.

*Belief State Inference* yielded an accuracy of **84.84%** (SD = 8.90, 95% CI: [80.00, 89.68]). By topic, accuracies were **87.50%** for *Surveillance* (SD = 8.29, 95% CI: [82.81, 92.19]), **81.54%** for *Zoning* (SD = 15.11, 95% CI: [73.32, 89.75]), and **84.53%** for *Healthcare* (SD = 13.90, 95% CI: [76.97, 92.09]).

*Belief Dynamics Update* achieved an accuracy of **85.66%** (SD = 7.66, 95% CI: [80.91, 90.40]) and a mean absolute error of **0.68** (SD = 0.20, 95% CI: [0.55, 0.80]). Across topics, accuracy was **85.75%** for *Healthcare* (SD = 10.68), **85.33%** for *Surveillance* (SD = 12.98), and **85.86%** for *Zoning* (SD = 10.08). Corresponding mean absolute errors were **0.66** (SD = 0.28, 95% CI: [0.49, 0.83]) for Healthcare, **0.72** (SD = 0.35, 95% CI: [0.50, 0.93]) for Surveillance, and **0.67** (SD = 0.28, 95% CI: [0.49, 0.84]) for Zoning.

In the *Belief Dynamics Update* tasks, we decompose directional accuracy into two components: (1) **change detection**—the model's ability to detect whether a belief change has occurred, and (2) **direction inference**—the model's ability to predict the direction of that change (increase, decrease, or no change) (detailed definitions are provided in Appendix S.1).

Overall, the directional accuracy was **88.92%** (SD = 9.89, 95% CI: [82.09, 96.24]). The results across topics are summarized as follows:

- **Healthcare:** change detection = 80.00% (SD = 25.82, 95% CI: [61.53, 98.47]); direction inference = 96.67% (SD = 10.54, 95% CI: [89.13, 100.00]); directional accuracy = 91.67% (SD = 11.06, 95% CI: [83.76, 99.58]).

- **Surveillance:** change detection = 90.00% (SD = 16.10, 95% CI: [78.48, 100.00]); direction inference = 88.33% (SD = 19.33, 95% CI: [74.51, 100.00]); directional accuracy = 88.83% (SD = 15.15, 95% CI: [77.99, 99.67]).

- **Zoning:** change detection = 80.00% (SD = 23.31, 95% CI: [63.33, 96.67]); direction inference = 90.00% (SD = 21.08, 95% CI: [74.92, 100.00]); directional accuracy = 87.00% (SD = 18.14, 95% CI: [74.03, 99.97]).

HUMAN DEMOGRAPHIC BREAKDOWN

To provide a clearer view of the diversity represented in our human participant cohort, we report detailed demographic distributions across age, income, education level, and occupation. As shown below, the 54 participants span a broad range of backgrounds, supporting the use of this sample for individualized reasoning analysis.

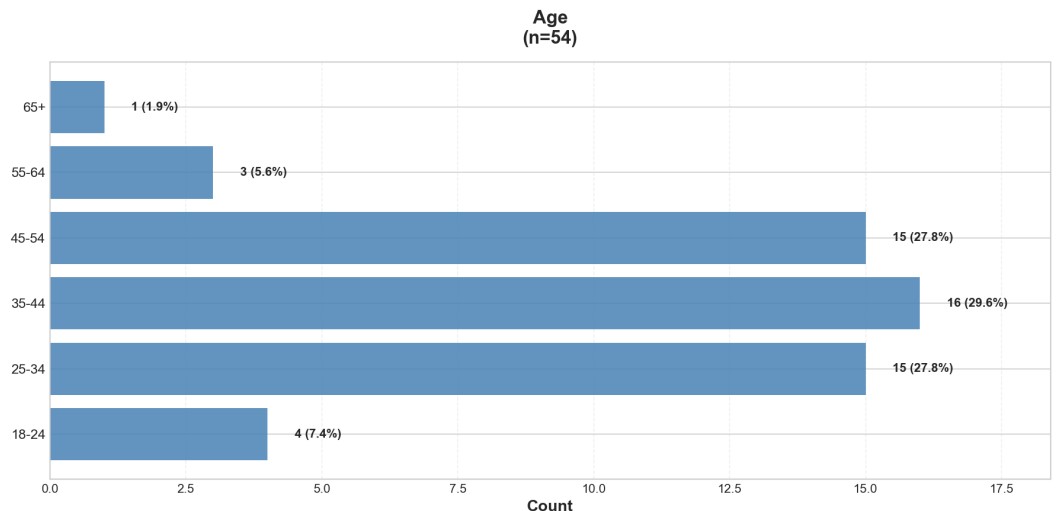

Figure 4: Age distribution of the 54 participants.

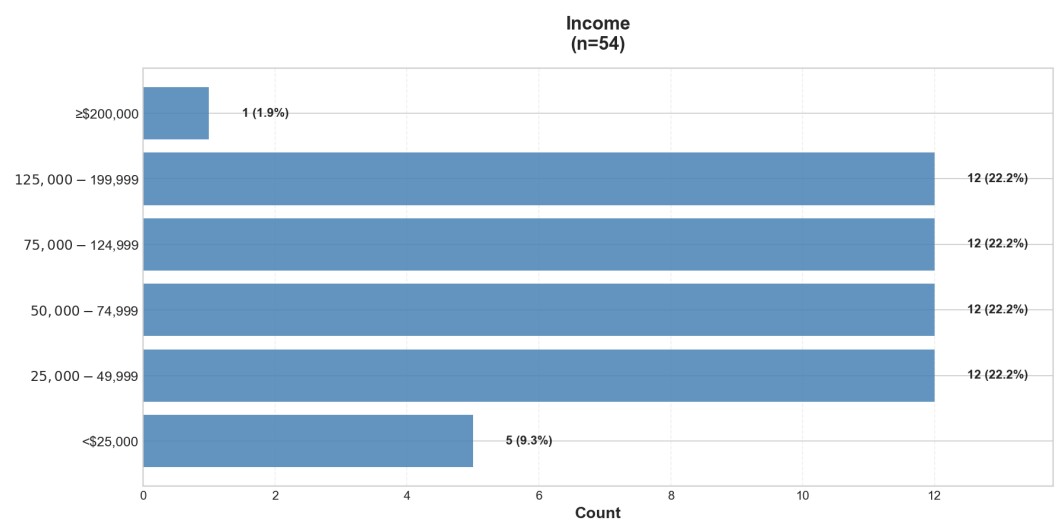

Figure 5: Income distribution of the 54 participants.

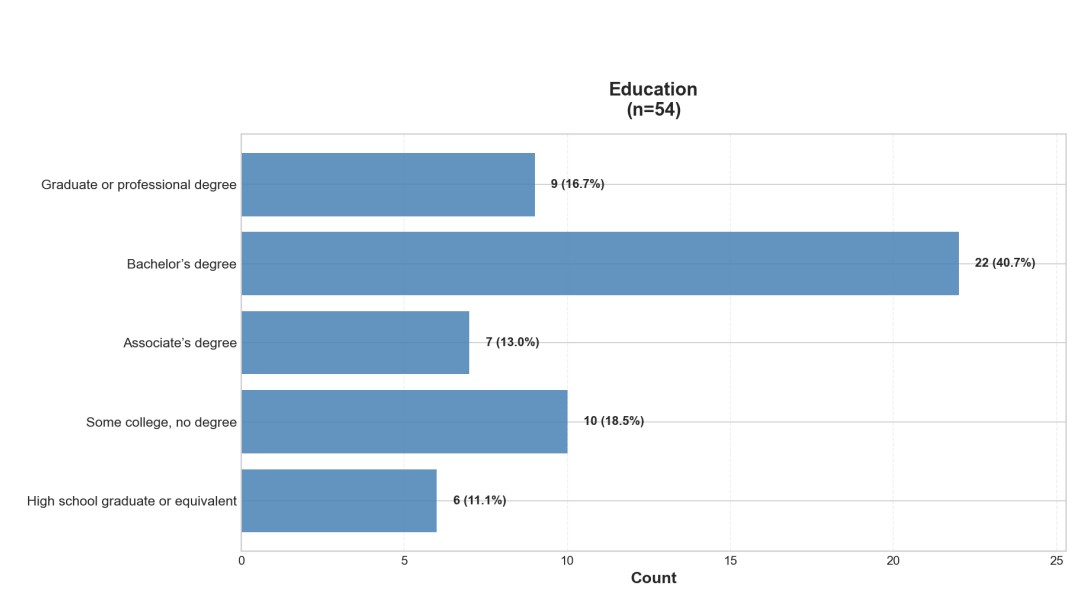

Figure 6: Education level distribution of the 54 participants.

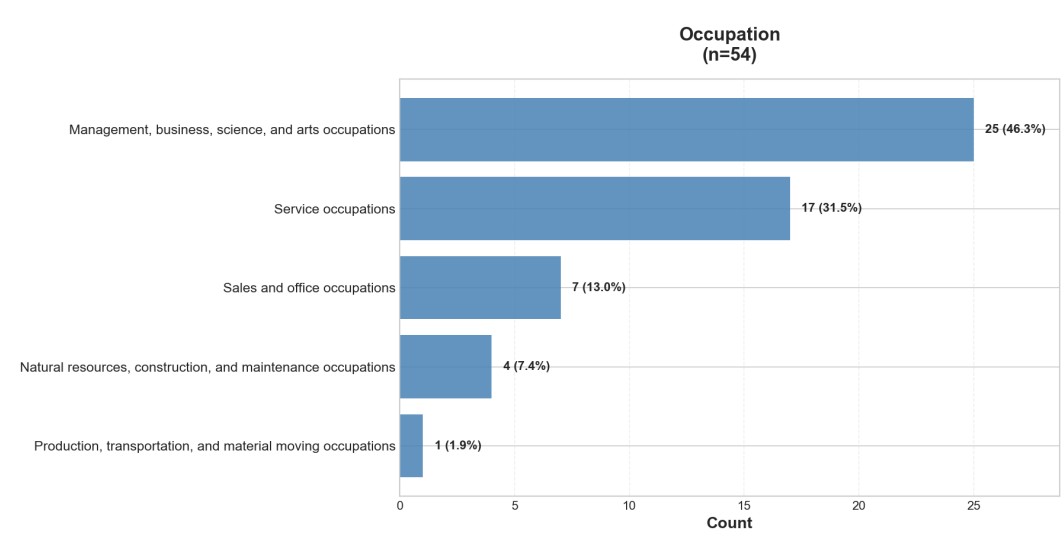

Figure 7: Occupation distribution of the 54 participants.

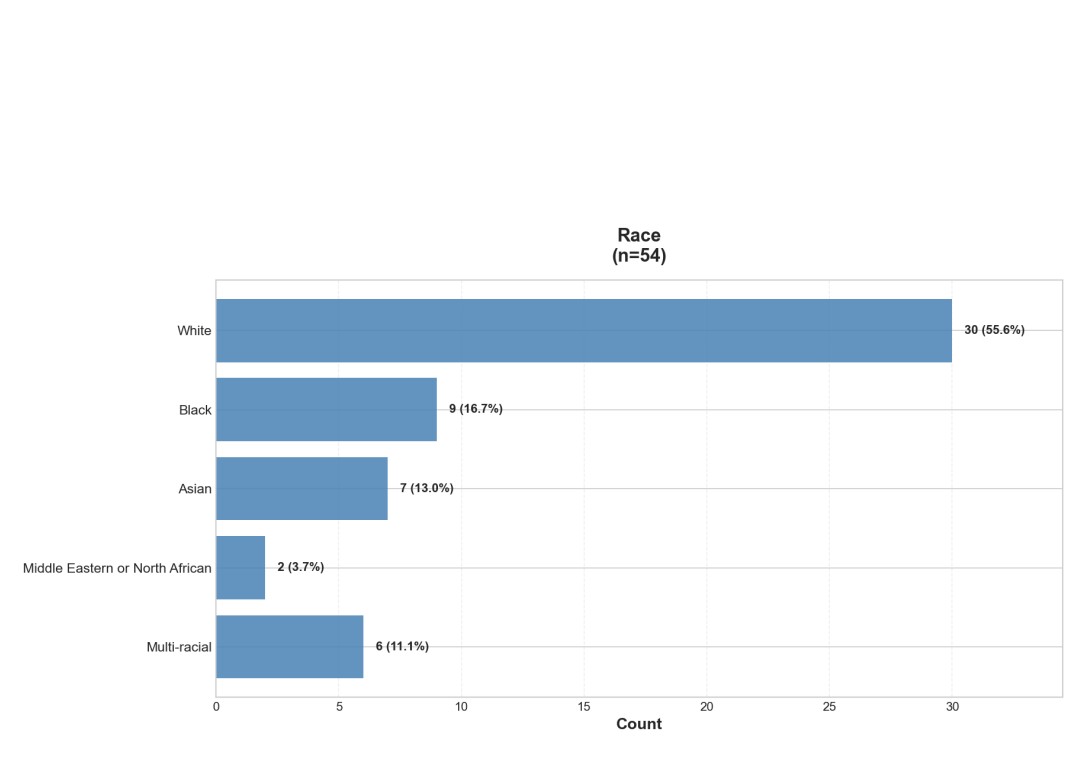

Figure 8: Race distribution of the 54 participants.

FULL RESULTS

| Model | Belief State Inference | Belief Dynamics Update | | | ATI (% ↑) |
|---|---|---|---|---|---|
| | Acc. (% ↑) | Acc. (% ↑) | MAE (↓) | Dir. Acc. (% ↑) | |
| Human | **84.84** | **85.66** | **0.68** | **88.92** | **100.00** |
| *OpenAI Models* | | | | | |
| GPT-4o | $74.66^{\pm0.24}$ | $63.11^{\pm0.14}$ | $1.29^{\pm0.00}$ | $82.27^{\pm1.02}$ | $67.29^{\pm0.74}$ |
| GPT-5-mini | $75.30^{\pm0.79}$ | $58.21^{\pm0.50}$ | $1.43^{\pm0.01}$ | $77.02^{\pm2.02}$ | $61.53^{\pm2.09}$ |
| GPT5.1 Rea. High | $73.36^{\pm2.16}$ | $67.13^{\pm0.46}$ | $\mathbf{1.17}^{\pm0.01}$ | $80.94^{\pm3.28}$ | $64.66^{\pm3.79}$ |
| o3-mini | $75.12^{\pm0.78}$ | $64.54^{\pm0.32}$ | $1.22^{\pm0.01}$ | $71.29^{\pm2.87}$ | $60.92^{\pm1.68}$ |
| *Other Closed-Source Models* | | | | | |
| Claude Sonnet 4.5 | $76.04^{\pm0.01}$ | $\mathbf{68.61}^{\pm0.09}$ | $1.18^{\pm0.01}$ | $78.73^{\pm0.10}$ | $67.39^{\pm0.11}$ |
| Gemini 2.0 Flash | $69.95^{\pm0.12}$ | $60.55^{\pm0.08}$ | $1.35^{\pm0.00}$ | $\mathbf{83.31}^{\pm0.19}$ | $59.76^{\pm0.18}$ |
| Gemini 2.5 Pro | $75.45^{\pm1.13}$ | $64.87^{\pm0.39}$ | $1.27^{\pm0.01}$ | $78.65^{\pm1.17}$ | $64.29^{\pm1.58}$ |
| DeepSeek-R1 | $75.55^{\pm0.34}$ | $64.88^{\pm0.33}$ | $1.29^{\pm0.01}$ | $79.69^{\pm0.91}$ | $67.20^{\pm1.03}$ |
| Qwen-plus | $\mathbf{77.57}^{\pm0.44}$ | $58.93^{\pm0.22}$ | $1.40^{\pm0.00}$ | $77.17^{\pm0.61}$ | $65.48^{\pm0.87}$ |
| Qwen-max | $77.40^{\pm0.27}$ | $58.86^{\pm0.25}$ | $1.40^{\pm0.00}$ | $77.17^{\pm0.48}$ | $65.21^{\pm0.49}$ |
| *Open-Source Models* | | | | | |
| LLaMA 3.3 70B | $76.64^{\pm0.18}$ | $67.57^{\pm0.20}$ | $1.24^{\pm0.00}$ | $79.56^{\pm0.36}$ | $\mathbf{69.84}^{\pm0.27}$ |
| Qwen2.5-32B-instr. | $77.17^{\pm0.32}$ | $58.96^{\pm0.05}$ | $1.40^{\pm0.00}$ | $76.88^{\pm0.29}$ | $64.71^{\pm0.23}$ |
| Qwen2.5-7B-instr. | $77.18^{\pm0.73}$ | $58.82^{\pm0.20}$ | $1.40^{\pm0.00}$ | $77.12^{\pm0.80}$ | $64.83^{\pm1.21}$ |
| *Memory-Augmented Baselines* | | | | | |
| RAG[1] | $75.46^{\pm0.47}$ | $51.90^{\pm0.39}$ | $1.57^{\pm0.05}$ | $72.25^{\pm0.57}$ | $54.96^{\pm0.87}$ |
| RAG-FC[1] | $77.56^{\pm0.38}$ | $59.97^{\pm0.23}$ | $1.39^{\pm0.00}$ | $76.80^{\pm0.80}$ | $65.65^{\pm0.61}$ |
| Generative Agents[2] | $76.19^{\pm0.36}$ | $58.22^{\pm0.43}$ | $1.40^{\pm0.01}$ | $76.13^{\pm2.45}$ | $62.43^{\pm1.74}$ |
| *Non-Learning Baselines* | | | | | |
| Global Majority | $65.77^{\pm0.00}$ | $58.18^{\pm0.00}$ | $2.54^{\pm0.00}$ | $17.93^{\pm0.00}$ | $4.44^{\pm0.00}$ |
| Random Guess | $51.89^{\pm3.96}$ | $43.12^{\pm0.78}$ | $1.88^{\pm0.05}$ | $46.74^{\pm3.28}$ | $0.00^{\pm5.80}$ |

Table 8: Average performance across all domains for *Belief State Inference* and *Belief Dynamics Update*. Results are reported as *mean±std* over 5 runs. Best-performing model (below human upper bound) per column is highlighted in **bold**. Real data only.

[1]RAG (Lewis et al., 2020), RAG-FC = RAG with Full Context (Lewis et al., 2020).
[2]Generative Agents (Park et al., 2023).

| Model | Human Dataset (N=54) | | | | | | | | |
| --- | --- | --- | --- | --- | --- | --- | --- | --- | --- |
| | Belief State Inference Acc. (% ↑) | | | Belief Dynamics Update Acc. (% ↑) | | | MAE (↓) | | |
| | Health | Surveil. | Zoning | Health | Surveil. | Zoning | Health | Surveil. | Zoning |
| **Human** | **84.53** | **87.50** | **81.54** | **85.75** | **85.33** | **85.86** | **0.66** | **0.72** | **0.67** |
| *OpenAI Models* | | | | | | | | | |
| GPT-4o | $71.11^{\pm0.77}$ | $75.25^{\pm0.90}$ | $77.62^{\pm0.35}$ | $61.57^{\pm0.19}$ | $63.74^{\pm0.58}$ | $64.04^{\pm0.20}$ | $1.27^{\pm0.01}$ | $1.35^{\pm0.01}$ | $1.26^{\pm0.01}$ |
| GPT5.1 Rea. High | $72.22^{\pm1.01}$ | $77.54^{\pm1.91}$ | $70.32^{\pm0.46}$ | $63.43^{\pm0.39}$ | $68.19^{\pm1.13}$ | $69.78^{\pm0.65}$ | $1.22^{\pm0.02}$ | $1.16^{\pm0.02}$ | $1.12^{\pm0.01}$ |
| GPT-5-mini | $73.15^{\pm2.36}$ | $76.89^{\pm0.90}$ | $75.87^{\pm0.43}$ | $59.92^{\pm1.05}$ | $55.16^{\pm1.09}$ | $59.56^{\pm0.73}$ | $1.37^{\pm0.02}$ | $1.54^{\pm0.02}$ | $1.40^{\pm0.01}$ |
| o3-mini | $69.26^{\pm1.21}$ | $77.38^{\pm1.70}$ | $\mathbf{78.73}^{\pm0.87}$ | $60.08^{\pm0.28}$ | $62.53^{\pm1.16}$ | $71.02^{\pm0.79}$ | $1.31^{\pm0.02}$ | $1.29^{\pm0.03}$ | $\mathbf{1.07}^{\pm0.01}$ |
| *Other Closed-Source Models* | | | | | | | | | |
| Claude Sonnet 4.5 | $73.15^{\pm0.01}$ | $81.15^{\pm0.01}$ | $73.81^{\pm0.01}$ | $\mathbf{65.21}^{\pm0.08}$ | $\mathbf{72.69}^{\pm0.22}$ | $67.93^{\pm0.09}$ | $\mathbf{1.17}^{\pm0.01}$ | $\mathbf{1.13}^{\pm0.01}$ | $1.24^{\pm0.01}$ |
| Gemini 2.0 Flash | $63.89^{\pm0.00}$ | $68.20^{\pm0.37}$ | $77.78^{\pm0.01}$ | $56.23^{\pm0.12}$ | $62.25^{\pm0.37}$ | $63.16^{\pm0.10}$ | $1.39^{\pm0.01}$ | $1.33^{\pm0.01}$ | $1.33^{\pm0.01}$ |
| Gemini 2.5 Pro | $73.70^{\pm1.99}$ | $78.03^{\pm1.31}$ | $74.60^{\pm1.00}$ | $62.12^{\pm0.36}$ | $62.53^{\pm0.57}$ | $69.96^{\pm0.53}$ | $1.29^{\pm0.01}$ | $1.31^{\pm0.01}$ | $1.21^{\pm0.01}$ |
| DeepSeek-R1-0528 | $76.11^{\pm0.41}$ | $75.41^{\pm1.16}$ | $74.76^{\pm0.66}$ | $61.44^{\pm0.58}$ | $68.30^{\pm0.63}$ | $64.91^{\pm0.41}$ | $1.33^{\pm0.00}$ | $1.27^{\pm0.02}$ | $1.26^{\pm0.01}$ |
| Qwen-plus_2025-07-28 | $\mathbf{78.70}^{\pm0.65}$ | $76.39^{\pm1.22}$ | $77.62^{\pm0.66}$ | $56.74^{\pm0.59}$ | $57.86^{\pm0.15}$ | $62.18^{\pm0.18}$ | $1.44^{\pm0.01}$ | $1.43^{\pm0.01}$ | $1.34^{\pm0.00}$ |
| Qwen-max_2024-10-15 | $78.52^{\pm0.77}$ | $76.39^{\pm0.37}$ | $77.30^{\pm0.43}$ | $56.78^{\pm0.64}$ | $57.69^{\pm0.27}$ | $62.11^{\pm0.35}$ | $1.43^{\pm0.01}$ | $1.44^{\pm0.01}$ | $1.34^{\pm0.00}$ |
| *Open-Source Models* | | | | | | | | | |
| LLaMA_3.3_70B | $70.74^{\pm0.51}$ | $\mathbf{80.66}^{\pm0.45}$ | $77.78^{\pm0.00}$ | $64.28^{\pm0.38}$ | $65.33^{\pm0.23}$ | $\mathbf{73.09}^{\pm0.31}$ | $1.31^{\pm0.01}$ | $1.29^{\pm0.01}$ | $1.14^{\pm0.01}$ |
| Qwen2.5-32B-instr. | $77.96^{\pm0.41}$ | $76.56^{\pm0.45}$ | $76.98^{\pm0.79}$ | $56.82^{\pm0.41}$ | $57.80^{\pm0.25}$ | $62.25^{\pm0.16}$ | $1.43^{\pm0.00}$ | $1.43^{\pm0.01}$ | $1.33^{\pm0.01}$ |
| Qwen2.5-7B-instr | $78.33^{\pm0.51}$ | $75.74^{\pm2.06}$ | $77.46^{\pm0.43}$ | $56.82^{\pm0.41}$ | $57.64^{\pm0.45}$ | $62.00^{\pm0.34}$ | $1.43^{\pm0.01}$ | $1.43^{\pm0.01}$ | $1.34^{\pm0.01}$ |
| *Memory-Augmented Baselines* | | | | | | | | | |
| RAG[1] | $71.67^{\pm0.51}$ | $78.36^{\pm1.37}$ | $76.35^{\pm0.35}$ | $50.64^{\pm0.58}$ | $53.68^{\pm0.60}$ | $51.38^{\pm0.24}$ | $1.60^{\pm0.01}$ | $1.50^{\pm0.00}$ | $1.60^{\pm0.00}$ |
| RAG-FC[1] | $77.22^{\pm0.83}$ | $76.72^{\pm0.93}$ | $\mathbf{78.73}^{\pm0.87}$ | $58.73^{\pm0.55}$ | $59.34^{\pm0.34}$ | $61.85^{\pm0.15}$ | $1.43^{\pm0.01}$ | $1.41^{\pm0.01}$ | $1.33^{\pm0.01}$ |
| Generative Agents[2] | $76.11^{\pm0.41}$ | $77.21^{\pm0.37}$ | $75.24^{\pm0.66}$ | $57.67^{\pm0.76}$ | $55.66^{\pm0.69}$ | $61.35^{\pm0.38}$ | $1.43^{\pm0.01}$ | $1.44^{\pm0.01}$ | $1.34^{\pm0.00}$ |

Table 9: Results on **Human** dataset for *belief state inference* and *belief dynamics update* tasks across three policy topics (Health, Surveillance, Zoning). Values are mean$^{\pm\text{std}}$ over 5 runs. For *belief dynamics update*, both Accuracy and MAE are reported separately. Best non-human results per column are in **bold**.

| Model | Synthetic Dataset (N=50) | | | | | | | | |
| --- | --- | --- | --- | --- | --- | --- | --- | --- | --- |
| | Belief State Inference Acc. (% ↑) | | | Belief Dynamics Update Acc. (% ↑) | | | MAE (↓) | | |
| | Health | Surveil. | Zoning | Health | Surveil. | Zoning | Health | Surveil. | Zoning |
| *OpenAI Models* | | | | | | | | | |
| GPT-4o | $60.76^{\pm0.00}$ | $75.00^{\pm0.77}$ | $68.03^{\pm0.58}$ | $52.71^{\pm0.29}$ | $\mathbf{65.00}^{\pm0.35}$ | $58.11^{\pm0.48}$ | $1.72^{\pm0.00}$ | $1.18^{\pm0.01}$ | $1.41^{\pm0.01}$ |
| GPT-5-mini | $50.38^{\pm1.88}$ | $72.39^{\pm1.24}$ | $65.79^{\pm1.34}$ | $51.09^{\pm0.91}$ | $64.44^{\pm0.46}$ | $56.23^{\pm1.63}$ | $1.74^{\pm0.01}$ | $1.21^{\pm0.02}$ | $1.46^{\pm0.02}$ |
| o3-mini | $58.23^{\pm1.55}$ | $\mathbf{78.48}^{\pm1.19}$ | $72.30^{\pm1.47}$ | $46.90^{\pm0.81}$ | $58.46^{\pm0.28}$ | $64.07^{\pm0.37}$ | $1.80^{\pm0.02}$ | $1.29^{\pm0.00}$ | $\mathbf{1.26}^{\pm0.01}$ |
| *Other Closed-Source Models* | | | | | | | | | |
| Gemini 2.0 Flash | $51.90^{\pm0.00}$ | $63.48^{\pm1.24}$ | $63.61^{\pm0.45}$ | $45.28^{\pm0.19}$ | $62.96^{\pm0.00}$ | $55.93^{\pm0.22}$ | $1.77^{\pm0.00}$ | $1.18^{\pm0.00}$ | $1.49^{\pm0.00}$ |
| DeepSeek-R1-0528 | $62.78^{\pm1.13}$ | $75.87^{\pm1.42}$ | $73.61^{\pm0.69}$ | $\mathbf{60.52}^{\pm0.18}$ | $64.75^{\pm0.40}$ | $58.21^{\pm0.45}$ | $\mathbf{1.59}^{\pm0.01}$ | $\mathbf{1.15}^{\pm0.01}$ | $1.40^{\pm0.01}$ |
| Qwen-plus_2025-07-28 | $63.80^{\pm1.44}$ | $67.83^{\pm1.97}$ | $71.80^{\pm0.73}$ | $45.11^{\pm0.37}$ | $64.94^{\pm0.17}$ | $55.83^{\pm0.53}$ | $1.80^{\pm0.00}$ | $1.18^{\pm0.01}$ | $1.50^{\pm0.01}$ |
| Qwen-max_2024-10-15 | $62.78^{\pm2.12}$ | $68.48^{\pm0.00}$ | $71.97^{\pm0.37}$ | $45.20^{\pm0.44}$ | $64.57^{\pm0.14}$ | $55.83^{\pm0.46}$ | $1.80^{\pm0.00}$ | $1.18^{\pm0.01}$ | $1.50^{\pm0.01}$ |
| *Open-Source Models* | | | | | | | | | |
| LLaMA_3.3_70B | $61.27^{\pm1.13}$ | $76.09^{\pm0.00}$ | $\mathbf{77.87}^{\pm0.00}$ | $51.44^{\pm0.65}$ | $57.65^{\pm0.34}$ | $\mathbf{65.96}^{\pm0.21}$ | $1.82^{\pm0.01}$ | $1.32^{\pm0.00}$ | $1.32^{\pm0.00}$ |
| Qwen2.5-32B-instr. | $64.30^{\pm0.56}$ | $68.70^{\pm0.49}$ | $71.64^{\pm0.93}$ | $45.24^{\pm0.39}$ | $\mathbf{65.00}^{\pm0.17}$ | $55.88^{\pm0.54}$ | $1.80^{\pm0.00}$ | $1.18^{\pm0.00}$ | $1.50^{\pm0.01}$ |
| Qwen2.5-7B-instr | $63.80^{\pm1.13}$ | $68.48^{\pm1.33}$ | $71.15^{\pm0.90}$ | $45.11^{\pm0.29}$ | $64.81^{\pm0.22}$ | $56.08^{\pm0.25}$ | $1.80^{\pm0.00}$ | $1.18^{\pm0.01}$ | $1.50^{\pm0.01}$ |
| *Memory-Augmented Baselines* | | | | | | | | | |
| RAG[1] | $51.39^{\pm1.44}$ | $67.61^{\pm0.90}$ | $70.16^{\pm1.37}$ | $43.49^{\pm0.28}$ | $61.98^{\pm0.40}$ | $50.07^{\pm0.32}$ | $1.85^{\pm0.00}$ | $1.22^{\pm0.01}$ | $1.61^{\pm0.00}$ |
| RAG-FC[1] | $68.10^{\pm1.65}$ | $76.09^{\pm1.09}$ | $74.26^{\pm1.10}$ | $53.41^{\pm0.45}$ | $64.26^{\pm0.46}$ | $60.35^{\pm0.59}$ | $1.70^{\pm0.01}$ | $1.20^{\pm0.01}$ | $1.44^{\pm0.01}$ |
| Generative Agents[2] | $\mathbf{68.61}^{\pm0.57}$ | $77.39^{\pm0.91}$ | $74.43^{\pm0.37}$ | $46.24^{\pm0.68}$ | $64.26^{\pm0.26}$ | $55.78^{\pm0.21}$ | $1.79^{\pm0.01}$ | $1.17^{\pm0.00}$ | $1.52^{\pm0.01}$ |

Table 10: Results on **Synthetic** dataset for *belief state inference* and *belief dynamics update* tasks across three policy topics (Health, Surveillance, Zoning). Values are mean$^{\pm\text{std}}$ over 5 runs. For *belief dynamics update*, both Accuracy and MAE are reported separately. Best results per column are in **bold**.

---

[1]RAG (Lewis et al., 2020), RAG-FC = RAG with Full Context (Lewis et al., 2020).
[2]Generative Agents (Park et al., 2023).

## RESULTS ON HUMAN DATASET

| Model | Belief Dynamics Update | | | | | | | | |
|---|---|---|---|---|---|---|---|---|---|
| | Change Detection Acc. (%↑) | | | Direction Inference Acc. (%↑) | | | Dir. Acc. (%↑) | | |
| | Health | Surveil. | Zoning | Health | Surveil. | Zoning | Health | Surveil. | Zoning |
| **Human** | **80.00** | **90.00** | **80.00** | **96.67** | **88.33** | **90.00** | **91.67** | **88.83** | **87.00** |
| *OpenAI Models* | | | | | | | | | |
| GPT-4o | $49.36^{\pm1.59}$ | $65.83^{\pm3.12}$ | $57.14^{\pm3.01}$ | $88.89^{\pm0.00}$ | $\mathbf{91.49}^{\pm0.53}$ | $98.33^{\pm3.33}$ | $77.03^{\pm0.48}$ | $\mathbf{83.79}^{\pm1.31}$ | $85.98^{\pm1.71}$ |
| GPT5.1 Rea. High | $54.47^{\pm6.67}$ | $67.50^{\pm5.53}$ | $59.05^{\pm9.80}$ | $83.06^{\pm6.60}$ | $88.26^{\pm3.57}$ | $98.00^{\pm4.00}$ | $74.48^{\pm6.22}$ | $82.03^{\pm3.37}$ | $86.31^{\pm5.11}$ |
| GPT-5-mini | $46.81^{\pm4.66}$ | $65.83^{\pm6.12}$ | $60.95^{\pm3.56}$ | $82.95^{\pm2.56}$ | $90.67^{\pm5.33}$ | $82.05^{\pm9.48}$ | $72.11^{\pm2.84}$ | $83.22^{\pm3.06}$ | $75.72^{\pm7.17}$ |
| o3-mini | $50.64^{\pm4.54}$ | $55.83^{\pm4.25}$ | $59.05^{\pm5.71}$ | $76.43^{\pm4.84}$ | $79.58^{\pm3.64}$ | $78.59^{\pm12.06}$ | $68.69^{\pm3.33}$ | $72.45^{\pm3.10}$ | $72.72^{\pm8.67}$ |
| *Other Closed-Source Models* | | | | | | | | | |
| Claude Sonnet 4.5 | $45.96^{\pm1.04}$ | $62.50^{\pm0.01}$ | $61.90^{\pm0.01}$ | $87.50^{\pm0.01}$ | $84.62^{\pm0.01}$ | $92.31^{\pm0.01}$ | $75.04^{\pm0.31}$ | $77.98^{\pm0.01}$ | $83.19^{\pm0.01}$ |
| Gemini 2.0 Flash | $38.30^{\pm0.00}$ | $\mathbf{70.83}^{\pm0.00}$ | $62.86^{\pm1.90}$ | $\mathbf{100.00}^{\pm0.00}$ | $83.33^{\pm0.00}$ | $\mathbf{100.00}^{\pm0.00}$ | $\mathbf{81.49}^{\pm0.00}$ | $79.58^{\pm0.00}$ | $88.86^{\pm0.57}$ |
| Gemini 2.5 Pro | $49.36^{\pm1.59}$ | $54.17^{\pm5.27}$ | $\mathbf{70.48}^{\pm3.56}$ | $76.19^{\pm5.83}$ | $86.32^{\pm3.85}$ | $100.00^{\pm0.01}$ | $68.14^{\pm3.73}$ | $76.67^{\pm2.43}$ | $91.14^{\pm1.07}$ |
| DeepSeek-R1-0528 | $54.47^{\pm2.17}$ | $64.67^{\pm5.20}$ | $52.38^{\pm3.01}$ | $83.62^{\pm2.10}$ | $84.42^{\pm3.75}$ | $100.00^{\pm0.00}$ | $74.87^{\pm1.86}$ | $78.50^{\pm1.50}$ | $85.71^{\pm0.90}$ |
| Qwen-plus_2025-07-28 | $34.04^{\pm2.33}$ | $63.01^{\pm1.94}$ | $66.67^{\pm0.00}$ | $85.24^{\pm0.95}$ | $83.03^{\pm0.61}$ | $92.31^{\pm0.00}$ | $69.88^{\pm0.97}$ | $77.02^{\pm0.89}$ | $84.62^{\pm0.00}$ |
| Qwen-max_2024-10-15 | $34.89^{\pm1.04}$ | $63.33^{\pm1.67}$ | $66.67^{\pm0.00}$ | $85.71^{\pm0.00}$ | $82.05^{\pm2.56}$ | $92.31^{\pm0.00}$ | $70.47^{\pm0.31}$ | $76.44^{\pm1.29}$ | $84.62^{\pm0.00}$ |
| *Open-Source Models* | | | | | | | | | |
| LLaMA_3.3_70B | $\mathbf{68.09}^{\pm3.01}$ | $54.17^{\pm0.00}$ | $53.33^{\pm1.90}$ | $80.00^{\pm0.00}$ | $85.71^{\pm0.00}$ | $\mathbf{100.00}^{\pm0.00}$ | $76.43^{\pm0.90}$ | $76.25^{\pm0.00}$ | $86.00^{\pm0.57}$ |
| Qwen2.5-32B-instr. | $33.19^{\pm1.04}$ | $62.50^{\pm0.00}$ | $\mathbf{66.67}^{\pm3.01}$ | $84.29^{\pm1.17}$ | $83.33^{\pm0.00}$ | $92.29^{\pm0.38}$ | $68.96^{\pm1.05}$ | $77.08^{\pm0.00}$ | $84.60^{\pm1.17}$ |
| Qwen2.5-7B-instr | $33.62^{\pm1.59}$ | $62.50^{\pm0.00}$ | $65.63^{\pm1.40}$ | $84.29^{\pm1.17}$ | $83.33^{\pm0.00}$ | $93.57^{\pm3.26}$ | $69.09^{\pm1.03}$ | $77.08^{\pm0.00}$ | $85.19^{\pm2.30}$ |
| *Memory-Augmented Baselines* | | | | | | | | | |
| RAG[1] | $42.55^{\pm1.90}$ | $55.83^{\pm5.00}$ | $66.67^{\pm0.00}$ | $66.67^{\pm0.00}$ | $86.51^{\pm3.85}$ | $85.71^{\pm0.00}$ | $59.43^{\pm0.57}$ | $77.30^{\pm1.94}$ | $80.00^{\pm0.00}$ |
| RAG-FC[1] | $45.11^{\pm2.08}$ | $62.43^{\pm6.00}$ | $60.95^{\pm1.90}$ | $80.00^{\pm0.00}$ | $90.91^{\pm0.00}$ | $86.03^{\pm2.82}$ | $69.53^{\pm0.63}$ | $82.36^{\pm1.80}$ | $78.50^{\pm1.40}$ |
| Generative Agents[2] | $36.17^{\pm1.90}$ | $61.06^{\pm2.27}$ | $60.95^{\pm1.90}$ | $81.00^{\pm9.70}$ | $83.64^{\pm3.64}$ | $93.85^{\pm3.08}$ | $67.55^{\pm6.41}$ | $76.86^{\pm2.20}$ | $83.98^{\pm1.58}$ |

Table 11: Directional accuracy results on **Human** dataset for *belief dynamics update* task across three policy topics (Health, Surveillance, Zoning). Values are mean$^{\pm\text{std}}$ over 5 runs. For each topic, we report Change Detection Accuracy, Direction Inference Accuracy, and Directional Accuracy (Dir. Acc). Best results per column are in **bold**.

---

[1]RAG (Lewis et al., 2020), RAG-FC = RAG with Full Context (Lewis et al., 2020).
[2]Generative Agents (Park et al., 2023).

| Metric | Domain Transfer | GPT-4o | | Qwen2.5-32B-instr. | |
|---|---|---|---|---|---|
| | | Cross-Domain | In-Domain | Cross-Domain | In-Domain |
| **Belief State Inference (Accuracy % ↑)** | | | | | |
| | Health → Surv | 58.52±0.77 | 71.11±0.77 | 53.52±1.52 | 77.96±0.41 |
| | Surv → Zone | 65.41±1.58 | 75.25±0.90 | 58.69±1.10 | 76.56±0.45 |
| | Zone → Health | 51.75±1.03 | 77.62±0.35 | 55.08±1.20 | 76.98±0.79 |
| | *Average* | *58.56±0.82* | *74.66±0.24* | *55.76±1.09* | *77.17±0.32* |
| **Belief Dynamics Update (Accuracy % ↑)** | | | | | |
| | Health → Surv | 49.49±0.87 | 61.57±0.19 | 44.41±0.98 | 56.82±0.41 |
| | Surv → Zone | 42.47±0.50 | 63.74±0.58 | 24.84±0.60 | 57.80±0.25 |
| | Zone → Health | 41.96±0.30 | 64.04±0.20 | 30.15±0.35 | 62.25±0.16 |
| | *Average* | *44.64±0.36* | *63.11±0.14* | *33.13±0.48* | *58.96±0.05* |
| **Belief Dynamics Update (MAE ↓)** | | | | | |
| | Health → Surv | 1.55±0.01 | 1.27±0.01 | 1.69±0.01 | 1.43±0.00 |
| | Surv → Zone | 1.60±0.00 | 1.35±0.01 | 1.94±0.01 | 1.43±0.01 |
| | Zone → Health | 1.71±0.00 | 1.26±0.01 | 2.05±0.01 | 1.33±0.01 |
| | *Average* | *1.62±0.00* | *1.29±0.00* | *1.89±0.01* | *1.40±0.00* |
| **ATI (% ↑)** | | | | | |
| | Health → Surv | 23.96±3.83 | 59.43±1.36 | 10.04±2.71 | 60.56±1.19 |
| | Surv → Zone | -19.05±2.32 | 66.01±1.49 | 12.54±1.80 | 60.25±0.65 |
| | Zone → Health | 24.72±1.67 | 78.92±1.48 | 21.78±2.00 | 75.41±0.95 |
| | *Average* | *9.64±1.66* | *67.29±0.74* | *14.36±1.49* | *64.71±0.23* |

Table 12: Cross-domain swap test: models are trained with QA context from one domain and evaluated on another domain for the *same participant*. We report performance for *Healthcare →  Surveillance*, *Surveillance → Zoning*, *Zoning → Healthcare*, and their average. Reported as *mean ± std* over 5 runs.

| Metric | Domain | GPT-4o | | | Gemini 2.0 Flash | | | Qwen2.5-32B-instr. | | |
|---|---|---|---|---|---|---|---|---|---|---|
| | | 5 QAs | 10 QAs | 20+ QAs | 5 QAs | 10 QAs | 20+ QAs | 5 QAs | 10 QAs | 20+ QAs |
| **Belief State Inference (Accuracy % ↑)** | | | | | | | | | | |
| | Health | 68.89±0.51 | 67.59±1.73 | 71.11±0.77 | 61.30±0.41 | 60.19±0.65 | 63.89±0.00 | 68.33±1.21 | 75.37±4.47 | **77.96±0.41** |
| | Surv. | 73.93±1.07 | 72.30±0.37 | 75.25±0.90 | 65.25±0.45 | 64.75±0.00 | 68.20±0.37 | 73.11±2.68 | 75.90±1.80 | **76.56±0.45** |
| | Zone | 72.22±0.00 | 72.06±0.66 | 77.62±0.35 | 65.08±0.00 | 71.27±0.35 | **77.78±0.00** | 65.08±0.56 | 67.78±1.06 | 76.98±0.79 |
| | Avg | 71.68±0.47 | 70.65±0.49 | 74.66±0.24 | 63.87±0.13 | 65.40±0.25 | 69.95±0.12 | 68.84±0.73 | 73.02±1.66 | **77.17±0.32** |
| **Belief Dynamics Update (Accuracy % ↑)** | | | | | | | | | | |
| | Health | 61.48±0.35 | **62.29±0.33** | 61.57±0.19 | 59.70±0.09 | 58.22±0.28 | 56.23±0.12 | 59.45±0.55 | 58.52±0.74 | 56.82±0.41 |
| | Surv. | **67.80±0.45** | 64.56±0.39 | 63.74±0.58 | 65.71±0.30 | 62.58±0.23 | 62.25±0.37 | 62.53±0.57 | 60.66±0.60 | 57.80±0.25 |
| | Zone | 63.27±0.36 | **65.82±0.39** | 64.04±0.20 | 62.25±0.21 | 65.20±0.16 | 63.16±0.10 | 63.38±0.69 | 63.02±0.47 | 62.25±0.16 |
| | Avg | 64.19±0.22 | **64.22±0.20** | 63.11±0.14 | 62.56±0.18 | 62.00±0.13 | 60.55±0.08 | 61.79±0.46 | 60.73±0.46 | 58.96±0.05 |
| **Belief Dynamics Update (MAE ↓)** | | | | | | | | | | |
| | Health | **1.26±0.01** | 1.28±0.00 | 1.27±0.01 | 1.36±0.00 | 1.37±0.00 | 1.39±0.00 | 1.36±0.00 | 1.41±0.01 | 1.43±0.00 |
| | Surv. | **1.19±0.01** | 1.26±0.01 | 1.35±0.01 | 1.23±0.00 | 1.30±0.01 | 1.33±0.00 | 1.31±0.01 | 1.36±0.01 | 1.43±0.01 |
| | Zone | 1.24±0.00 | **1.21±0.01** | 1.26±0.01 | 1.34±0.00 | 1.26±0.00 | 1.33±0.00 | 1.27±0.01 | 1.27±0.01 | 1.33±0.01 |
| | Avg | **1.23±0.00** | 1.25±0.00 | 1.29±0.00 | 1.31±0.00 | 1.31±0.00 | 1.35±0.00 | 1.31±0.00 | 1.35±0.00 | 1.40±0.00 |
| **ATI (% ↑)** | | | | | | | | | | |
| | Health | 55.33±1.05 | 55.54±2.52 | 59.43±1.36 | 31.13±0.84 | 34.19±0.78 | 50.03±0.06 | 47.57±1.16 | 58.18±5.94 | **60.56±1.19** |
| | Surv. | 69.38±1.54 | 60.07±0.27 | 66.01±1.49 | 50.19±0.21 | 46.56±1.71 | 52.20±0.51 | 57.95±5.12 | 59.36±2.68 | 60.25±0.65 |
| | Zone | 70.60±0.49 | 67.45±1.02 | 78.92±1.48 | 62.63±0.11 | 67.57±0.57 | **80.62±0.57** | 62.15±0.94 | 65.47±2.27 | 75.41±0.95 |
| | Avg | 64.46±0.81 | 60.46±0.98 | **67.29±0.74** | 46.87±0.35 | 48.24±0.76 | 59.76±0.18 | 55.28±1.46 | 60.58±2.58 | 64.71±0.23 |

Table 13: Question masking / length scaling for both Belief State Inference and Belief Dynamics Update. Reported as *mean ± std* over 5 runs. Best results per row are highlighted in **bold**.

| Metric | Domain | GPT-4o | | Qwen2.5-32B-instr. | |
|---|---|---|---|---|---|
| | | No-Context | Full-Context | No-Context | Full-Context |
| **Belief State Inference (Accuracy % ↑)** | | | | | |
| | Health | 55.93±1.40 | 71.11±0.77 | 45.19±1.66 | 77.96±0.41 |
| | Surv. | 66.07±1.37 | 75.25±0.90 | 59.02±1.16 | 76.56±0.45 |
| | Zone | 53.49±1.99 | 77.62±0.35 | 48.57±1.30 | 76.98±0.79 |
| | Avg | 58.49±0.43 | 74.66±0.24 | 50.92±0.55 | 77.17±0.32 |
| **Belief Dynamics Update (Accuracy % ↑)** | | | | | |
| | Health | 34.62±0.68 | 61.57±0.19 | 34.58±0.55 | 56.82±0.41 |
| | Surv. | 48.24±0.31 | 63.74±0.58 | 33.90±0.60 | 57.80±0.25 |
| | Zone | 36.62±0.52 | 64.04±0.20 | 27.89±0.38 | 62.25±0.16 |
| | Avg | 39.83±0.45 | 63.11±0.14 | 32.12±0.24 | 58.96±0.05 |
| **Belief Dynamics Update (MAE ↓)** | | | | | |
| | Health | 1.77±0.01 | 1.27±0.01 | 1.89±0.01 | 1.43±0.00 |
| | Surv. | 1.51±0.01 | 1.35±0.01 | 1.75±0.01 | 1.43±0.01 |
| | Zone | 1.84±0.01 | 1.26±0.01 | 1.99±0.01 | 1.33±0.01 |
| | Avg | 1.70±0.01 | 1.29±0.00 | 1.88±0.01 | 1.40±0.00 |
| **ATI (% ↑)** | | | | | |
| | Health | 19.76±2.31 | 59.43±1.36 | -7.58±2.70 | 60.56±1.19 |
| | Surv. | 35.66±2.50 | 66.01±1.49 | 15.31±1.64 | 60.25±0.65 |
| | Zone | 26.20±3.73 | 78.92±1.48 | 14.92±2.15 | 75.41±0.95 |
| | Avg | 26.98±1.01 | 67.29±0.74 | 6.88±0.81 | 64.71±0.23 |

Table 14: Comparison of *No-Context* (population prior) and *Full-Context* (with individual transcripts) settings. Reported as *mean ± std* over 5 runs.

| Metric | Domain | Gemini 2.0 Flash | Qwen2.5-32B-instr. | GPT-4o |
|---|---|---|---|---|
| **Belief State Inference (Accuracy % ↑)** | | | | |
| | Health | 52.78±0.65 | 58.15±0.77 | 50.93±7.29 |
| | Surv. | 52.79±1.49 | 64.75±0.00 | 61.31±0.37 |
| | Zone | 47.46±0.66 | 58.41±0.90 | 51.11±0.71 |
| | Avg | 51.01±0.52 | 60.44±0.47 | 54.45±2.30 |
| **Belief Dynamics Update (Accuracy % ↑)** | | | | |
| | Health | 34.19±0.12 | 33.77±0.73 | 37.37±0.19 |
| | Surv. | 40.71±0.36 | 40.77±0.63 | 43.13±0.75 |
| | Zone | 39.35±0.16 | 39.93±0.46 | 37.38±0.28 |
| | Avg | 38.08±0.13 | 38.16±0.44 | 39.30±0.22 |
| **Belief Dynamics Update (MAE ↓)** | | | | |
| | Health | 2.13±0.01 | 2.19±0.01 | 2.02±0.01 |
| | Surv. | 1.85±0.01 | 1.87±0.01 | 1.80±0.01 |
| | Zone | 2.05±0.00 | 2.01±0.00 | 1.98±0.01 |
| | Avg | 2.01±0.00 | 2.02±0.01 | 1.93±0.01 |

Table 15: Human *Cross_Person* test: models trained on one participant and evaluated on another. Reported as mean ± std over 5 runs.

RESULTS ON SYNTHETIC DATASET

| Metric | Domain | GPT-4o | | | Gemini 2.0 Flash | | | Qwen2.5-32B-instr. | | |
|---|---|---|---|---|---|---|---|---|---|---|
| | | 5 QAs | 10 QAs | 20+ QAs | 5 QAs | 10 QAs | 20+ QAs | 5 QAs | 10 QAs | 20+ QAs |
| **Belief State Inference (Accuracy % ↑)** | | | | | | | | | | |
| | Health | 56.96±1.27 | 62.28±1.06 | 60.76±0.00 | 37.97±0.00 | 43.04±0.00 | 51.90±0.00 | 47.34±2.12 | 59.24±1.39 | **64.30±0.57** |
| | Surv. | **73.48±0.60** | 74.13±0.49 | 75.00±0.77 | 63.48±0.60 | 61.74±0.49 | 63.48±1.24 | 67.61±1.19 | 68.70±0.91 | 68.70±0.49 |
| | Zone | 62.30±1.00 | 66.89±0.45 | 68.03±0.58 | 57.38±0.00 | 62.13±0.90 | 63.61±0.45 | 66.56±0.37 | 70.82±0.73 | **71.64±0.93** |
| | Avg | 64.25±0.57 | 67.76±0.51 | 67.93±0.32 | 52.94±0.20 | 55.64±0.33 | 59.66±0.48 | 60.50±0.58 | 66.25±0.77 | **68.21±0.30** |
| **Belief Dynamics Update (Accuracy % ↑)** | | | | | | | | | | |
| | Health | **53.23±0.71** | 52.18±0.44 | 52.71±0.29 | 42.75±0.18 | 41.79±0.25 | 45.28±0.20 | 45.37±0.57 | 44.76±0.41 | 45.24±0.39 |
| | Surv. | 64.20±0.00 | **65.12±0.44** | 65.00±0.35 | 63.64±0.14 | 63.02±0.14 | 62.96±0.00 | 63.70±0.71 | 64.75±0.14 | 65.00±0.17 |
| | Zone | 56.28±0.27 | 55.53±1.78 | **58.11±0.48** | 54.09±0.18 | 55.73±0.14 | 55.93±0.22 | 53.00±0.54 | 54.74±0.62 | 55.88±0.54 |
| | Avg | 57.90±0.27 | 57.61±0.51 | **58.61±0.28** | 53.49±0.13 | 53.51±0.09 | 54.73±0.13 | 54.02±0.44 | 54.75±0.16 | 55.37±0.24 |
| **Belief Dynamics Update (MAE ↓)** | | | | | | | | | | |
| | Health | **1.68±0.00** | 1.73±0.01 | 1.72±0.00 | 1.82±0.00 | 1.83±0.00 | 1.77±0.00 | 1.84±0.01 | 1.83±0.01 | 1.80±0.00 |
| | Surv. | **1.18±0.01** | **1.17±0.01** | 1.18±0.01 | 1.18±0.00 | 1.19±0.00 | 1.18±0.00 | 1.18±0.01 | 1.18±0.01 | 1.18±0.00 |
| | Zone | 1.43±0.01 | 1.44±0.00 | **1.41±0.01** | 1.54±0.00 | 1.52±0.00 | 1.49±0.00 | 1.56±0.01 | 1.52±0.01 | 1.50±0.01 |
| | Avg | **1.43±0.01** | 1.45±0.00 | 1.44±0.00 | 1.51±0.00 | 1.51±0.00 | 1.48±0.00 | 1.53±0.00 | 1.51±0.01 | 1.49±0.01 |

Table 16: Synthetic *belief state inference* and *belief dynamics update* performance across question lengths (5Q, 10Q, 20Q) for *GPT-4o*, *Gemini 2.0 Flash* and *Qwen2.5-32B-instr.*. Reported as mean ± std over 5 runs. Best in each **row (across all models and contexts)** is bolded.

| Task | Model | Health → Surv | Surv → Zone | Zone → Health | Avg. |
|---|---|---|---|---|---|
| Belief State Inference (Accuracy (% ↑)) | GPT-4o | $18.99^{\pm 0.90}$ | $50.22^{\pm 0.49}$ | $39.67^{\pm 0.73}$ | $36.29^{\pm 0.51}$ |
| | Gemini 2.0 Flash | $17.97^{\pm 2.26}$ | $38.91^{\pm 0.91}$ | $20.49^{\pm 1.30}$ | $25.79^{\pm 1.03}$ |
| | Qwen2.5-32B-instr. | $22.78^{\pm 1.55}$ | $48.26^{\pm 1.97}$ | $45.57^{\pm 0.73}$ | $38.87^{\pm 1.06}$ |
| Belief Dynamics Update (Accuracy (% ↑)) | GPT-4o | $43.62^{\pm 0.71}$ | $55.43^{\pm 0.64}$ | $35.58^{\pm 0.28}$ | $44.88^{\pm 0.37}$ |
| | Gemini 2.0 Flash | $36.03^{\pm 0.22}$ | $41.73^{\pm 0.51}$ | $52.06^{\pm 0.32}$ | $43.27^{\pm 0.23}$ |
| | Qwen2.5-32B-instr. | $34.28^{\pm 0.56}$ | $23.40^{\pm 0.34}$ | $26.65^{\pm 0.89}$ | $28.11^{\pm 0.41}$ |
| Belief Dynamics Update (MAE ↓) | GPT-4o | $1.64^{\pm 0.01}$ | $1.45^{\pm 0.01}$ | $1.73^{\pm 0.00}$ | $1.61^{\pm 0.00}$ |
| | Gemini 2.0 Flash | $1.97^{\pm 0.00}$ | $1.70^{\pm 0.01}$ | $1.62^{\pm 0.01}$ | $1.76^{\pm 0.00}$ |
| | Qwen2.5-32B-instr. | $2.16^{\pm 0.01}$ | $1.88^{\pm 0.01}$ | $2.12^{\pm 0.01}$ | $2.05^{\pm 0.00}$ |

Table 17: Synthetic *Cross_Domain* test: models are trained in one topic domain and evaluated on another. Reported as mean ± std over 5 runs.

| Metric | Domain | GPT-4o | Gemini 2.0 Flash | Qwen2.5-32B-instr. |
|--------|--------|--------|------------------|---------------------|
| **Belief State Inference (Accuracy % ↑)** | | | | |
| | Health | 41.52±1.06 | 30.89±0.69 | 42.28±1.13 |
| | Surv. | 54.35±0.00 | 47.83±0.00 | 55.00±0.60 |
| | Zone | 33.11±0.73 | 33.93±0.45 | 34.75±0.45 |
| | Avg | 42.99±0.46 | 37.55±0.30 | 44.01±0.46 |
| **Belief Dynamics Update (Accuracy % ↑)** | | | | |
| | Health | 40.17±0.22 | 30.13±0.15 | 32.10±0.53 |
| | Surv. | 52.90±0.35 | 50.93±0.00 | 52.78±0.22 |
| | Zone | 36.72±0.50 | 35.53±0.11 | 35.04±0.32 |
| | Avg | 43.27±0.19 | 38.86±0.06 | 39.97±0.04 |
| **Belief Dynamics Update (MAE ↓)** | | | | |
| | Health | 2.04±0.00 | 2.16±0.00 | 2.14±0.01 |
| | Surv. | 1.54±0.00 | 1.52±0.00 | 1.49±0.00 |
| | Zone | 2.00±0.00 | 2.06±0.00 | 2.06±0.00 |
| | Avg | 1.86±0.00 | 1.91±0.00 | 1.90±0.00 |

Table 18: Synthetic *Cross_Person* test: models trained on one participant and evaluated on another. Reported as mean ± std over 5 runs.

# D    CHATBOT DESIGN

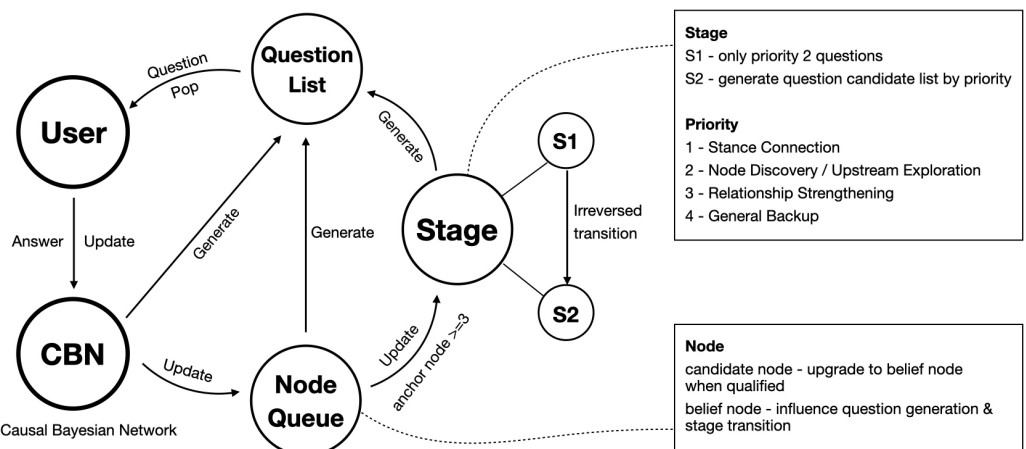

Figure 9: Overview of the QA loop and data structures. The system integrates the Causal Belief Network (CBN), Node Queue, and Question List to guide interaction. Stages regulate question priorities, with an irreversible transition from Stage 1 to Stage 2 once anchor nodes $\geq 3$.

CORE DATA STRUCTURES

The system is built on three key data structures: (i) the **Causal Belief Network (CBN)**, (ii) the **Node Queue**, and (iii) the **Question List**. Together with a staged QA loop, these structures support the dynamic modeling of user beliefs and the generation of targeted questions.

CAUSAL BELIEF NETWORK (CBN)

The CBN is the central representation of the user's belief system. It organizes concepts as nodes and captures their relations as edges.

- **Nodes.** Nodes represent concepts in the belief system.
  - *Candidate Nodes*: new concepts detected in user answers, under evaluation.
  - *Belief Nodes*: stable concepts that have been upgraded from candidates (e.g., due to repeated mentions or high user confidence).
  - *Anchor Nodes*: a subset of belief nodes that play a special role in question generation and stage transition.
- **Edges.** Edges encode causal or influence relations between nodes. They specify direction (source $\rightarrow$ target), polarity (positive/negative), and optionally strength.

NODE QUEUE

The Node Queue maintains candidate nodes that may become belief nodes.

- **Entry Condition:** a new concept first appears in user responses.
- **Upgrade Condition:** node is promoted to belief node when thresholds are met (e.g., frequency of mention, confidence expressed by the user).

QUESTION LIST

The Question List stores both guiding questions and follow-up questions. It is dynamically updated based on the current CBN and node queue, and it serves as the buffer for delivering the next question to the user.

QA LOOP OVERVIEW

The overall interaction loop proceeds as follows:

1. **Initialization.** The CBN contains only a stance node.

2. **User Input.** User provides a new answer.

3. **Update.** Update the CBN and node queue based on the answer.

4. **Question Generation.** Generate new questions using the CBN and queues; append them to the question list.

5. **Next Question.** Select and ask the next question from the list.

TWO-STAGE DESIGN

The QA loop has two stages. The transition occurs when the number of anchor nodes $\geq 3$. This transition is one-way; Stage 2 never returns to Stage 1.

- **Stage 1.** Only Priority 2 questions are allowed. Rationale: with too few anchors, meaningful relationship questions are not possible. The system must first accumulate important concepts to avoid premature exploration.

- **Stage 2.** Questions are selected from a candidate list according to priority. Higher-priority items are chosen first.

QUESTION PRIORITIES

- **Priority 1 – Stance Connection.**
  Purpose: connect essential concepts to the user's stance.
  Condition: isolated anchor (out-degree = 0, not connected to stance).
  *Format: "How does {anchor} affect your support for {stance}? Positive or negative? How strong?"*
  *Example:* "How does privacy protection affect your support for surveillance?"

- **Priority 2 – Node Discovery / Upstream Exploration.**
  Purpose: discover new concepts or explore influencing factors of anchors.
  Condition: in Stage 1 or anchor has fewest in-degrees.
  *Format (Stage 1): "Tell me more about {concept}."*
  *Format (Stage 2): "What factors influence {anchor}? Positive or negative?"*
  *Examples:* "Tell me more about public safety."; "What factors influence government oversight?"

- **Priority 3 – Relationship Strengthening.**
  Purpose: quantify the strength and direction of existing relationships.
  Condition: edge requires parameters or graph pattern needs completion.
  *Format: "How strong is the relationship between {A} and {B}? Positive or negative?"*
  *Example:* "How does technological advancement affect privacy protection? Strong or weak?"

- **Priority 4 – General Backup.**
  Purpose: fill in missing information at the end of the interview.
  Condition: remaining questions $\leq 3$ and candidate pool insufficient.
  *Format: "Anything else important we have not discussed?"*
  *Example:* "Any clarifications on your previous answers?"

# E  SEMI-STRUCTURED INTERVIEW AND CAUSAL BELIEF NETWORK FORMALIZATION

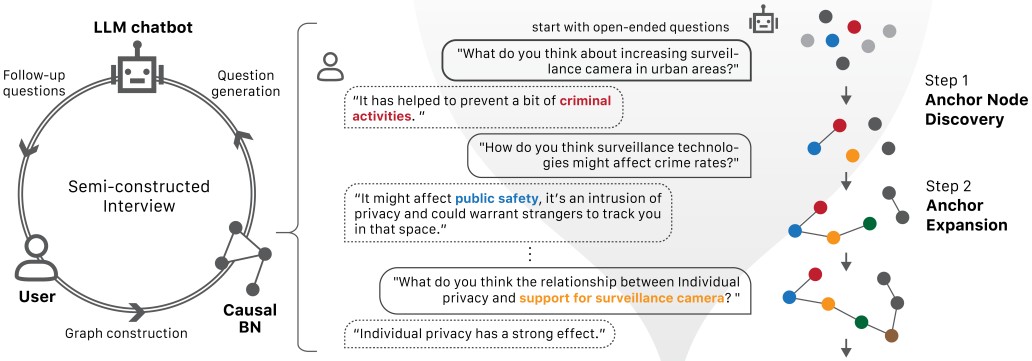

Figure 10: Illustration of the semi-structured interview process and causal belief network construction. The chatbot begins with open-ended questions and extracts candidate concepts from user responses (Anchor Node Discovery). Once three or more anchors are identified, it transitions to targeted follow-ups to expand causal relations (Anchor Expansion). Edges represent directional influences with polarity, forming the evolving CBN.

**Semi-Constructed Interview Design.**  We use GPT-4 (or Qwen for open-source deployments) as the backbone of a semi-structured interviewer. The model follows a two-phase logic:

1. **Anchor Node Discovery:** From initial open-ended responses, the system uses noun-phrase mining and causal phrase detection to extract candidate belief variables. Candidates that appear in multiple QA pairs or show causal centrality are promoted to *anchor nodes*, representing key ideas around which reasoning is structured.

2. **Anchor Expansion:** For each anchor node, the system asks targeted follow-ups (e.g., "What causes this?" or "What does this influence?"). These responses are parsed into edges, which represent directional causal relations with confidence scores and modifiers (positive or negative influence).

**causal BN Formalization.**  Each participant's graph is a Directed Acyclic Graph (DAG), with nodes $v_i$ labeled by semantically grounded belief variables, and edges $e_{ij}$ denoting belief in the causal influence from $v_i \rightarrow v_j$. We capture the following metadata for each element:

- **Node-level:**  Label, frequency across QAs, semantic role (`external_state`, `internal_affect`, `behavioral_intention`), layer depth (e.g., experience $\rightarrow$ value $\rightarrow$ stance).

- **Edge-level:** Confidence (based on question phrasing), polarity (positive or negative), and QA provenance.

**Edge Probability Estimation.**  Each edge is assigned a probability $P(v_j|v_i)$ based on linguistic indicators in the answer and motif alignment scores:

$$P(v_j|v_i) = \sigma(w_1 \cdot s_{\text{causal}} + w_2 \cdot s_{\text{linguistic}} + w_3 \cdot s_{\text{motif}}) \tag{1}$$

where $s_{\text{causal}}$ captures explicit causal phrasing, $s_{\text{linguistic}}$ measures structural confidence from the model, and $s_{\text{motif}}$ reflects alignment to previously seen cognitive motif patterns. $\sigma$ is the logistic function.

**Demographic Consideration.**  To support downstream generalization and population modeling (Phase III), each interview is paired with structured demographic data (age, housing status, transportation mode, etc.). These attributes allow later stages to interpolate motif distributions and simulate representative reasoning across diverse population groups.

**Stopping Criteria.** The system continues alternating between node discovery and causal expansion until one or more termination conditions are met: (1) no new anchor nodes emerge, (2) motif-based reasoning paths reach convergence, or (3) information gain across simulated stances falls below a threshold.

**Forward Simulation and Inference.** Once an intervention is identified, the causal BN is used to simulate the effects of this intervention. The intervention is applied to the graph as a DO-operation which cuts all incoming edges to the intervened node and updates its distribution. This is followed by a forward simulation to propagate the effects through the network.

Post-processing includes analyzing changes in node probabilities and identifying significant shifts, particularly those related to policy objectives. These results help explain the agent's behavior and evaluate proposed interventions. This structured method empowers stakeholders to make data-driven decisions based on causal dynamics.

# F QUESTIONNAIRE GENERAL DESIGN

The questionnaire serves as the foundational layer of HugAgent, designed to capture both baseline beliefs and structured reasoning factors before participants engage in interactive chatbot interviews. The survey was administered through the Prolific platform, ensuring a diverse and demographically balanced pool of respondents. Importantly, not all participants were asked to complete the chatbot phase; instead, all participants began with the questionnaire, and only a subset was later recruited for semi-structured chatbot interviews. This two-stage design allows us to ground conversational transcripts in an already standardized and validated set of structured responses.

The questionnaire is structured into three complementary components. First, participants provide **demographic information**, including age, gender, education, income, housing status, neighborhood context, and transportation habits. These variables are aligned with U.S. Census and urban planning survey standards, enabling stratified analyses of systematic variability in beliefs across groups (e.g., renters versus homeowners, high-income versus low-income). Second, participants answer **stance and intervention items**, rating their support on a 1–10 scale and updating their stance under hypothetical scenarios (e.g., reduced rent under upzoning, reduced household costs under universal healthcare, reduced crime under surveillance). Third, each topic includes a standardized **reason pool**, a set of common factors such as affordability, fairness, privacy, safety, and neighborhood character. After reporting their stance, participants rate on a 1–5 scale how strongly each reason influences their opinion. This structure provides interpretable ground-truth (GT) data for reasoning dimensions and enables cross-participant comparability, since all individuals evaluate the same set of reasons. By aggregating these structured ratings, we can test whether models not only predict overall support levels but also recover the latent weighting of reasons that drive human decision-making. These structured ratings also serve as a reference for aligning open-ended chatbot responses with quantitative belief factors, creating a consistent bridge between free-text explanations and structured data.

## QUESTION TYPES

We define distinct question types to systematically probe both interpretive reasoning (inferring hidden beliefs) and predictive reasoning (anticipating belief change).

**Type 1.1: Stance elicitation (baseline beliefs).** Participants report initial support levels on a 1–10 scale (e.g., "How much do you support allowing taller apartment buildings in your neighborhood?"). This provides the starting point for belief state modeling.

**Type 1.2: Reason evaluation.** Participants rate how strongly predefined reasons (e.g., economic benefits, fairness, neighborhood character, privacy, efficiency) influence their stance on a 1–5 scale. The reason pools are shared across all respondents within a topic, allowing structured comparison across individuals and providing ground-truth data on how value dimensions shape beliefs.

**Type 1.3: Contextualized interview beliefs.** Through chatbot dialogue, participants explain or justify their stance in natural language. These free-form responses provide latent belief evidence, which models must interpret to infer hidden attitudes. The transcripts can be cross-validated against the structured reason evaluations for consistency.

**Type 2.1: Scenario-based interventions.** Participants evaluate counterfactual scenarios (e.g., "If rent prices fall by 15% after upzoning, how would your stance change?"). This probes dynamic updating of beliefs in response to outcomes.

**Type 2.2: Normative fairness interventions.** Scenarios manipulate fairness dimensions (e.g., "If upzoning applied equally to wealthy neighborhoods" or "If cameras were controlled by local boards"). These tasks test whether models capture fairness-based belief shifts.

**Type 2.3: Conditional trade-offs.** Participants consider hybrid conditions (e.g., "Universal healthcare exists alongside private insurance" or "Surveillance footage stored for 48 hours only"). These tasks require reasoning under institutional or design constraints.

| Task Type | Upzoning | Surveillance Cameras | Universal Healthcare |
|-----------|----------|----------------------|----------------------|
| **Belief Inference** | Q: "On a scale from 1 to 10, how much do you support allowing taller apartment buildings in your neighborhood?" A: "Probably around 3. I worry it changes the character of the area." **Target:** Low support (3/10); belief: upzoning harms neighborhood character. | Q: "How comfortable do you feel being monitored by public cameras?" A: "Honestly, it makes me uneasy. I don't trust how the footage is used." **Target:** Low comfort; belief: privacy concerns about surveillance. | Q: "Do you feel your current health insurance provides adequate coverage?" A: "Not really, I often avoid going to specialists due to cost." **Target:** Insurance inadequate; belief: high costs limit access. |
| **Reaction Prediction** | Scenario: "After the city allows more apartments, rent prices drop 15%. Your monthly rent is noticeably lower." **Target:** Support increases (e.g., +2 on 1–10 scale). | Scenario: "After installing cameras, neighborhood break-ins fall and robberies drop by 20%." **Target:** Support increases (stronger acceptance). | Scenario: "After switching to universal healthcare, household out-of-pocket costs fall by $3,000 annually." **Target:** Support increases (e.g., from 6/10 to 9/10). |

Table 19: Illustrative examples of HugAgent questionnaire and interview tasks. Each domain includes both belief inference and reaction prediction items, enabling evaluation of models on stance attribution and dynamic belief updating.

# G  SCALAR SENSITIVITY ANALYSIS

Scalar responses are sometimes viewed as potentially sensitive to sampling noise in LLMs. To evaluate the stability of LLM-generated scalar outputs, we analyze the variance of model responses on both 1–5 and 1–10 scales across three domains (healthcare, surveillance, and zoning). For clarity, we visualize one representative model from each category (OpenAI, other closed-source, and open-source). All remaining models demonstrate similar stability patterns; full results are available upon request.

## G.1  CONSISTENCY OF MODEL PREDICTIONS

Across all scalar questions, model predictions remain highly consistent across five independent runs. With temperature fixed at 0.1 and identical prompting conditions, the variance across runs is minimal, indicating that LLMs produce stable scalar outputs. Figures below report the standard deviation across runs for each model.

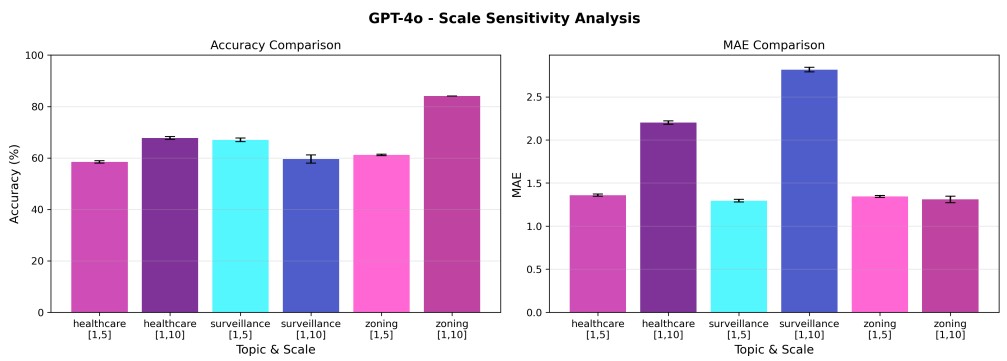

Figure 11: Scalar sensitivity analysis for GPT-4o.

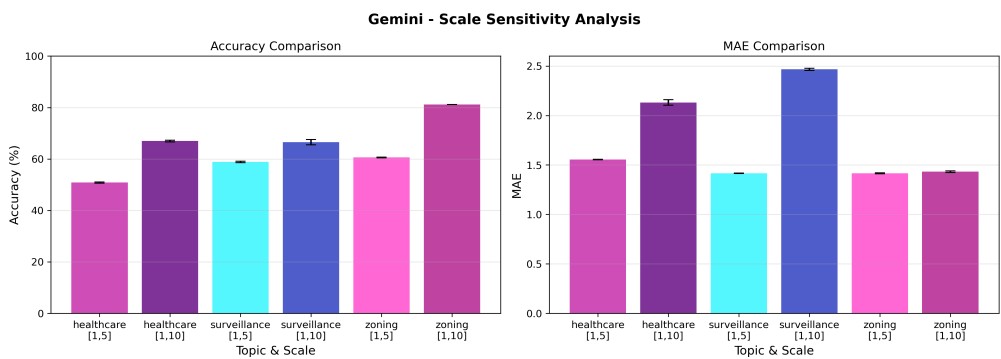

Figure 12: Scalar sensitivity analysis for Gemini 2.0 Flash.

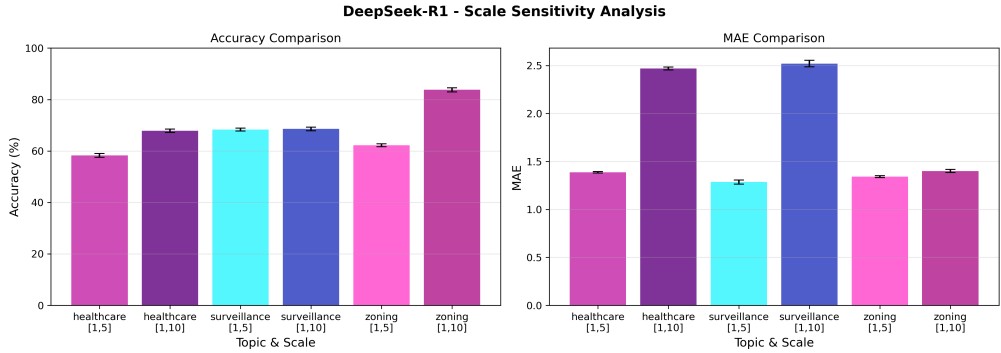

Figure 13: Scalar sensitivity analysis for DeepSeek-R1.

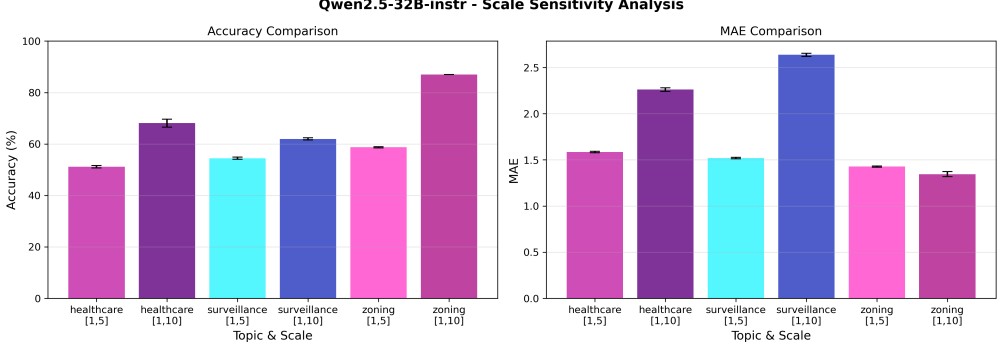

Figure 14: Scalar sensitivity analysis for Qwen2.5-32B-Instr.

## G.2 ALIGNMENT WITH HUMAN RESPONSE DISTRIBUTIONS

Beyond run-to-run stability, we also evaluate whether model-generated scalar scores align with empirical human response distributions. Across all three domains and both scalar ranges, the models achieve Jensen–Shannon Divergence (JSD) values below $0.10$ and Pearson correlation coefficients of $r \geq 0.2$. JSD quantifies the similarity between two probability distributions, where values below $0.10$ are commonly interpreted as indicating close alignment in distributional shape. Pearson's $r$ measures the correlation between the model and human mean scores, capturing alignment in central tendencies. Together, these complementary metrics provide evidence that models capture both the overall distributional patterns and the relative ordering of human scalar judgments. Across most settings, we observe low JSD ($0.02$–$0.08$) and moderate positive correlations ($r \approx 0.2$–$0.4$), consistent with prior findings on LLM stability. Although certain tasks (e.g., zoning on the 1–10 scale) exhibit slightly higher variability, the broader distributional trends remain similar to human responses. Taken together, these results demonstrate that model-generated scalar outputs are stable, well-aligned with human response patterns, and that any numerical noise at the item level does not affect the main conclusions of the study.

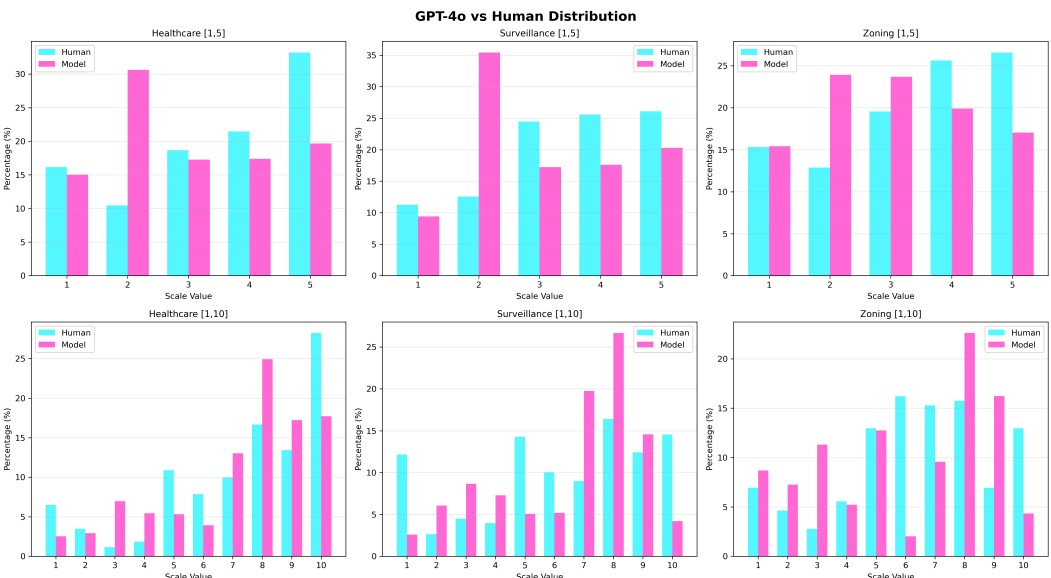

Figure 15: Scalar response distributions for GPT-4o across three topics and two scales.

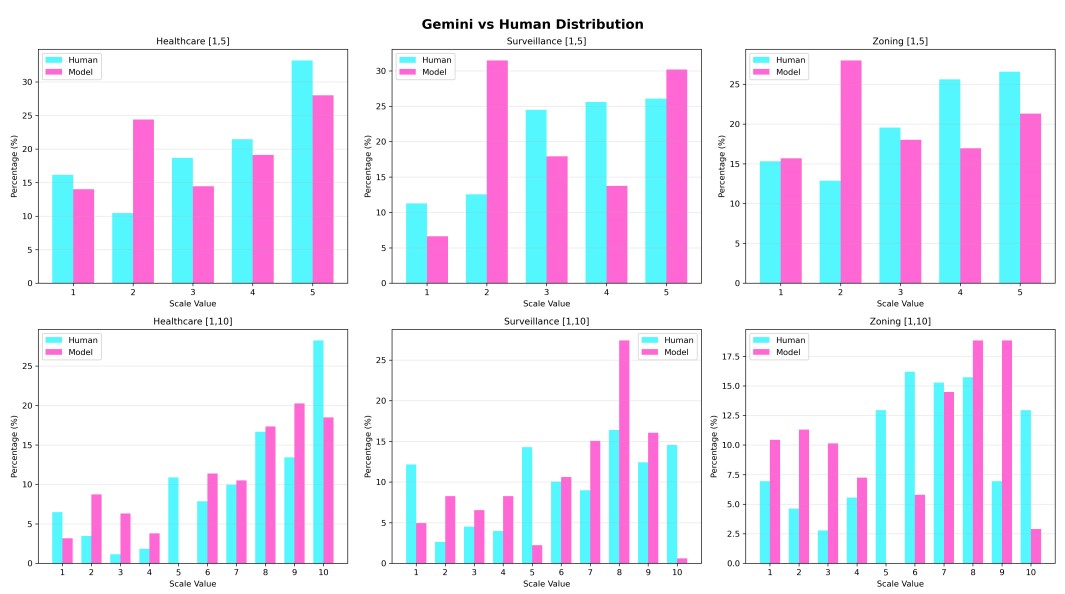

Figure 16: Scalar response distributions for Gemini-2.0-Flash across three topics and two scales.

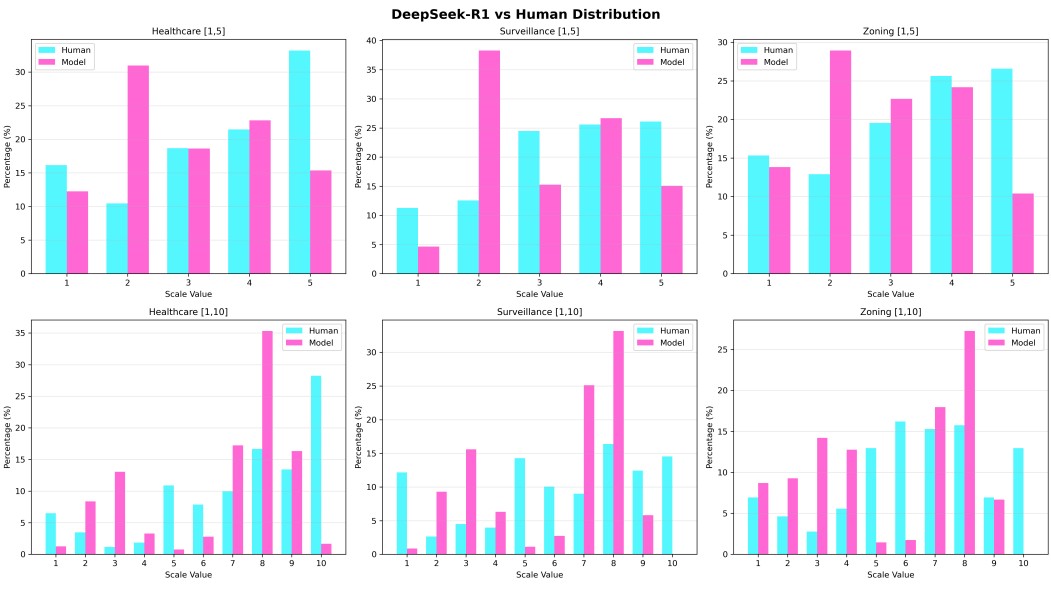

Figure 17: Scalar response distributions for DeepSeek-R1 across three topics and two scales.

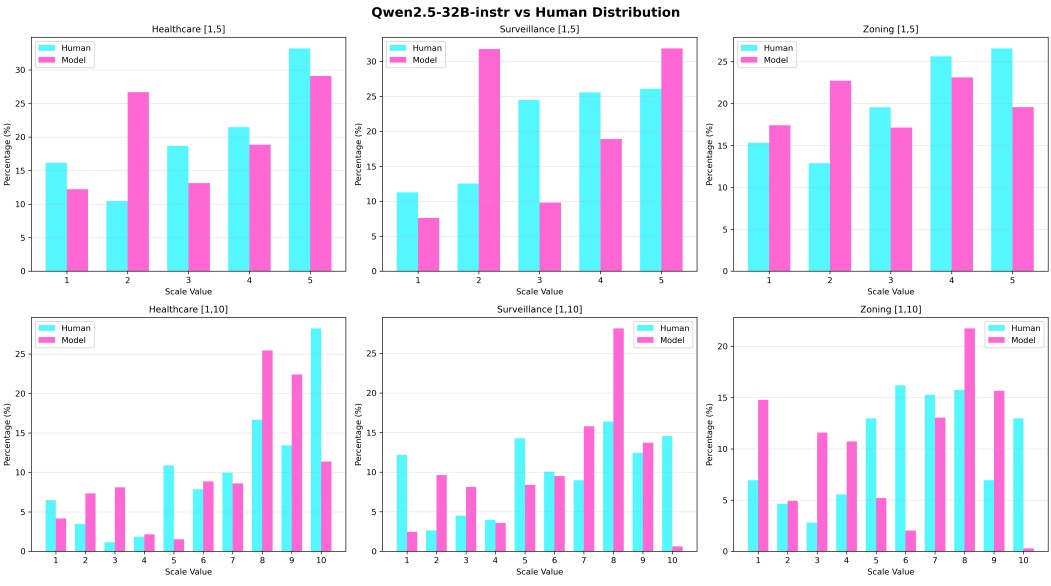

Figure 18: Scalar response distributions for Qwen2.5-32B-Instr. across three topics and two scales.

## H    ZONING OPINION QUESTIONNAIRE (HUMAN EVALUATION)

To rigorously evaluate the fidelity of our generative agents' responses against real human participants, we conducted a structured public opinion survey titled *General Housing & Upzoning Public Opinion Survey*. The survey was carefully designed to facilitate comparison between human-generated responses and those from LLM-based agents, specifically targeting residents of United states.

### MOTIVATION AND OBJECTIVES

This survey aimed to assess public opinion on urban upzoning scenarios, capturing nuanced attitudes toward housing policies and their underlying reasoning. Our goal was to determine whether generative agents could reliably replicate human response patterns, especially regarding sensitive issues such as neighborhood change, density increases, and emotional responses like YIMBY (Yes In My Backyard) and NIMBY (Not In My Backyard).

### SURVEY STRUCTURE AND METHODOLOGY

The survey comprised two primary sections:

**Section 1: Demographic and Background Information**    Participants provided detailed demographic data aligned with U.S. Census Bureau categories:

- Age
- Housing status (owner or renter)
- Income levels
- Occupation
- Marital status
- Presence of children
- Transportation mode
- Monthly rent as a percentage of income
- Residential mobility
- ZIP code or proximity-based location verification

To ensure data quality, participants were required to explicitly answer an attention check question.

**Section 2: Scenario-Based Opinion Measurement**    Participants were first asked general zoning questions and rated their support for allowing larger, taller apartment buildings in their neighborhood on a 1–10 Likert scale (1 = strongly oppose, 10 = strongly support). Each scenario was accompanied by a set of related factors, which participants evaluated on a 1–5 scale (1 = no impact, 5 = very large impact), regardless of whether the impact was positive or negative. The factors included:

- Housing supply and availability
- Affordability for low- and middle-income residents
- Neighborhood character and visual compatibility
- Traffic and parking availability
- Walkability and access to amenities
- Noise, congestion, or infrastructure strain
- Fairness and distribution of development
- Economic vitality for local businesses
- Building height/scale relative to surroundings
- Property values or homeownership concerns

Clarifying examples were provided to ensure consistent interpretation of impact ratings.

### DATA COLLECTION AND IMPLEMENTATION

The survey was implemented using Google Forms and distributed via the Prolific platform, with compensation set at \$12/hour. Participants were guided through the survey flow with embedded instructions and examples to ensure comprehension and engagement.

TRANSPARENCY

All survey items, design rationales, and filtering criteria are publicly documented to support repro-ducibility and public trust. This enables rigorous evaluation of generative agents' ability to simulate human attitudes under complex, emotionally and politically sensitive policy conditions.

# I   UNIVERSAL HEALTHCARE QUESTIONNAIRE

## MOTIVATION AND OBJECTIVE

This survey was designed to evaluate whether a structured reasoning system—based on Bayesian networks extracted from interviews and conditioned large language models (LLMs)—can simulate or recover human judgments on complex policy issues. In this case, we focus on universal healthcare, a topic involving tradeoffs across fairness, cost, autonomy, and trust.

Rather than simply measuring stance, the survey was constructed to expose the participant's reasoning pathway, enabling fidelity evaluation at both outcome and process levels.

## SURVEY STRUCTURE AND METHODOLOGY

The survey design draws on the four-stage cognitive model of survey response (Tourangeau et al., 2000):

- **Comprehension**: Questions were phrased clearly and definitions were provided (e.g., what universal healthcare entails).
- **Retrieval**: Participants were asked to recall relevant experiences (e.g., delays in care, interactions with public systems).
- **Judgment**: Participants evaluated tradeoffs and reflected on personal values.
- **Response**: Structured Likert scales captured quantified opinions.

## SURVEY COMPONENTS

The survey includes:

- **Stance Rating**: Support for universal healthcare on a 1–10 scale.
- **Personal Experience**: Items capturing healthcare access and insurance adequacy.
- **Baseline Reason Evaluation**: Participants rated 13 carefully constructed reasons (e.g., fairness, efficiency, innovation) for their general influence on stance.

## COUNTERFACTUAL SCENARIOS

To probe reasoning dynamics and test the model's sensitivity to causal perturbations, four counterfactual scenarios were introduced, each followed by a stance re-rating and a focused subset of reasons. Scenarios included:

1. National cost reduction with increased wait times.
2. Household savings of $3,000 annually.
3. Retention of private insurance alongside a public system.
4. Coverage limited to essential services.

Participants re-evaluated selected reasons in the context of each scenario (e.g., "I worry about tax increases" or "Universal healthcare might reduce personal choice in care") on a 1–5 scale, allowing analysis of belief shifts.

## REASON DESIGN

Reasons were drawn from qualitative policy discourse and refined to:

- Reflect distinct value dimensions (e.g., equality, responsibility, institutional trust).
- Avoid biasing language (neutral framing, no moral triggers).
- Enable both positive and negative stance justifications across political orientations.

Each reason was independently interpretable and mapped to latent causal factors in the underlying Bayesian model. Subsets of reasons were assigned to each counterfactual scenario to ensure relevance while reducing redundancy.

## J    SURVEILLANCE CAMERA QUESTIONNAIRE

### MOTIVATION AND OBJECTIVE

This survey is designed to evaluate the reasoning fidelity of structured models such as Bayesian Networks (BNs) when paired with large language models (LLMs). Specifically, it tests whether a BN+LLM system can simulate human responses to policy questions about public surveillance more faithfully than a baseline persona-based LLM. To do this, we use controlled question design inspired by cognitive science and causal reasoning frameworks.

### SURVEY STRUCTURE AND METHODOLOGY

The survey design draws on the four-stage cognitive model of survey response (Tourangeau et al., 2000):

1. **Comprehension**: Understand the question and context.
2. **Retrieval**: Recall relevant experiences and beliefs.
3. **Judgment**: Synthesize and evaluate relevant considerations.
4. **Response**: Map judgment to a scale-based response.

This model guides both our baseline attitude elicitation and our counterfactual design. The survey consists of:

**Section 1: Baseline Stance and Experience**    Participants rate their general support for public surveillance (1–10), followed by personal experiences such as feelings of safety, comfort, and negative interactions with surveillance technology.

**Section 2: General Reason Evaluation**    Participants evaluate the importance of twelve potential reasons (1–5 Likert scale) influencing their baseline stance, including factors like privacy, crime prevention, power misuse, and behavioral impacts.

**Section 3: Counterfactual Scenarios and Dynamic Reasoning**    Participants are then presented with three hypothetical surveillance policy changes:

- Crime Reduction vs. False Arrest Tradeoff
- Limited Data Retention (48h)
- Community-Controlled Surveillance

For each scenario:

- Participants rate how the new information affects their stance (1–10 scale).
- Then, they re-evaluate a scenario-specific subset of 3–5 reasons (1–5 scale) that are most relevant under the new condition.

This design allows us to evaluate whether the model (and human) responses adjust not only the final stance, but also the internal reasoning paths—a critical distinction for validating structural cognitive models.

### DESIGN HIGHLIGHTS

- **Cognitive fidelity**: Question wording avoids surface cues and forces reasoning across multiple values (e.g., privacy vs. safety, trust vs. control).
- **Counterfactual sensitivity**: Each scenario targets a specific edge in the causal BN, enabling us to observe how reason weights shift under perturbation.
- **Explanation delta**: By comparing reason weights before and after each scenario, we quantify whether the model exhibits structural adaptation or static stance mimicry.

### DATA COLLECTION AND IMPLEMENTATION

The survey was implemented using Google Forms and distributed via the Prolific platform, with compensation set at \$12/hour. Participants were guided through the survey flow with embedded instructions and examples to ensure comprehension and engagement.

TRANSPARENCY

All survey items, design rationales, and filtering criteria are publicly documented to support repro-ducibility and public trust. This enables rigorous evaluation of generative agents' ability to simulate human attitudes under complex, emotionally and politically sensitive policy conditions.

# K PROMPT

## K.1 TASK FORMATTING PROMPT

**System Prompt**

zoning: "You are an expert at analyzing conversations about urban policy to extract causal beliefs.
surveillance": "You are an expert at analyzing conversations about surveillance and public safety to extract causal beliefs.
healthcare": "You are an expert at analyzing conversations about healthcare policy to extract causal beliefs.

**User Prompt**

Based on the following conversation about {conversation_topic}, identify ALL question-answer pairs that reveal the person's beliefs about causal relationships between different factors.
Conversation: {context_text} Your task: 1. Find ALL Q&A pairs that show how the person believes one factor affects another (up to 10 pairs) 2. For each pair, create a direct question asking about the influence level using everyday language 3. Based on the person's answer, determine their belief about the effect.
Selection rule: - PRIORITIZE items with dependency_level >= 1 (needs-context). If fewer than 10 such items exist, then fill the remainder with the best dependency_level = 0 items. - Prefer diverse factor pairs; avoid near-duplicates.

Return JSON format as an array.
{answer_options_text}
Use simple, everyday language for the factors. Examples by topic:
Zoning: "building more housing" instead of "upzoning policies", "traffic congestion", "neighborhood character" .
Surveillance: "installing cameras" instead of "surveillance systems", "crime rates", "privacy concerns" .
Healthcare: "universal coverage" instead of "healthcare policy", "wait times", "healthcare costs".
Return up to 10 belief inference questions maximum.

## K.2 EVALUATION PROMPT

**System Prompt**

You are an expert psychologist specializing in Theory of Mind and belief inference.
Your task: analyze conversation transcripts to infer what the participant believes about causal relationships. Focus on understanding their mental model - what they think causes what, not what is objectively true.
Consider their background, conversation patterns, and implicit beliefs expressed through their responses. Base your inference strictly on evidence from their statements, not general assumptions.

**User Prompt**

{Context QA + Demographic information}
Based on the evidence above (including Conversation History and Person's Background), respond with ONLY the single letter (options_str) that best represents this person's belief.)

## L  EXTENDED DESIGN PRINCIPLES

**Open-ended reasoning as principle**   Our benchmark targets *reasoning as a dynamic and individualized process*, rather than static prediction. We therefore adopt an **open-ended elicitation principle**: instead of pre-defining fixed question banks, HugAgent uses a single *guiding question* to initiate a semi-structured conversation. All follow-up questions are generated adaptively within the same dialogue, grounded in the participant's own responses. This design enables *deep, conversational reasoning* to unfold while minimizing artificial scaffolding from the chatbot itself. We do not claim fully open-world coverage; rather, we emphasize *open-domain extensibility*: by simply swapping the guiding question, the benchmark can be ported to new domains while maintaining consistency in evaluation. Such minimal-interaction protocols align with prior work showing that lightweight conversational scaffolds preserve ecological validity in human reasoning studies (Van Someren et al., 1994; Clark, 1996; Sap et al., 2019b; Driess et al., 2023). **This principle directly motivates the guiding-question chatbot protocol we describe in Appendix D.**

**Proxy tasks of reasoning**   To evaluate whether models capture not only what individuals believe but also how their beliefs evolve, we operationalize reasoning through two proxy tasks: *belief state inference* (recovering stance and factor polarity from context) and *belief dynamics update* (predicting stance shifts and reweighting under new evidence). These tasks follow the tradition of modeling belief revision as a tractable proxy for underlying cognitive processes (Gopnik & Schulz, 2007; Sloman, 2009). While other proxies could be envisioned, these two are the most direct operationalizations of *individual reasoning trajectories*, balancing interpretability and task difficulty. **This motivates our benchmark's two-task structure, detailed in Appendix D.**

**Dual-track design: human and synthetic agents**   Human data provide ecological validity: rich, idiosyncratic reasoning paths embedded in natural language. Synthetic agents, by contrast, provide controllability and scale: fully specified stance profiles and deterministic update rules allow stress-testing model adaptation under known ground truth. Together, the two tracks are complementary: *humans as ecological baselines*, *synthetics as controlled stress tests*. This mirrors dual-track designs in cognitive science and simulation benchmarks, where naturalistic and synthetic data jointly enhance validity and reproducibility (Lake et al., 2017; Battaglia et al., 2018). Synthetic agents are not intended to replace human data but to serve as a complementary axis of evaluation. **This dual-track design is what anchors HugAgent between ecological realism and controlled generalization tests.**

**Extended Rationale: Synthetic Stage Justification**   Synthetic data in HugAgent follows the same legitimacy principles as established ToM and social reasoning benchmarks. Rather than letting LLMs freely invent beliefs, we first define a formal structure—a causal belief graph specifying nodes (beliefs), edges (causal relations), and interventions (external stimuli). Synthetic agents then evolve along this graph to generate new belief states and reasoning trajectories. The graph itself provides the ground truth for evaluation (e.g., stance updates, trajectory alignment), while LLMs merely render these states into natural language explanations. This ensures that labels are independently controlled and falsifiable, avoiding the risk of self-validation. As in prior benchmarks, sampled human verification is performed for quality assurance.

**Upper bound via test–retest reliability**   A natural question is whether human annotators could serve as the benchmark baseline. While this is common in many benchmarks, HugAgent tasks present unique challenges: they involve *long, naturalistic transcripts* and fine-grained belief trajectories. In principle, annotators could be asked to re-read transcripts and label stance updates, but such procedures are slow, error-prone, and risk conflating annotators' own heuristics with the original participant's reasoning. This creates a fidelity–feasibility tradeoff: while feasible, the outcome would be a proxy of *third-party interpretation*, rather than a faithful measure of the individual's reasoning process.

Instead, we adopt *test–retest reliability* as the human ceiling. Here, the same participant is re-sampled or re-interviewed, and the consistency of their own responses provides a direct measure of reliability. This practice is well established in psychology and survey research, and has been adopted in recent large-scale reasoning datasets facing similar challenges (Park et al., 2023; Toubia et al., 2025). Compared to annotator baselines, test–retest reliability offers a more precise and ecologically valid upper bound for model performance, aligned with the benchmark's goal of capturing intra-individual reasoning fidelity. **This principle defines how we report the human ceiling in HugAgent.**

## M  DATA FULL EXAMPLE

User's Demographic:

| Attribute | Example (Anonymized Participant) |
|---|---|
| Housing Experience | Has lived in the same residence for several years |
| Age | 30 |
| Moved Last Year | Same house 1 year ago |
| Housing Status | Owner-occupied |
| Transportation | Car / Truck / Van |
| Household Income | $75,000–$99,999 |
| Occupation | Sales and office occupations |
| Marital Status | Not married |
| Children | Has children |
| Neighborhood Safety | Very safe. I rarely worry about crime |
| Health Insurance | Private insurance, no disability |
| Education | High school graduate or equivalent |
| Citizenship | Native-born U.S. citizen |
| Financial Situation | Gets by, but money is tight |

Table 20: Example anonymized participant profile used in analysis (for illustration only). Personally identifiable details have been generalized or omitted.

INTERVIEW QA

*Note: The following excerpt reflects a simulated or anonymized participant's responses. It may contain biased or stereotypical opinions that do not represent the authors' or dataset creators' views. It is included purely for analysis of belief attribution and reasoning behavior.*

1. **Q:** To what extent do you support or oppose upzoning policies that allow for higher density housing in traditionally single-family neighborhoods? Please explain your reasoning.
   **A:** I don't support it at all. I'm worried that it'll cause overcrowding if cheaper apartments or housing were made. Aside from that, we know that statistically, lower income people tend to have more of the criminal population in them, isn't that right? So this might cause the crime rates to go up!

2. **Q:** What do you think are the most significant impacts, positive or negative, of increasing housing density in residential neighborhoods?
   **A:** I've mentioned the potential for crime rates to go up, that's the real worry here. Lots of new lower income people, lots of potential criminals.

3. **Q:** How do you think upzoning policies might affect housing affordability in urban areas?
   **A:** They'd most likely lower the price of rent because of "competition". But at what cost? The safety of the people!

4. **Q:** What impact do you believe increased housing density might have on neighborhood character and quality of life?
   **A:** Safety for sure. Low income places simply have more potential for crimes due to people being tempted to commit criminal acts for survival.

5. **Q:** How do you think upzoning might affect transportation systems and traffic congestion in cities?
   **A:** It's going to worsen! Look, there was a time when I used to take the bus to get to work every day when I still didn't have a car. I live in a big city and sometimes, the bus couldn't take all of us! That caused me to get late a couple of times since there wasn't even any standing room. So imagine, a rush of new low income people to this area, probably they don't have cars so they'll rely on buses, it'll just be extra strain on the buses and not everyone would be able to get on the bus at all.

6. **Q:** What role do you believe local government should play in regulating housing development and density?
   **A:** The government really shouldn't be too involved with many things. Just minimally involved. Less government involvement, the better.

7. **Q:** How might environmental concerns factor into decisions about urban density and zoning?
   **A:** I don't personally care about these so-called "environmental concerns". I'm not some

kind of environmental activist or terrified climate change believer. As long as something doesn't dump toxic waste or all sorts of hazardous material in my area, then it's good.

8. **Q:** What economic effects, both positive and negative, might result from changing zoning laws to allow more multi-family housing?

   **A:** More new people, more potential customers for businesses in the area obviously. BUT we also have to think that these are low income people if we're talking about low income housing. So businesses targeting low income people would most likely benefit, but the more upscale ones wouldn't.

9. **Q:** How do you think the interests of current residents versus future residents should be balanced when making zoning decisions?

   **A:** The current residents should ALWAYS be prioritized, they were there first. New people should always be considerate of the people living wherever they're planning to move to. It's just basic human decency.

10. **Q:** What role do you think social equity and access to opportunity play in discussions about zoning and housing policy?

    **A:** I am totally against EQUITY. Equity means taking opportunities away from someone in order to give it to somebody else who probably didn't earn it. I don't like the idea of redistributing what a successful person has.

11. **Q:** How confident are you that changes in Higher density housing lead to changes in Support for Upzoning? Does it have a positive effect (increasing it) or a negative effect (decreasing it)? How strong is this effect?

    **A:** It's going to be NEGATIVE. If we're talking about people, it's not just quantity that we're supposed to worry about, but also the quality. So we can say "Don't judge a book by their cover", but we also must think that people are in the situation they are for a reason. So if we're going to get flooded by low income people, we have to ask, "Why are they low income?" Of course not all low income people are bad, but majority of criminals are low income people.

12. **Q:** What factors do you think influence Support for Upzoning, and how strong is their impact? Please also indicate if these influences are positive (increasing) or negative (decreasing).

    **A:** Definitely the idea of SAFETY is a huge factor. Just imagine you live in a peaceful neighborhood where crime isn't really a problem, then suddenly a huge number of new low income people flood in to your community and suddenly kids start getting bullied at the playground, people start getting mugged left and right. Safety is really a big concern!

13. **Q:** Does Crime rates have a positive or negative effect on Support for Upzoning, and how significant is this effect? Does it have a positive effect (increasing it) or a negative effect (decreasing it)? How strong is this effect?

    **A:** That's what I've been talking about this entire conversation, the potential for CRIME! As I've already stated numerous times, it's a MAJOR concern and an influx of low income people would definitely affect the crime rate!

14. **Q:** Would small changes in Housing affordability lead to noticeable changes in Support for Upzoning, or would it take larger shifts? Does it have a positive effect (increasing it) or a negative effect (decreasing it)? How strong is this effect?

    **A:** At first people would probably think things will be better because rent might go down a bit, BUT that's not guaranteed. Second, SAFETY is really something that people are probably not willing to compromise.

15. **Q:** Would small changes in Safety lead to noticeable changes in Support for Upzoning, or would it take larger shifts? Does it have a positive effect (increasing it) or a negative effect (decreasing it)? How strong is this effect?

    **A:** If there's really no way about avoiding the creation of some kind of tall low income apartment building for the sake of "equity", then the next best thing would be to thoroughly do background checks on all the renters. For example, there should be strictly nobody in there with a criminal record.

16. **Q:** How would you describe the relationship between Low Income People and Support for Upzoning? Is it a strong or weak connection? Does it have a positive effect (increasing it) or a negative effect (decreasing it)? How strong is this effect?

    **A:** Well of course low income people would support the creation of low income rental

building. But the problem is that people already living in the community, like me, wouldn't support it at all for fears of safety worsening.

17. **Q:** Is the effect of Minimal Regulation on Support for Upzoning immediate, or does it take time to develop? Does it have a positive effect (increasing it) or a negative effect (decreasing it)? How strong is this effect?
**A:** Any policy takes TIME to develop. Rushed policies just end up in disaster because it won't be well thought out.

18. **Q:** Would small changes in Impact on businesses lead to noticeable changes in Support for Upzoning, or would it take larger shifts? Does it have a positive effect (increasing it) or a negative effect (decreasing it)? How strong is this effect?
**A:** No. As I've said, low income people will only provide benefit to businesses targeting low income customers. Mid to upscale businesses wouldn't benefit from them because they won't be able to afford their products and services. In short, not all businesses would be in support of having some kind of low income housing in the area if all they're going to be able to afford are low income stuff.

19. **Q:** How would you describe the relationship between Basic human decency and Support for Upzoning? Is it a strong or weak connection? Does it have a positive effect (increasing it) or a negative effect (decreasing it)? How strong is this effect?
**A:** There are people who make decisions based on feelings alone. Yes, they'll think it's "decent" to allow low income people to have low income housing in their community, BUT often, these people don't think about the consequences that would affect the people already living in the community. They are too focused on helping others that they don't realize they are causing harm to themselves.

20. **Q:** Is the effect of Redistribution on Support for Upzoning immediate, or does it take time to develop? Does it have a positive effect (increasing it) or a negative effect (decreasing it)? How strong is this effect?
**A:** That's definitely going to be a huge NEGATIVE right away. Nobody in their right mind would want themselves to be compromised for others. So let's think about what happens if in a moderately wealthy area, they allowed low income housing in the name of "equity". For actual home owners (not renters), the value of their properties would go down. These are properties that they've worked for years to maintain, and suddenly, in the name of "equity", is it alright to allow the values to go down? No of course not! So we have to always think about how low income housing would affect the people already living in the community.

21. **Q:** Would small changes in Negative Effect lead to noticeable changes in Support for Upzoning, or would it take larger shifts? Does it have a positive effect (increasing it) or a negative effect (decreasing it)? How strong is this effect?
**A:** No, it's called "Negative Effect" because it affects people in a bad way. Nobody would support anything like that knowingly.

22. **Q:** What factors affect Upzoning policies, and which ones have the strongest influence? Please also indicate if these influences are positive (increasing) or negative (decreasing).
**A:** As I've been saying this entire conversation, the major facor that affects people's support for low income housing is the SAFETY, the potential for crime rates to go up, and these things will definitely always affect support for low income housing negatively.

23. **Q:** Does Community Resistance to Upzoning have a positive or negative effect on Support for Upzoning, and how significant is this effect? Does it have a positive effect (increasing it) or a negative effect (decreasing it)? How strong is this effect?
**A:** Of course community resistance won't support low income housing, that's the point. People would resist these places from being built in order to protect the community from potential safety concerns.

24. **Q:** How would you describe the relationship between Time for policy development and Support for Upzoning? Is it a strong or weak connection? Does it have a positive effect (increasing it) or a negative effect (decreasing it)? How strong is this effect?
**A:** Of course "Time" will always have something to do with whether low income housing would be allowed or not. For example, maybe a politician would take his time forming some kind of bill concerning low income housing and he'll wait for enough public support before officially launching it in order to increase its chances of succeeding.

25. **Q:** Would small changes in Low Income Housing lead to noticeable changes in Support for Upzoning, or would it take larger shifts? Does it have a positive effect (increasing it) or a

negative effect (decreasing it)? How strong is this effect?
**A:** No, even small changes in low income housing won't change people's support for it because it will negatively affect the community. People already know it's most likely going to be the cause of many safety concerns aside from property devaluation.

26. **Q:** Does Equity have a positive or negative effect on Support for Upzoning, and how significant is this effect? Does it have a positive effect (increasing it) or a negative effect (decreasing it)? How strong is this effect?
**A:** Equity has NEGATIVE effects on people already living in the community, because the point of equity is to take from those people (land space) and to redistribute it to other people (the low income people). People might try to frame it as "helping the poor", but you can help poor people in other ways without harming the community.

27. **Q:** How would you describe the relationship between Support for low income housing and Support for Upzoning? Is it a strong or weak connection? Does it have a positive effect (increasing it) or a negative effect (decreasing it)? How strong is this effect?
**A:** People are directly against low income housing because it's more likely to bring bad stuff with it that good ones. The consequences outweigh the positives.

SAMPLE SURVEY: HOUSING / UPZONING

BASELINE STANCE

- Q1. On a scale from 1 to 10, how much do you support or oppose allowing bigger, taller apartment buildings in your neighborhood?

| 1 | 2 | 3 | 4 | 5 | 6 | 7 | 8 | 9 | 10 |
|---|---|---|---|---|---|---|---|---|---|
| Strongly Oppose | | | | Neutral | | | | | Strongly Support |

REASON EVALUATION (BASELINE)

Q1r. How much do the following reasons influence your general opinion on upzoning?

| Reason | Scale (1–5) |
|---|---|
| Building more homes helps with the housing crisis. (A) | |
| This gives more housing choices for middle- and lower-income people. (B) | |
| Taller buildings might change the look and feel of the neighborhood. (C) | |
| More traffic and parking is a real concern. (D) | |
| I'm worried about my property value or investment. (I) | |

SCENARIO 1: RENT DROP

- Q2. After the city allows more apartments in low-density areas, rent prices drop 10–15%. Your monthly rent is noticeably lower. It's easier to find a decent place. How would this affect your stance?

| 1 | 2 | 3 | 4 | 5 | 6 | 7 | 8 | 9 | 10 |
|---|---|---|---|---|---|---|---|---|---|
| Much Less Supportive | | | | Neutral | | | | | Much More Supportive |

REASON EVALUATION (SCENARIO 1)

Q2r. To what extent do the following reasons influence your stance?

| Reason | Scale (1–5) |
|---|---|
| Building more homes helps with the housing crisis. (A) | |
| More traffic and parking is a real concern. (D) | |
| Everyone should help handle the growth. (N) | |
| More people means more business for local shops. (O) | |
| I'd worry about noise and crowding on my block. (P) | |

## N  RATIONALE FOR INDIVIDUAL CROSS-DOMAIN TRANSFER

We assume that cross-domain personalization is feasible because individuals express stable, value-laden cues throughout natural language. These cues are not tied to a single domain; rather, they reflect underlying principles that consistently shape preferences across contexts.

**Quantitative Evidence: Consistent In- to Out-of-Domain Transfer**  To empirically support this assumption, we report cross-domain transfer results for four individuals. For each person, GPT-4o was evaluated across five independent runs. Across all individuals and all runs, we observe the same ordering:

$$\text{In-domain accuracy} > \text{Cross-domain accuracy} > \text{No-context accuracy}.$$

This strict ordering across all 20 settings (4 individuals $\times$ 5 runs) provides a statistically grounded indication that the observed cross-domain transfer is reliable and not an artifact of noise.

| User | Experiment | Belief State Inference (BSI) | Belief Dynamics Update (BDU) | | | ATI (% ↑) |
|---|---|---|---|---|---|---|
| | | Acc. (% ↑) | Acc. (% ↑) | MAE (↓) | Dir. Acc. (% ↑) | |
| User 1 (n-BSI = 9,  n-BDU = 48) | In Domain | $88.89^{\pm 0.01}$ | $91.67^{\pm 0.01}$ | $0.57^{\pm 0.02}$ | $85.00^{\pm 0.01}$ | $87.88^{\pm 0.05}$ |
| | Cross Domain | $55.56^{\pm 0.01}$ | $54.58^{\pm 3.33}$ | $1.38^{\pm 0.03}$ | $50.00^{\pm 0.01}$ | $55.28^{\pm 0.51}$ |
| | No Context | $33.33^{\pm 0.01}$ | $8.33^{\pm 0.01}$ | $2.33^{\pm 0.02}$ | $18.00^{\pm 3.67}$ | $27.42^{\pm 0.91}$ |
| User 2 (n-BSI = 13,  n-BDU = 41) | In Domain | $92.31^{\pm 0.01}$ | $91.71^{\pm 1.95}$ | $0.58^{\pm 0.02}$ | $65.00^{\pm 0.01}$ | $84.57^{\pm 0.30}$ |
| | Cross Domain | $92.31^{\pm 0.01}$ | $83.90^{\pm 1.19}$ | $0.97^{\pm 0.02}$ | $73.00^{\pm 9.80}$ | $84.37^{\pm 2.49}$ |
| | No Context | $30.77^{\pm 9.73}$ | $39.02^{\pm 3.45}$ | $1.68^{\pm 0.04}$ | $65.00^{\pm 0.01}$ | $43.78^{\pm 4.51}$ |
| User 3 (n-BSI = 5,  n-BDU = 24) | In Domain | $66.67^{\pm 0.01}$ | $64.23^{\pm 2.48}$ | $1.26^{\pm 0.04}$ | $57.50^{\pm 0.01}$ | $64.29^{\pm 0.38}$ |
| | Cross Domain | $66.67^{\pm 0.01}$ | $28.57^{\pm 1.88}$ | $1.87^{\pm 0.04}$ | $65.00^{\pm 0.01}$ | $59.80^{\pm 0.34}$ |
| | No Context | $46.67^{\pm 9.79}$ | $19.22^{\pm 1.21}$ | $1.95^{\pm 0.03}$ | $50.00^{\pm 0.01}$ | $44.63^{\pm 5.05}$ |
| User 4 (n-BSI = 7,  n-BDU = 38) | In Domain | $57.14^{\pm 0.01}$ | $42.11^{\pm 3.33}$ | $1.72^{\pm 0.04}$ | $92.50^{\pm 0.01}$ | $64.10^{\pm 0.52}$ |
| | Cross Domain | $51.43^{\pm 0.69}$ | $30.00^{\pm 2.11}$ | $1.79^{\pm 0.02}$ | $85.00^{\pm 0.01}$ | $57.63^{\pm 3.47}$ |
| | No Context | $31.43^{\pm 5.71}$ | $30.00^{\pm 1.29}$ | $1.79^{\pm 0.03}$ | $77.50^{\pm 0.01}$ | $45.73^{\pm 2.76}$ |

Table 21:  Comparison of GPT-4o performance across In-domain, Cross-domain, and No-context conditions for four representative users (mean $\pm$ std over five runs).

**Qualitative Evidence: Stable Value Dimensions Across Domains**  To complement the quantitative results, we provide a qualitative analysis of one participant's transcript. The individual expresses a coherent set of value-laden principles that appear across healthcare, surveillance, and zoning. These principles naturally support cross-domain generalization from a single personalized conversation.

The following quotations reflect individual participant beliefs and not normative statements endorsed by the authors.

1. **A safety-first orientation that generalizes across contexts.**
   In zoning, the participant frames upzoning as a threat to public safety:

   > "If cheaper apartments were made, it'll cause overcrowding, and that means more low-income people moving in... which will make crime rates go up!"

   In surveillance, the same concern motivates strong support for camera deployment:

   > "Cameras are a tool of preventing potential crime and it's effective."

   A safety-oriented value extracted from zoning thus directly predicts pro-surveillance attitudes.

2. **A stable opposition to redistribution across domains.**
   In healthcare:

   > "Nobody is entitled to other people's money. Taxpayers shouldn't be forced to pay for other people's medical needs."

   In zoning:

   > "Equity means taking opportunities away from some to redistribute them to others. I don't like redistributing what a successful person has."

   The same anti-redistribution stance explains resistance to equity-oriented policies in both domains.

3. **A consistent negative framing of low-income groups as a societal risk.**
   In zoning:

   > "Lots of new low-income people means lots of potential criminals... majority of criminals are low income."

   In healthcare:

   > "The only people who'd support universal healthcare are those who can't afford healthcare themselves."

   This stable attribution shapes preferences across otherwise unrelated policy areas.

4. **Selective distrust of government intervention—except in policing.**
   In healthcare:

   > "If the government runs it, everything becomes standardized and we're forced to pay for others."

   In zoning:

   > "The government really shouldn't be too involved... less government involvement, the better."

   Yet in surveillance:

   > "If police use cameras to catch criminals, support will grow."

   This produces a predictable cross-domain pattern: skepticism toward government redistribution and regulation, but support for expanded government authority in security contexts.

**Summary**   These four dimensions (1) safety orientation, (2) redistribution aversion, (3) negative out-group attribution, and (4) selective distrust of government are expressed consistently across the participant's interview.

Taken together, the quantitative seed-level consistency and qualitative evidence provide a clear explanation for why individual cross-domain personalization is expected and why our empirical findings are robust.

## O    QUALITY-CONTROL PROTOCOL

We applied a standardized protocol to ensure that only participants with reliable and reproducible data were retained. The following criteria were applied sequentially:

1. **Redundant responses**: cases where the participant repeatedly produced near-identical statements without substantive variation.

2. **Meta-level questioning**: transcripts dominated by repeated challenges to the validity of the task itself rather than substantive reasoning about the topic.

3. **Insufficient length**: responses falling below a minimum threshold of tokens or turns, preventing meaningful inference of reasoning structure.

4. **Sparse causal belief networks**: chatbot elicitation yielding fewer than five unique nodes, limiting the interpretability of downstream causal graph construction.

This filtering ensured that the retained dataset reflects consistent engagement with the task, while minimizing artifacts that could compromise the validity of subsequent analyses.

## P  SYNTHETIC AGENT CONSTRUCTION (ALGORITHMIC)

### P.1  PROBLEM SETTING AND NOTATION

We construct a synthetic agent population $\mathcal{A}$ for topics $\mathcal{T} = \{\text{ZONING, HEALTHCARE, SURVEILLANCE}\}$. Each agent $a \in \mathcal{A}$ for topic $t \in \mathcal{T}$ has (i) a Causal Belief Network (CBN) $G = (V, E)$, (ii) a demographic profile $d$, (iii) an initial belief state $b_0 = (s_0, \mathbf{w}_0)$ with stance $s_0 \in \{1, \dots, 10\}$ and reason weights $\mathbf{w}_0 \in \Delta^{K-1}$, and (iv) a deterministic update operator $U$ that maps $(b_t, e) \mapsto b_{t+1}$ given an intervention $e$.

**Topic-level statistics.** From human CBN corpora we estimate topic-specific sufficient statistics $\Theta_t = \{\mu_n, \sigma_n, \mu_e, \sigma_e, \alpha_{\text{imp}}, \beta_{\text{imp}}, \alpha_{\text{conf}}, \beta_{\text{conf}}, \mathcal{V}\}$: node/edge count moments $(\mu_n, \sigma_n)$ and $(\mu_e, \sigma_e)$, Beta parameters for node importance and edge confidence, and a topic vocabulary $\mathcal{V}$ for label generation. A content hash over source JSON files ensures cache validity.

### P.2  CBN SAMPLING MODEL

**Graph size.** We draw $|V| \sim \text{TruncNorm}(\mu_n, \sigma_n, [n_{\min}, n_{\max}])$ and $|E| \sim \min\{\text{LogNormal}(\mu'_e, \sigma'_e), |V|(|V|-1)/2\}$ with log-space moments chosen to match $(\mu_e, \sigma_e)$.

**Node attributes.** For each $v \in V$:

$$\text{importance}(v) \sim \text{Beta}(\alpha_{\text{imp}}, \beta_{\text{imp}}), \quad \text{evidence}(v) \sim \text{Poisson}(\lambda_t).$$

Text label $\ell(v)$ is generated by sampling $m \in \{2, 3, 4\}$ tokens from $\mathcal{V}$ (frequency-weighted) and filling a topic template.

**Edge attributes and topology.** We form a hub–spoke backbone plus random residual edges. For each selected hub $h$ (top-$q$ by importance), connect $h \to u$ for $u \in V \setminus \{h\}$ with probability proportional to $0.7 \cdot \text{importance}(u) + 0.3$ until a target degree. Each edge $e = (u \to v)$ gets:

$$\text{conf}(e) \sim \text{Beta}(\alpha_{\text{conf}}, \beta_{\text{conf}}), \quad \text{sign}(e) \in \{-1, 0, +1\} \text{ with } p_+, p_0, p_-, \quad \text{weight}(e) \sim \text{Beta}(\alpha_w, \beta_w).$$

We then add random non-duplicate edges until $|E|$ is reached.

**Stance node selection.** Let $\deg(v)$ be (undirected) degree. Define score $\sigma(v) = 0.7 \cdot \text{importance}(v) + 0.3 \cdot \deg(v)/\max_u \deg(u)$. Sample stance node $v^\star$ from top-3 nodes according to $\sigma(v)$ (softmax). Ensure weak connectivity from $v^\star$ to all nodes (add minimal edges if needed).

### P.3  DETERMINISTIC UPDATE OPERATOR

Let reasons be a fixed topic-specific set $\{r_1, \dots, r_K\}$ aligned to $V$. Given intervention $e$ encoded as factor deltas $\Delta \mathbf{f} \in \mathbb{R}^{|V|}$, we update stance and reason weights:

$$s_{t+1} = \text{clip}\left(s_t + \eta_s \sum_{(u \to \text{stance}) \in E} \text{sign}_u \, w_u \, \Delta f_u, \, 1, \, 10\right), \tag{2}$$

$$\mathbf{w}_{t+1} = \text{Normalize}(\mathbf{w}_t + \eta_w \mathbf{M} \Delta \mathbf{f}), \quad \mathbf{M}_{k,u} = g(r_k, u), \tag{3}$$

where $w_u$ is the edge weight into stance, $\eta_s, \eta_w$ are step sizes, and $g(\cdot, \cdot)$ aligns reasons to graph nodes (one-hot or soft map). This yields reproducible ground truth for stance shifts and reweighting.

### P.4  DEMOGRAPHIC GENERATOR AND COUPLING

We sample a demographic profile $d$ with correlated marginals: age bands, gender, education, income, housing status, employment, location, children, and rent-burden. Simple rules induce a small stance prior $\delta(d)$ (e.g., renter $\Rightarrow +\delta$ on pro-development; older age $\Rightarrow -\delta$ on rapid change). We set $s_0 \leftarrow \text{clip}(s_0 + \delta(d), 1, 10)$.

### P.5  NATURAL LANGUAGE REALIZATION

A realization module renders $(G, e, b_t, b_{t+1})$ to text: (i) paraphrase $e$ with templates; (ii) describe reasons using top-$m$ nodes connected to $v^\star$ by high-confidence edges; (iii) optionally ask/answer interview-style QAs. LLMs are used strictly as a *renderer*; labels remain from the scripted dynamics in §P.3 to avoid circularity.

---

**Algorithm 1:** GenerateAgent($t, \Theta_t,$ seeds)

---

**Input:** topic $t$, stats $\Theta_t$, RNG seeds
**Output:** agent $(G, d, b_0, \{(e_j, b_j)\}_{j=1}^m, \text{text})$

1  Sample $|V|, |E|$; create $V$ with `importance`, `evidence`, label from $\mathcal{V}$;
2  Build $E$ via hubs + random; assign `conf`, `sign`, `weight`;
3  Select stance node $v^\star$; ensure connectivity;
4  Sample demographics $d$; set $b_0 = (s_0, \mathbf{w}_0)$ and apply prior $\delta(d)$;
5  **for** $j = 1$ **to** $m$ **do**
6      Sample intervention $e_j$ (topic-specific deltas $\Delta \mathbf{f}$);
7      $b_j \leftarrow U(b_{j-1}, e_j)$ using Eqns. (1)–(2);
8      Realize $(e_j, b_{j-1}, b_j)$ to text; append to transcript;
9  **return** packaged JSON: graph $G$, demographics $d$, $\{(e_j, b_{j-1}, b_j)\}$, transcript;

---

### P.6 END-TO-END GENERATION

### P.7 COMPLEXITY AND SCALING

Graph sampling is $O(|V| + |E|)$; hub wiring adds $O(|V| \log |V|)$ for sorting. Per-agent conversation of $m$ turns is $O(m)$ render calls. The pipeline trivially parallelizes across agents and topics.

### P.8 QUALITY CONTROL AND DETERMINISM

**Graph validity:** degree bounds, stance reachability, parameter ranges. **Topic relevance:** label vocabulary coverage threshold. **Determinism:** all stochastic steps are seeded; topic stats are cached with file hashes. **Leakage control:** interventions and post-update labels never appear in the dialogue context used for model evaluation.

### P.9 RELEASE SCHEMA

Records are released as JSON:

- `belief_graph`: nodes with `label`, `importance`; edges with `source`, `target`, `sign`, `weight`, `confidence`; stance node id.
- `demographic`: age, gender, education, income, housing, employment, location, children, burden.
- `state_before`/`state_after`: stance (1–10), reason weights (1–5 or normalized).
- `intervention`: structured deltas and a natural-language paraphrase.
- `transcript`: ordered QA pairs (renderer output).

### P.10 HUMAN–SYNTHETIC SIMILARITY

To assess whether synthetic agents provide a faithful approximation of human reasoning structures, we compare structural statistics of belief graphs across the two tracks. As shown in Figure 19, the distributions of key properties—including graph size (nodes, edges), sparsity (edge density, average degree), and semantic alignment (importance, confidence, anchor-node ratio)—exhibit strong overlap between real and synthetic agents. Notably, synthetic graphs reproduce the long-tailed variation in node and edge counts observed in human data, while maintaining comparable distributions of stance-related weights. This alignment suggests that the synthetic track can serve as a scalable proxy for human reasoning traces, capturing core structural regularities even as it abstracts away from individual variability.

### P.11 LIMITATIONS

Synthetic agents offer coverage and ablation control but abstract from human variability (noise, inconsistency, framing sensitivity). Thus, results on this track are *stress tests* and should be interpreted alongside the human-grounded track, which provides the ecological ceiling.

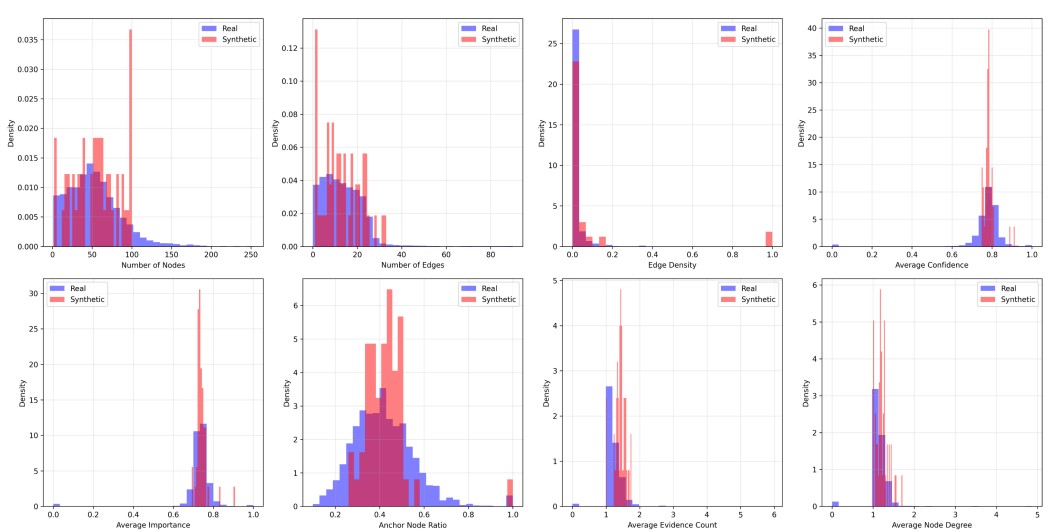

Figure 19: Distributional comparison of structural statistics between human-grounded (blue) and synthetic (red) belief graphs. Synthetic graphs replicate key patterns such as node/edge counts, confidence levels, and anchor-node ratios, supporting their use as controlled stress-test agents.

## Q  BENCHMARK TASK STRUCTURE

To clarify how HugAgent maps input materials to evaluation tasks, we provide here a consolidated overview of the benchmark structure. As shown in Figure 20, raw inputs include (i) demographic profiles, (ii) structured questionnaires, and (iii) open-ended chatbot transcripts. These inputs are transformed into two core task families:

- **Task 1: Belief State Inference.** Given a participant's responses and contextual cues, models must infer the person's stance and factor-level attribution. Example questions include: "Does the respondent view low-income housing as a positive or negative effect on property values?"

- **Task 2: Belief Dynamics Update.** After an intervention (e.g., rent decrease, policy change, technological improvement), models must predict both the stance shift (1–10 scale) and the reweighting of reasons (1–5 scale). Example questions include: "How would a 10% reduction in rents affect the respondent's stance on upzoning?"

Each topic domain—*zoning*, *healthcare*, and *surveillance*—is instantiated with multiple scenarios and corresponding reason mappings. This ensures comparability across domains while preserving topic-specific ecological validity.

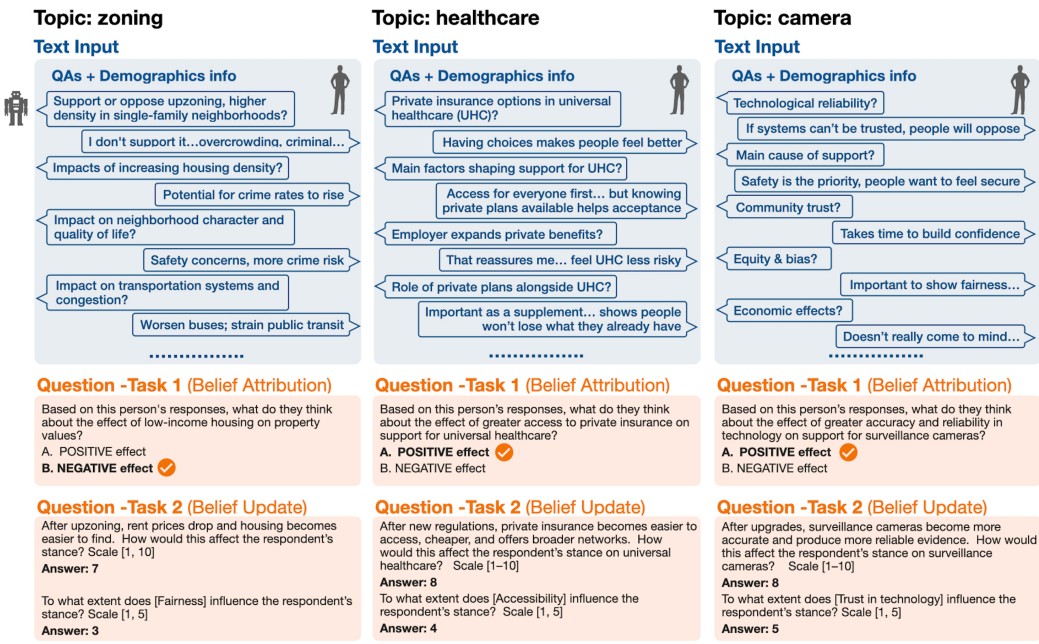

Figure 20: Overview of the HugAgent benchmark structure. Inputs (demographics, questionnaires, and transcripts) are mapped to outputs, including **belief state inference** (Task 1) and **belief dynamics update** (Task 2).

## R    USER JOURNEY AND USE CASES OF TRACE-YOUR-THINKING (A SEMI-STRUCTURED CHATBOT ELICITING HUMAN REASONING)

This appendix provides a detailed user guide and representative use cases for TRACE-YOUR-THINKING, our semi-structured chatbot system designed to elicit human reasoning at scale. We describe both participant-facing (user) and researcher-facing (admin) views, followed by system outputs and illustrative use cases. We use open science practices as an example here. Our design emphasizes three goals: (i) lowering barriers for participants, (ii) giving researchers flexible and reliable control, and (iii) producing structured outputs that make reasoning analyzable at scale.

PARTICIPANT JOURNEY (USER VIEW)

Participants experience a streamlined workflow that reduces friction while maximizing the richness of collected reasoning.

**Step 1: Consent and ID submission.** Recruitment begins on the Prolific platform, where participants are shown eligibility criteria and compensation details. Upon accepting the study, they are redirected to a Google Form where they confirm basic requirements (age $\geq 18$, residence within a specified region, consent for anonymous data usage). Entering their Prolific ID links the responses to the recruitment system, enabling follow-ups without storing personal identifiers, as is shown in Figure 21.

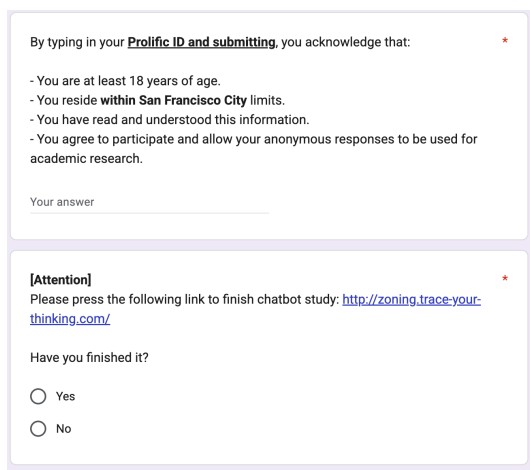

Figure 21: Participant view of the onboarding and interview flow (Consent)

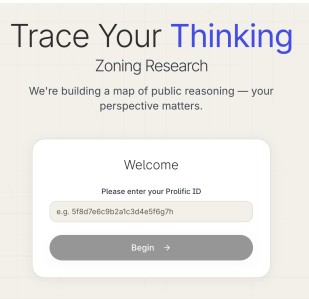

Figure 22: Participant view of the welcome page

**Step 2: Login and onboarding.** Participants are then redirected to the TRACE-YOUR-THINKING website. After inputting their Prolific ID, they are guided through a short tutorial. This tutorial introduces input modalities (typed text vs. voice-to-text) and explicitly informs users that all answers can be revised either immediately or retrospectively. A persistent progress bar at the top of the interface communicates task completion, reducing dropout risk by making expectations transparent. This part is shown in Figure 22.

**Step 3: Semi-structured interview.** The core of the participant journey is the semi-structured interview, which unfolds in a guided yet flexible flow: introduction $\rightarrow$ guiding questions $\rightarrow$ follow-up probes $\rightarrow$ final review. This design balances standardization with open-ended flexibility, allowing for wide variation in content, style, and depth of responses.

Participants can choose between typed responses and spoken input, enabling a think-aloud protocol that captures more spontaneous reasoning processes. Figure 23 illustrates the onboarding screen where both input modes are explained, while Figure 24 shows a participant actively using voice

Trace Your Thinking                                                                ID: test

Current Question
You can switch between voice and text modes at any time. In voice mode, the microphone button will
show as red when recording is active. During transcription, you'll see 'Transcribing your speech...' and
your answer will be automatically submitted once ready. Feel comfortable with the system now?

Interview Progress                                                                          18%

Welcome to the Trace Your Thinking interview system. This platform is designed to
capture your thoughts about urban zoning and housing development. Would you like me
to guide you through using this system?

1

You have two options for answering questions: voice recording or typing. For voice
recording, press the microphone button (🎤) or the Space key to start recording. When
finished, press the microphone button or Space key again to stop. You can also press
ESC to cancel. The system will automatically transcribe and submit your response after
recording. Would you like to try voice recording now?

Yes, please

For text input, click the message icon (💬) to switch modes. Type your response in the
text box. Press Ctrl+Enter or click the send button (➤) to submit. You can press ESC to
clear your text. Would you like to try text input now?

Let me have a try

You can switch between voice and text modes at any time. In voice mode, the
microphone button will show as red when recording is active. During transcription, you'll
see 'Transcribing your speech...' and your answer will be automatically submitted once
ready. Feel comfortable with the system now?

🎤        Press mic button or Space key to start speaking        💬
Press Space to start recording • ESC to cancel

Figure 23: Participant view of the onboarding of the chatbot, including audio and text input

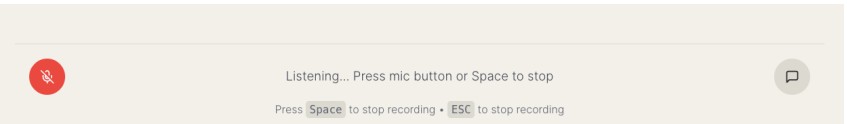

Figure 24: Participant view of using audio to give the answer

input to answer a question. Once the interview begins, participants can monitor their progress via a
persistent progress bar (Figure 25), which reduces fatigue by making task completion transparent.
During processing, the interface will show the processing status while still generating new questions
and allow users to answer (Figure 26).

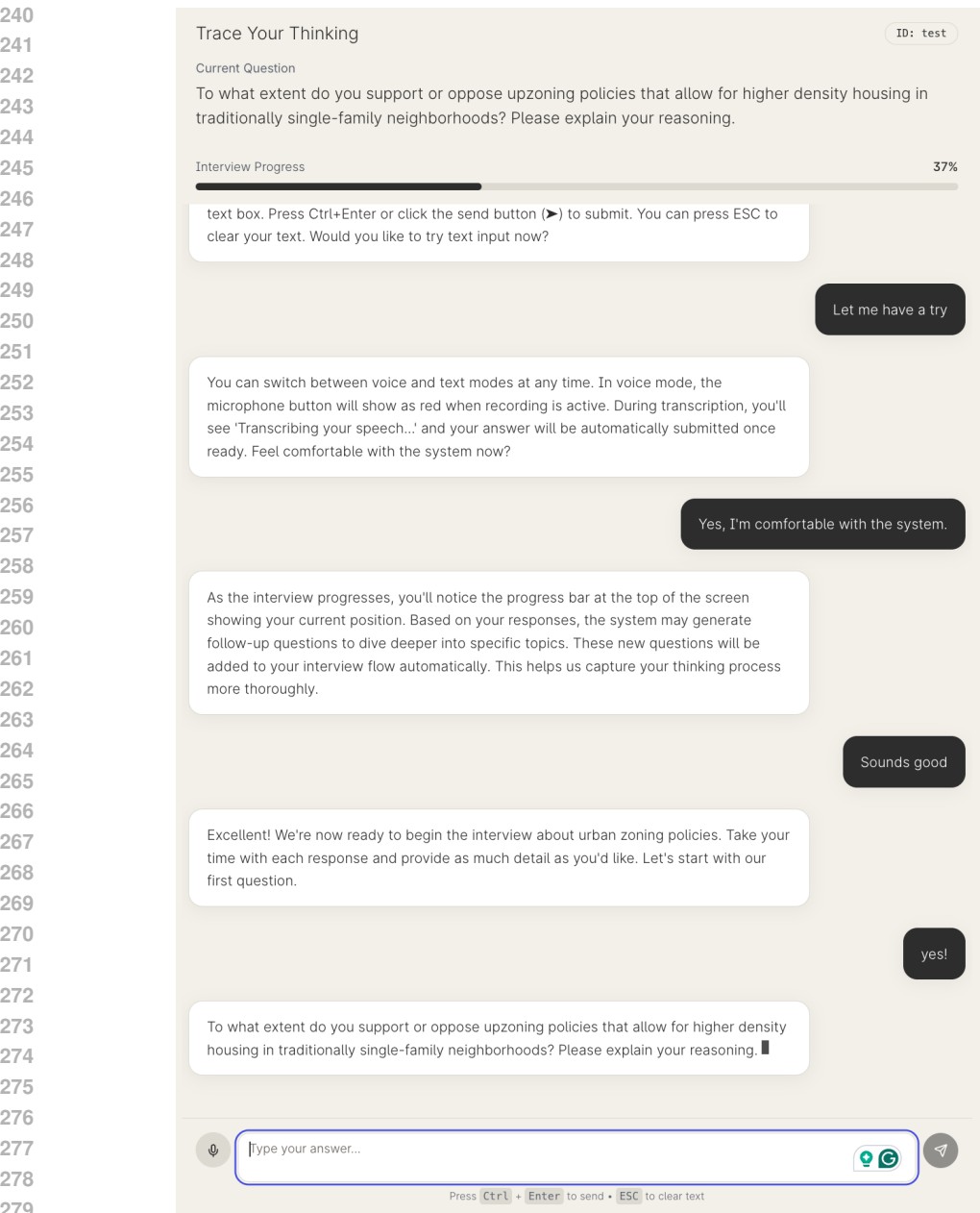

Figure 25: Participant view of answering questions and the interview progress

Participants can not only edit their responses immediately but also review the entire transcript at the end of the interview. As shown in Figures 27 and 28, the system presents an overview of all questions and answers, enabling users to backtrack, refine, and self-correct their reasoning. This mirrors how real-world reasoning often evolves over multiple passes rather than being fixed in a single draft.

**Step 4: Submission and compensation.** Once satisfied, participants submit their responses. Figures 29 and 30 demonstrate the submission stage, where participants re-enter their Prolific ID to confirm completion and finalize their session. The system redirects them back to Prolific, which automatically verifies completion and issues compensation. This tight integration ensures high-quality participation while minimizing administrative overhead.

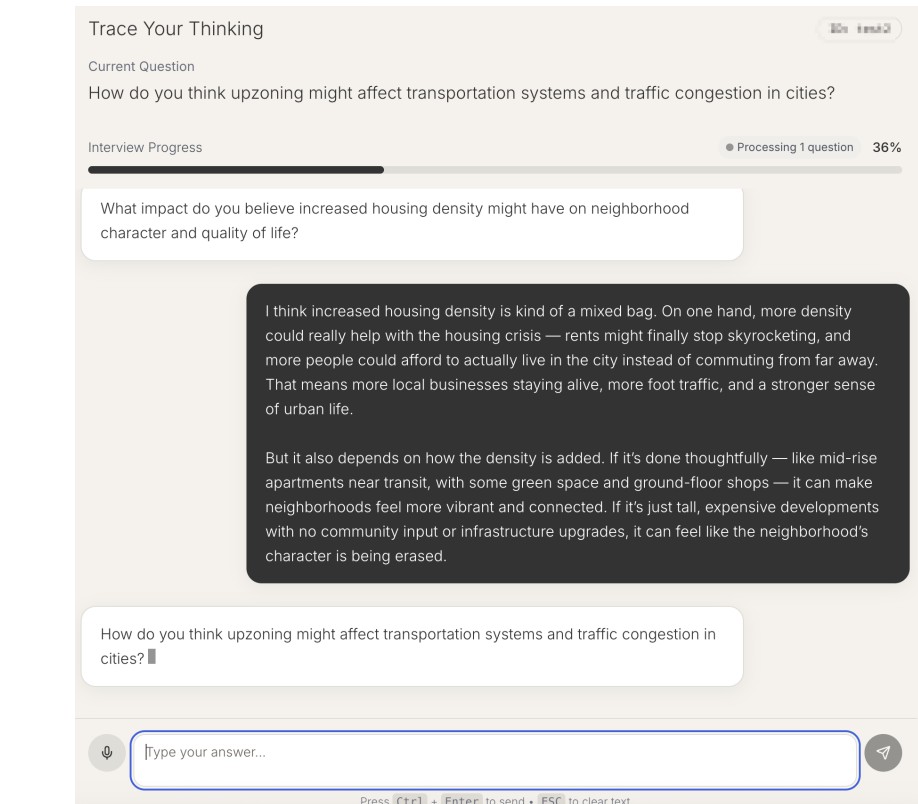

Figure 26: Participant view while AI is processing questions, but the participant can still answer the following questions without waiting

### R.1 RESEARCHER JOURNEY (ADMIN VIEW)

The system provides a dedicated control panel that makes data collection transparent, configurable, and scalable. Unlike static survey platforms, admins can adapt the study design on the fly and extract structured reasoning outputs.

**Recruitment integration.** Admins can publish tasks directly on Prolific, embedding the study link into recruitment posts. Prolific's filters (approval rate, demographics, geography) allow targeted participant pools, while stored Prolific IDs support longitudinal follow-ups. This design enables researchers to re-engage the same individuals across time or across topics, making it uniquely suitable for longitudinal reasoning studies.

**Session management.** The Session Management dashboard (Fig. 31) displays all ongoing and completed interviews with metadata including status, progress, and timestamps. From this panel, admins can (i) reorder questions, (ii) export raw QA data, or (iii) export causal graphs for downstream analysis. This unified view makes it easy to monitor study progress at scale and to recover high-fidelity reasoning traces.

**Configurable guiding questions.** Admins can design and adjust the interview protocol using a guiding question editor (Fig. 32). Each question has metadata (short text, full text, category), can be toggled on/off, and can be reordered dynamically. This flexibility makes it possible to test multiple hypotheses without rewriting the underlying system. In practice, this feature has been used to swap tutorial vs. research questions and to experiment with different probing strategies, making the platform versatile for diverse research programs.

**Global settings.** Admins can set a global interview topic (e.g., policy, healthcare, surveillance) with a single configuration (Fig. 33). This allows open-ended reasoning tasks to be deployed across arbitrary domains, ensuring that the platform is not tied to a fixed task. In effect, the system generalizes

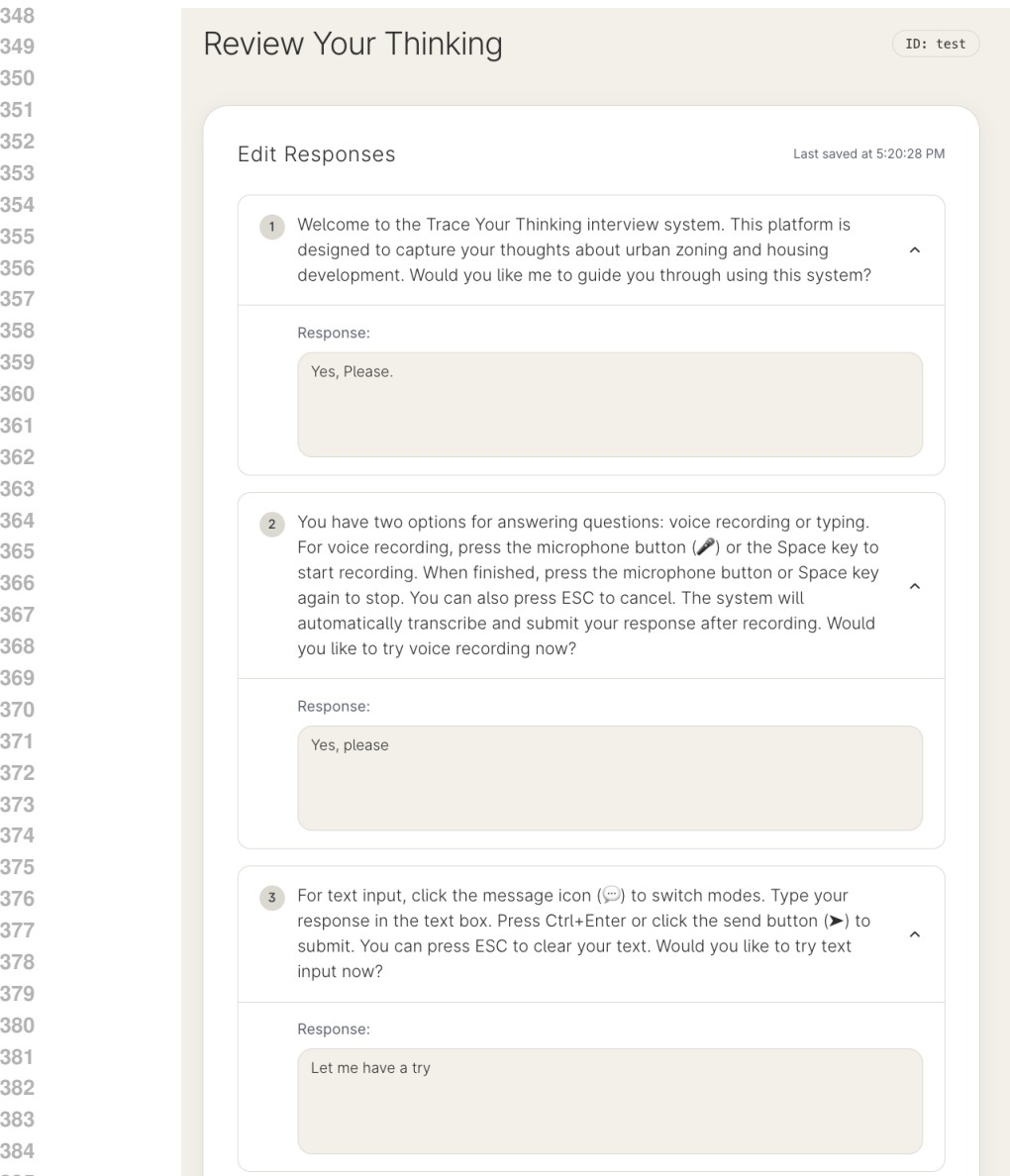

Figure 27: Participant view of the overview of the questions and answers. One can edit and go back to any of their answer.

beyond a dataset-collection tool to become a reusable infrastructure for eliciting reasoning in any domain.

## R.2 RESEARCH OUTPUTS (SYSTEM FEATURES)

The system is designed to produce outputs that go beyond raw transcripts, giving researchers structured and analyzable data.

**Raw QA transcripts.** All participant responses are preserved verbatim (Fig. 34). This ensures that qualitative nuances (hesitations, personal anecdotes, colloquial phrasing) are not lost. At the same time, transcripts provide the raw material for quantitative benchmarking, enabling evaluations of stance classification, belief calibration, and reasoning depth. The ability to capture both structured and noisy responses is a feature, not a limitation: it reflects the diversity of real-world human reasoning.

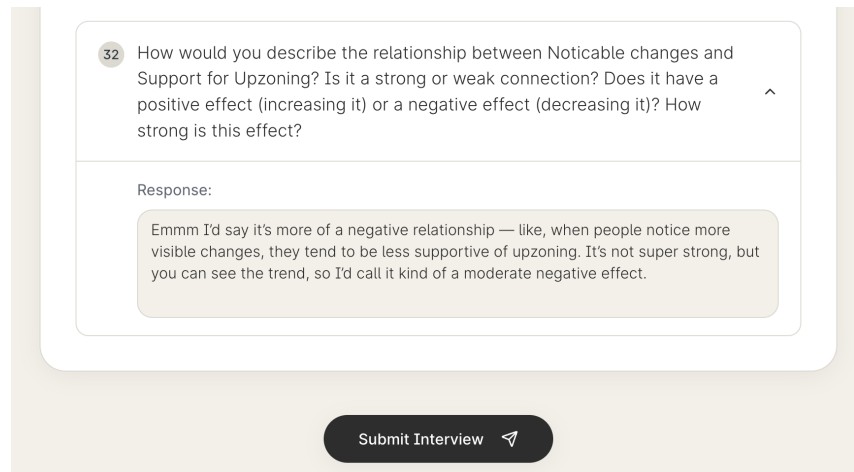

Figure 28: Participant view of the end of the overview of the questions and answers

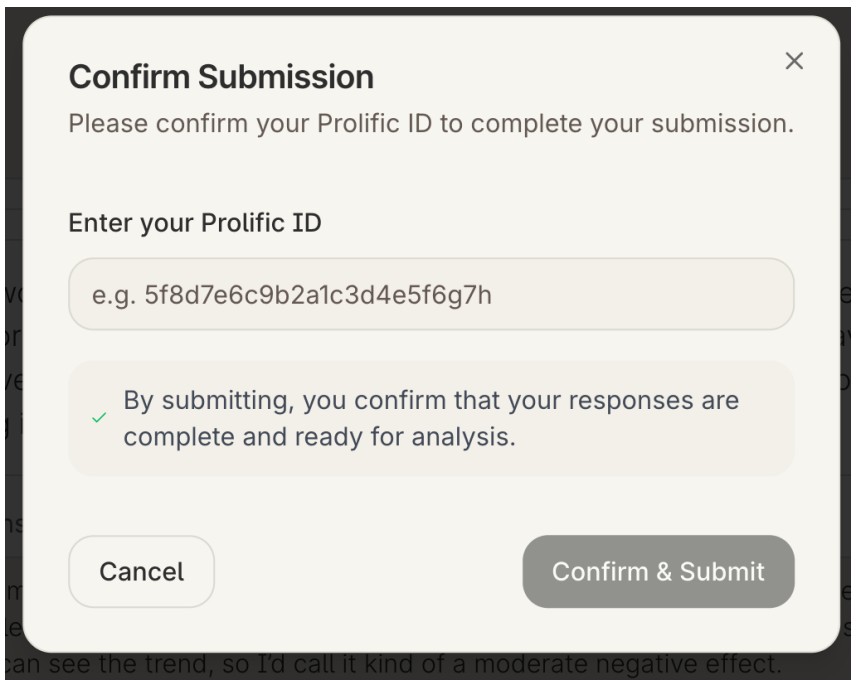

Figure 29: Participant view of reentering prolific id for submission

**Dynamic causal graphs.** The distinctive feature of TRACE-YOUR-THINKING is the automatic construction of causal graphs in real time (Fig. 35). As participants answer questions, the system incrementally extracts stance nodes (opinions), belief nodes (anchors), and candidate nodes (supporting reasons). The graph expands as reasoning unfolds, producing a structured representation of how beliefs and justifications interconnect. This design is important for two reasons: (i) it transforms unstructured reasoning into analyzable graph data, and (ii) it enables researchers to trace belief updates step by step, rather than relying only on final outcomes. These graphs can be exported for downstream tasks such as reasoning alignment, structural consistency evaluation, or cross-domain transfer prediction.

R.3   USE CASES

The flexibility of the system enables multiple research paradigms:

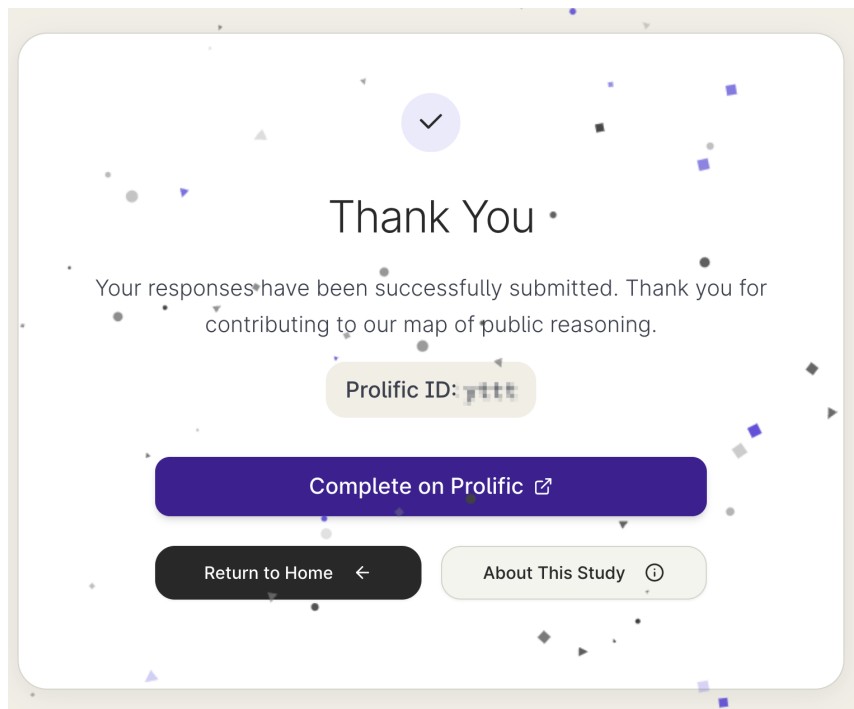

Figure 30: Participant view of the end of the test

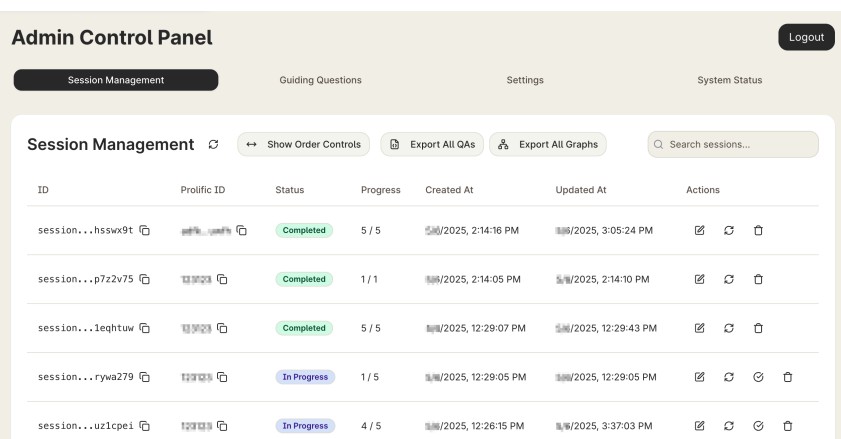

Figure 31: Admin session management panel with status tracking, progress monitoring, and export functionality. Researchers can monitor studies in real time and batch export reasoning data.

- **Baseline data collection:** Build large-scale corpora of reasoning traces in a controlled domain (e.g., housing policy), establishing benchmarks for human reasoning diversity.
- **Cross-domain transfer:** Instantly switch topics (e.g., from zoning to healthcare) by editing global settings, to study how reasoning patterns generalize across domains.
- **Longitudinal studies:** Re-engage the same participants over weeks or months via Prolific IDs, enabling the study of belief updates and reasoning drift.
- **Human–model benchmarking:** Compare LLM predictions against human causal graphs to quantify intra-agent fidelity, context sensitivity, and adaptation gaps.

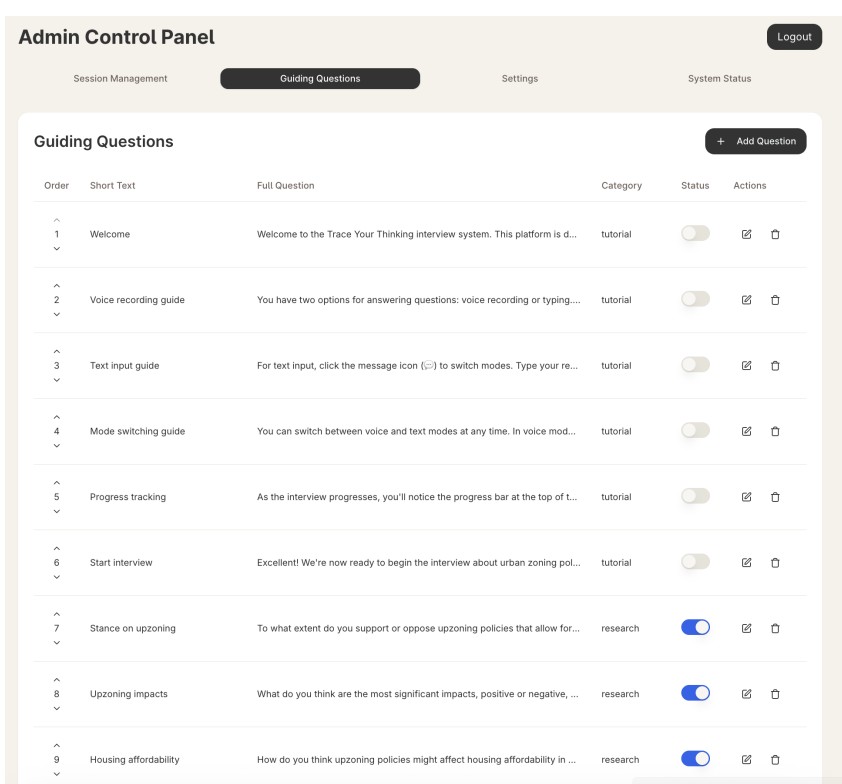

Figure 32: Guiding question editor. Researchers can toggle tutorial vs. research questions, reorder them dynamically, and experiment with alternative protocols. They can also choose to skip some questions by changing the status.

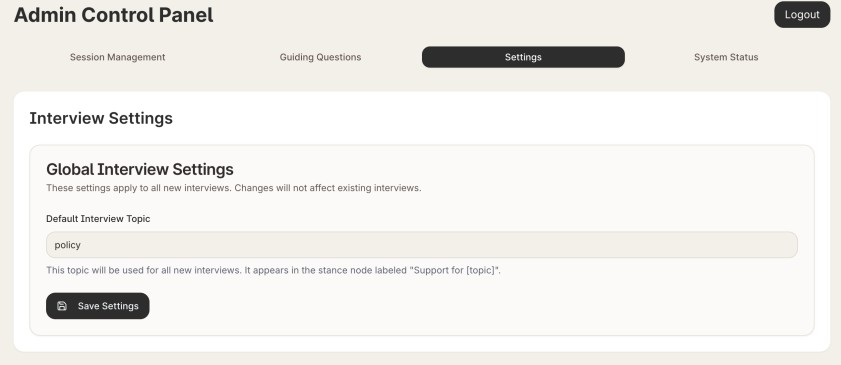

Figure 33: Global interview settings. With one change, the system can adapt to entirely new domains, enabling domain-agnostic deployment.

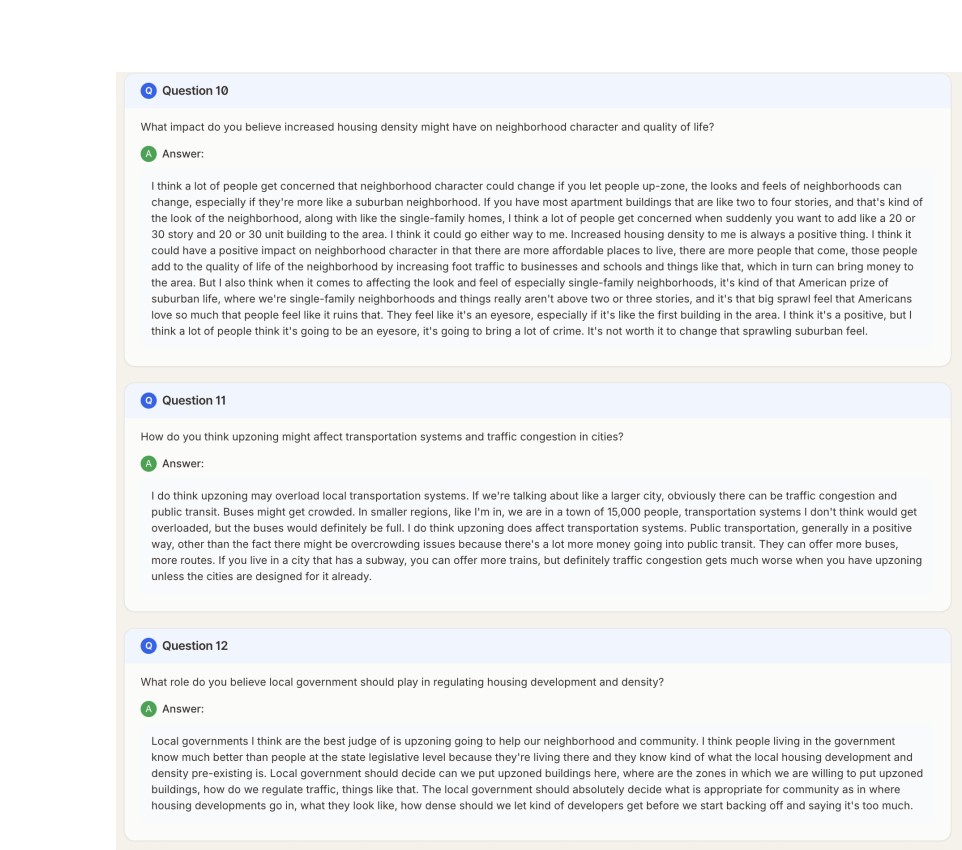

Figure 34: Sample QA transcripts highlighting variation in response depth and style. The system captures both structured argumentation and spontaneous informal commentary.

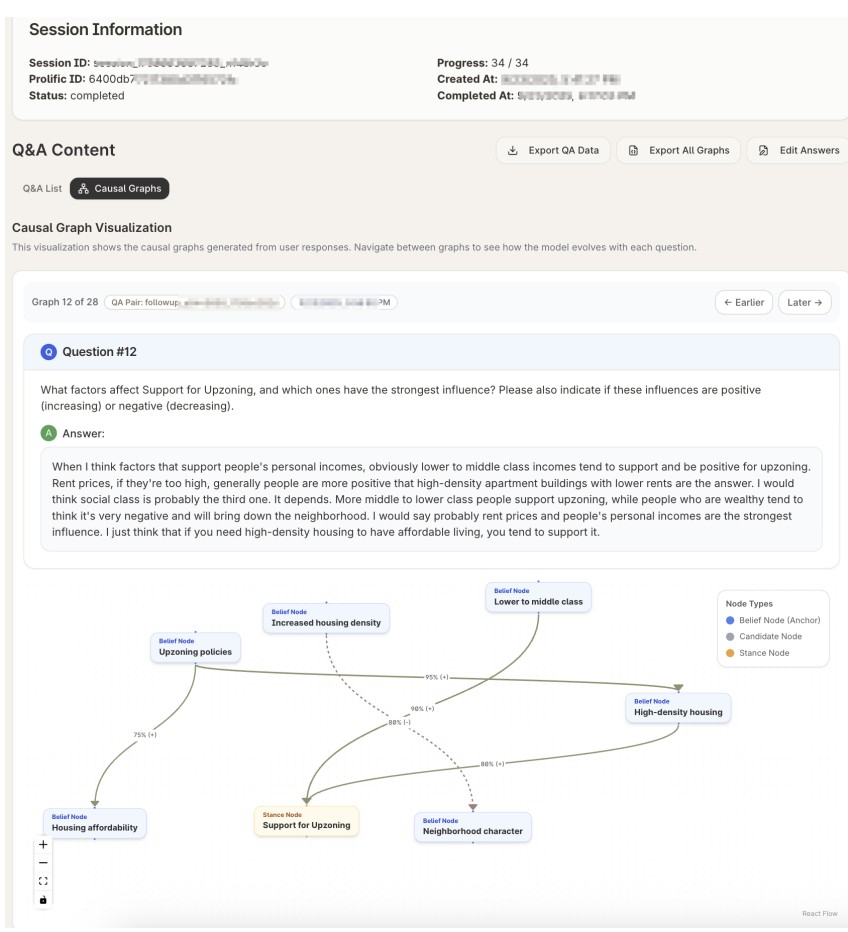

Figure 35: Dynamic causal graph visualization. Nodes capture beliefs, stances, and supporting reasons, updated continuously as the participant responds.

## S  EVALUATION

### S.1  EVALUATION METRICS

Let $y_i$ denote the ground-truth response for instance $i$, $\hat{y}_i$ the model prediction, and $N$ the total number of instances. For belief dynamics update tasks, let $y_i^{\text{prev}}$ denote the participant's pre-intervention score.

**Accuracy.**

$$\text{Acc} = \frac{1}{N} \sum_{i=1}^{N} \mathbf{1}\big[\, |\hat{y}_i - y_i| \leq \tau \,\big],$$

where $\tau$ is the tolerance band ($\tau = 1$ for 5-point scales, $\tau = 2$ for 10-point scales).

**Mean Absolute Error (MAE).**

$$\text{MAE} = \frac{1}{N} \sum_{i=1}^{N} |\hat{y}_i - y_i|.$$

**Directional Accuracy.**   We define directional accuracy as a two-stage weighted metric: (1) *Change Detection:* detecting whether a belief change occurred, and if so, (2) *Directional Inference:* correctly predicting the direction of change (*increase*, *decrease*, or *no change*). To better reflect the importance of directional reasoning, a higher weight is assigned to the second stage.

$$\text{DirAcc} = \lambda \cdot \frac{1}{N} \sum_{i=1}^{N} \mathbf{1}[(\Delta y_i = 0 \wedge \Delta \hat{y}_i = 0) \ \vee \ (\Delta y_i \neq 0 \wedge \Delta \hat{y}_i \neq 0)]$$

$$+ (1 - \lambda) \cdot \frac{1}{|\mathcal{C}|} \sum_{i \in \mathcal{C}} \mathbf{1}[\text{sgn}(\Delta y_i) = \text{sgn}(\Delta \hat{y}_i)], \tag{4}$$

where $\Delta \hat{y}_i = \hat{y}_i - \hat{y}_i^{\text{prev}}$ and $\Delta y_i = y_i - y_i^{\text{prev}}$ denote predicted and true belief changes, respectively. $\mathcal{C} = \{\, i \mid \Delta \hat{y}_i \neq 0 \wedge \Delta y_i \neq 0 \,\}$ is the set of samples where both predicted and true beliefs changed. We set $\lambda = 0.3$ by default, placing greater emphasis on directional correctness.

This weighting reflects the intuition that correctly inferring the direction of belief change is more informative than merely detecting whether a change occurred. While the first stage (*change detection*) captures a coarse perceptual judgment that can often be guessed in noisy or stable settings, the second stage(*directional inference*)reveals whether the model truly understands and reasons about belief dynamics. Hence, emphasizing the latter better reflects a model's fidelity to human-like reasoning processes and its capacity to simulate belief evolution.

**Average-to-Individual (ATI) Score.**   To provide a single comprehensive measure of model performance across both static and dynamic belief tasks, we define a unified score, *Average-to-Individual* (ATI) score, that integrates the *Belief State Inference* (BSI) and *Belief Dynamics Update* (BDU) components into a normalized value within $[0, 1]$. Specifically, $S_{\text{BSI}}$ denotes the normalized accuracy score for static belief state inference, while $S_{\text{BDU}}$ aggregates multiple metrics from the belief dynamics update task, including tolerance accuracy, normalized MAE, and directional reasoning accuracy. The unscaled ATI score is computed as:

$$\text{ATI}_{\text{unscaled}} = \tfrac{1}{2} S_{\text{BSI}} + \tfrac{1}{2}\left[\tfrac{1}{2}\big(\tfrac{1}{2} S_{\text{BDU-mae-norm}} + \tfrac{1}{2} S_{\text{BDU-acc}}\big) + \tfrac{1}{2} S_{\text{Directional-acc}}\right]. \tag{5}$$

where each subscore $S_{\cdot} \in [0, 1]$ represents a normalized evaluation metric. For MAE-based components, normalization is defined as:

$$S_{\text{BDU-mae-norm}} = \max\big(0, \ \min(1, \ 1 - \tfrac{\text{MAE}}{\text{MAE}_{\max}})\big), \tag{6}$$

where $\text{MAE}_{\max}$ denotes the task-specific upper bound of allowable error. The unified score assigns equal weights to the state component ($S_{\text{BSI}}$) and the update component ($S_{\text{BDU}}$). Within the update branch, tolerance accuracy and MAE are equally weighted (0.5 each) to ensure a balanced consideration of robustness and precision. The directional reasoning component ($S_{\text{Directional-acc}}$) is assigned an equal weight (0.5) relative to the combined MAE and accuracy branch, reflecting its comparable importance in capturing belief-updating dynamics.

To facilitate interpretation relative to human performance and baseline behavior, we linearly rescale $\text{ATI}_{\text{unscaled}}$ to a 0–100 scale:

$$\text{ATI} = \frac{\text{ATI}_{\text{unscaled}} - \text{ATI}_{\text{random}}}{\text{ATI}_{\text{human}} - \text{ATI}_{\text{random}}} \times 100, \tag{7}$$

where $\text{ATI}_{\text{random}}$ and $\text{ATI}_{\text{human}}$ denote the unscaled ATI scores of the random guess baseline and human upper bound, respectively. Under this rescaling, a score of 0 indicates random-level performance, while 100 represents human-level performance.

### S.2 COMPUTATION DETAILS AND TRACK USAGE

Unless otherwise specified, all quantitative analyses and unified average to individual (ATI) score computations are performed on the **human track**, which serves as the primary evaluation benchmark due to its ecological validity and authentic reasoning diversity.

The **synthetic track** follows the same survey and interview protocol but is designed for *auxiliary and extensibility testing*. It provides a controlled setting for examining model sensitivity, scaling behavior, and cross-domain generalization under scripted causal belief networks (CBNs). While the human track grounds evaluation in real participant reasoning, the synthetic track extends the benchmark toward scalable stress testing and potential future applications in simulated social environments.

In practice, model predictions for both tasks—*belief state inference* (BSI) and *belief dynamics update* (BDU)—are first computed independently. All metrics (accuracy, MAE, and directional accuracy) are averaged across the three domains (*healthcare*, *surveillance*, *zoning*) before aggregation into the unified score defined in Equation 5. Reported results in Section 4 and subsequent findings are therefore based on the human track unless explicitly noted otherwise.

## T    USE OF LLM USAGE

Large Language Models (LLMs) were used to aid in the writing and polishing of the manuscript.

It is important to note that the LLM was not involved in the ideation, research methodology, or experimental design. All research concepts, ideas, and analyses were developed and conducted by the authors. The contributions of the LLM were solely focused on improving the linguistic quality of the paper, with no involvement in the scientific content or data analysis.

The authors take full responsibility for the content of the manuscript, including any text generated or polished by the LLM. We have ensured that the LLM-generated text adheres to ethical guidelines and does not contribute to plagiarism or scientific misconduct.

