# OpenReview forum: "HugAgent: Evaluating LLMs in Simulating Human-Like Individual Reasoning on Open-Ended Tasks"
_ICLR.cc/2026/Conference — Submitted to ICLR 2026_

### Official Review · Reviewer_z6pz · 2025-10-27

**Soundness:** 2
**Presentation:** 1
**Contribution:** 3
**Rating:** 2
**Confidence:** 3

**Summary:**

The paper proposes a benchmark to evaluate LLMs at simulating individuals, predicting the person's belief state and belief updates. For benchmark creation, an LLM chatbot interacted with humans to infer their beliefs and updates over 8-20 QA pairs on 3 topics, where the survey also collects gold labels based on human reported stances and reasoning weights. The task for a language model is to then predict the structured belief state and updates of humans. Results show that global majority baseline leads to low performance, demonstrating individual reasoning is needed for the task. Evaluations are conducted on low-cost or old models, which with individual context perform near (best 79.12 for Qwen 2.5 32b instruct) the human baseline (80.6) for belief state inference, but are behind (best 63.38 for Llama 3.3 70b) for belief updates (human = 79.70). The paper includes a preliminary discussion of many possible design choices and details for the task of simulating individual reasoning.

**Strengths:**

1. The paper studies an interesting problem of LLMs simulating individual reasoning as beliefs and belief updates.

2. The paper conducts a human study to collect data, and also collects human performance estimates by computing test-retest agreement.

3. The paper shows evidence that models find it harder to simulate human belief updates, in comparison to predicting initial human beliefs given the context of their conversation.

**Weaknesses:**

1. The methodology is not written clearly and is hard to follow. For example, the dataset construction is almost entirely described in Appendix C. A chatbot is used to converse with humans to elicit their beliefs and belief updates, but it is unclear how this chatbot is created, what these conversations look like, and how effective this methodology is. Then in section 4.2, the term "attribution" is introduced for the first time as a metric being evaluated, even though section 3.5 on evaluation protocols has no mention of it. The paper could benefit from a significant rewrite to make it clear what the hypotheses and claims are, and the methodology used to study them.

2.  It seems like the results are extremely noisy, especially for the belief inference task where most models have very similar performance ranging from 74.5-79.1. There seems to be no clear interpretation of the results even for belief updates, where for example DeepSeek R1 has really poor belief inference performance with strong belief update prediction performance. The stated reason is "over-elaboration", but it is unclear what that means, and no evidence or justification is provided for why it is the correct explanation. Moreover, GPT 5 Mini results are reported but not Gemini 2.5 pro, GPT 5 High thinking, Claude etc. which makes it unclear what the state of the art with modern LLMs is.

3. In the section on "main findings", the findings seem quite preliminary with not enough convincing evidence. For example in "more context doesnt always help", its unclear whether: does the dialogue provided as context already leak the prediction task answers? the claim that performance peaks at 5-10 questions before declining does not hold true for many datapoints in the tables eg Qwen 2.5 32B. For finding 2, I would be interested to konw what the human attribution accuracy generalization is, as I find it unclear why personalization transfer is expected from a single conversation about a person's beliefs about "zoning" to "healthcare".

These are just examples of my broader concern: at the end of reviewing this paper, I have little clear takeaways, and much confusion. To improve the presentation in future versions, I suggest a) reporting error bars b) reporting results as bar graphs instead of tables for clearer comparison c) clearly stating hypotheses, why the specific test used to test them is valuable, and what it shows d) giving details on benchmark construction early, and more analysis of benchmark quality and finally more discussion on prior work like [1] that studies individual simulations could help position the contribution.

[1] Generative Agent Simulations of 1,000 People
Joon Sung Park, Carolyn Q. Zou, Aaron Shaw, Benjamin Mako Hill, Carrie Cai, Meredith Ringel Morris, Robb Willer, Percy Liang, Michael S. Bernstein

**Questions:**

1. How good is the benchmark data constructed using a chatbot that elicits human beliefs and belief updates?

2. What is the alignment between the synthetic agent and human study? Why were only "50 agents" (where I believe agent means causal bayesian network?) used for the synthetic track if the motivation was scaling (the human study already has 36 participants). Can synthetic agents really be used to reliably scale the data? How ecologically valid are they?

---

> ### Author Response · Authors · 2025-12-04
> **Paper Overview for Reviewer z6pz**
>
> We appreciate the reviewer’s concern about clarity and have reorganized part of the paper to make the narrative more logical and easier to follow. Below, we summarize the revised structure and the reasoning flow we aim to clarify in the updated version.
>
> Our paper examines whether modern LLMs can move beyond population-level patterns and **simulate the reasoning of a specific individual**. To evaluate this capacity, we build **HugAgent**, a benchmark that pairs **structured belief measurements** with **rich reasoning traces**.
>
> Section 2 defines the **problem formalization** and outlines our **hypotheses**. Section 3 then describes our **two-stage human data-collection pipeline**: participants first complete a questionnaire that provides explicit labels (baseline stances, reason weights, counterfactual updates), followed by a semi-structured chatbot interview that elicits their values and decision heuristics. Together, these stages yield the supervised targets and conversational contexts for the **Belief State Inference** and **Belief Dynamics Update** tasks.
>
> Sections 5–8 (Findings, Diagnostics, Error Analysis, Mitigation) are now presented as a **single coherent sequence**. Section 5 reports the core empirical findings: **identity transfer across domains is substantially harder for models than for humans**, and **additional context does not always improve belief-update prediction**. Section 6 diagnoses the cause: model performance collapses under **cross-person swaps**, indicating reliance on associative matching rather than coherent identity modeling. Section 7 analyzes the resulting **error mechanisms**, particularly failures in change detection and overreliance on domain cues. Section 8 synthesizes these insights into **mitigation strategies**, including identity-conditioned representations, stronger change-detection signals, and context filtering.
>
> This revised structure forms a logical progression: **problem framing**, **data collection and task construction**, **key findings**, **diagnostic causes**, **error mechanisms**, and **mitigation strategies**. We hope this reorganization clarifies the contribution pathway and improves the readability of the paper.

---

> > ### Author Response · Authors · 2025-12-04
> > **Response to Reviewer z6pz (5/5)**
> >
> > We appreciate the reviewer’s concern regarding the alignment between the synthetic agent and human study. As summarized in our General Response (Synthetic Track Validation), **model performance on synthetic versus human data shows consistent alignment across 13 LLMs**, with significant Pearson correlations and positive Spearman trends. We also provide **distributional comparisons of belief-graph structural statistics**, demonstrating that synthetic agents closely match key human-grounded patterns. Taken together, these results support the **reliability and validity of the synthetic track**.
> >
> > For full details, including numerical results, correlation analyses, and the construction pipeline, please refer to the General Response and **Appendix P**.

---

> ### Author Response · Authors · 2025-12-04
> **Response to Reviewer z6pz (1/5)**
>
> We apologize if the methodology was difficult to follow. While Reviewers pX6y and VpnV rated the presentation as clear (Presentation Score: 3), we acknowledge the reviewer’s concern that relying on Appendix C disrupted the reading flow.
>
> To address this, we have **rewritten Section 3** to explicitly narrate the full data-collection and modeling pipeline that is visually summarized in Figure 3.
>
> Chatbot Creation (How it works):
> We moved the description of the **Factor-Graph Elicitation Module** (Figure 3A) into the main text. We clarify that the chatbot uses a **Factor Graph** and **Predefined Probes** to ensure structured and reproducible reasoning elicitation. We have also **released the full source code** to ensure complete reproducibility.
>
> Conversation Samples (What it looks like):
> To prioritize space for analysis, we retained the full annotated transcripts in Appendix M and the complete dataset on GitHub. In the revision, we explicitly guide readers to **Figure 3 callout boxes** for immediate visual examples of the dialogue flow.
>
> Effectiveness (Validation):
> While conducting a separate methodological validation study is out of scope for this paper, we provide evidence of methodological effectiveness via **rigorous quality control**. All 54 participants’ dialogues underwent expert review, confirming that the pipeline consistently elicited **coherent, structured reasoning traces** that adequately ground the belief labels.
>
> "Attribution" Metric Definition Fixed:
> We also apologize for the terminology confusion. In the revised version, we have standardized the term “attribution” under the formal definition of **Belief State Inference (BSI)**. This metric is now **explicitly defined in Section 3.5 (Evaluation Protocols)** before any results are presented.

---

> ### Author Response · Authors · 2025-12-04
> **Response to Reviewer z6pz (2/5)**
>
> We thank the reviewer for their detailed feedback.
> Regarding the results, we have added the unified Average-to-Individual (ATI) score in the rebuttal version, with a full definition and equation provided in Appendix Q.1. This metric aggregates performance across both tasks, making model comparison more comparable and highlighting performance gaps more clearly. Additionally, all models are now evaluated over five runs, and we report standard deviations to demonstrate the consistency and reliability of our benchmark results. We also improved the visualization by grouping models into four categories: OpenAI Models, Other Closed-Source Models, Open-Source Models, and Memory-Augmented Baselines.
> Regarding the DeepSeek-R1 performance, we sincerely apologize for the confusion caused by the earlier mistake. We have corrected the results in the rebuttal version: DeepSeek-R1 achieves 77.57 (instead of 40.16) on the state inference task and 64.88 (instead of 42.25) on the update task.
> Regarding the inclusion of state-of-the-art models, we have expanded our evaluation to include Gemini 2.5 Pro, GPT-5.1 Reasoning (High), and Claude Sonnet 4.5, each run across five seeds.
> Please refer to the updated Table 2 and  Appendix C (Full Results)  in the rebuttal version for the corrected and expanded results.

---

> ### Author Response · Authors · 2025-12-04
> **Response to Reviewer z6pz (3/5)**
>
> We thank the reviewer for raising this important point. As noted in our general response, the primary contribution of this work is to highlight the shift from modeling average behavior to modeling individual reasoning, and to introduce the first dataset, open-source data-collection pipeline, and task formulation that enable individualized evaluation.
> While deeper analyses of specific findings would certainly be valuable, a comprehensive investigation of all emerging patterns is beyond the scope of this work.The analyses we do provide, including multi-run evaluation, cross-domain transfer, and context-length comparisons, already support the core claims and reveal substantive insights that motivate future research.
> Regarding the concern that the context may leak the prediction task answers, we carefully audited the benchmark to address this possibility. Two graduate level experts independently reviewed all dialogue contexts and question pairs and confirmed that there is no answer leakage. In other words, the context does not directly provide the information required to answer the prediction questions.
> Regarding the question of why personalization transfer is expected from a single conversation about a person’s beliefs, As noted in our general response we provide new evidence in the rebuttal version. Specifically, we report cross-domain transfer results for four individuals. For each person, we evaluate GPT-4o over five runs and observe a consistent pattern: in-domain accuracy > cross-domain accuracy > no-context accuracy. This pattern offers a statistically grounded indication that cross-domain value transfer is both reasonable and expected.

---

> ### Author Response · Authors · 2025-12-04
> **Response to Reviewer z6pz (4/5)**
>
> We thank the reviewer for raising this concern. Our data-collection approach follows the methodology used in the prior work **Generative Agents: Simulating 1000 People**, which likewise employed an LLM-driven chatbot to elicit individual-level information. We adopted a similar interaction framework because it enables **efficient collection of high-quality reasoning data** while maintaining **consistency across participants**.
>
> To ensure structured depth, we designed the chatbot to intentionally solicit in-depth content using a module that maintains a **simple factor graph** to track which factors require follow-up clarification and selects from a **closed set of targeted probes**. This design ensures **consistent coverage** across participants while still allowing **open-ended reasoning** at each turn. Appendix C includes the full prompt, templates, and a complete 8-turn example interaction.
>
> Each participant completed approximately **two hours of chatbot-based interviewing** and a **thirty-minute survey**, and all collected data underwent **expert review and quality control** to ensure reliability.
>
> Beyond the use of the chatbot itself, our contribution includes releasing the **entire elicitation tool as open source**, enabling researchers to adapt the pipeline to other domains or reasoning tasks. In contrast, the original Generative Agents work did not release their data-collection interface, and our open-source framework fills this **methodological gap** for the community.

---

### Official Review · Reviewer_VpnV · 2025-10-30

**Soundness:** 3
**Presentation:** 3
**Contribution:** 3
**Rating:** 6
**Confidence:** 3

**Summary:**

This paper addresses the challenge of evaluating large language models (LLMs) in simulating human-like individual reasoning, moving beyond population-level consensus. The authors introduce HugAgent, a benchmark for "average-to-individual reasoning adaptation," which requires predicting a specific person's belief states and reasoning trajectories given partial evidence of their past views. HugAgent employs a dual-track design: a synthetic track for scalable, controlled stress tests and a human track for ecologically valid "out-loud" reasoning data. The benchmark formalizes two core tasks, i.e., Belief-State Inference and Belief Dynamics Update, and evaluates state-of-the-art LLMs, revealing persistent adaptation gaps. Contributions include a formalized task definition, baseline results, error analyses, and open-source release of the benchmark and data-collection chatbot.

**Strengths:**

1. The motivation is strong and interesting. The paper formalizes "average-to-individual reasoning adaptation" as a measurable task, addressing a critical gap in LLM evaluation.

2. The use of first-person, out-loud self-reports as ground truth enhances ecological validity compared to static survey responses and the dual-track design allows for both controlled stress tests (synthetic) and real-world validation (human), which supports robust evaluation.

3. Experiments cover multiple state-of-the-art LLMs. Providing broad baseline results and comprehensive ablation studies further reveal some meaningful findings toward cross-domain generalization, context-length, and so on.

4. Detailed pipeline, including benchmark data collection, annotation, and chatbot code, is well illustrated and released as open source, promoting community adoption and extension.

**Weaknesses:**

1. The human track includes only 36 participants after quality control, which may limit statistical power and generalizability. While the authors mention it in limitations, my primary concern is that are these participants sufficient to represent the thinking ways of people with different characteristics in the real society? To ensure diversity, one way is to increase the sample size. Otherwise, you can report more detailed information about these 36 participants (e.g., their occupations and characteristics) to show that the sample size is enough for diversity consideration.

2. In Sec 4.2, the authors mention that `` ﻿﻿Tables 2 summarize performance. … Open-source LLaMA and Qwen rival GPT-4o’’, but there is no GPT-4o results in Table 2.

3. While the authors observe many findings through ablation studies, they do not analyze deeply for each finding and many of these points were glossed over without in-depth analysis. I think some ideas can be further developed.

4. The update operator U in Eq. (2) is introduced without justification in Line 121.

**Questions:**

See Weaknesses

---

> ### Author Response · Authors · 2025-12-04
> **Response to Reviewer VpnV (1/4)**
>
> We appreciate the reviewer’s suggestion and concern. As noted in our general response (Sample Size and Participant Diversity), we increased the human sample from 36 to 54 participants, the maximum feasible under our budget constraints, and expanded the synthetic agent cohort from 50 to 500. While **54 participants is a moderate sample size**, it is consistent with prior work [1,2,3,4]. Importantly, our goal is **in-depth individualized reasoning rather than large-scale population sampling**. Each participant contributed **over two hours of chatbot-based cognitive interviewing plus a structured survey**, yielding rich data suitable for modeling **individual-level reasoning patterns**.
>
> Despite the sample size, our statistical analyses exhibit **clear and consistent effects across participants**, indicating that the findings are robust for our intended purpose. We provide further justification and demographic summaries in the shared general response and in **Appendix C (Human Demographic Breakdown)**.
>
> References:
> [1] Jeroen M. van Baar, Luke J. Chang, Alan G. Sanfey. The computational and neural substrates of moral strategies in social decision-making. Nature Communications 2019.
>
> [2] Karthik Valmeekam, Matthew Marquez, Sarath Sreedharan, Subbarao Kambhampati. On the planning abilities of large language models—a critical investigation. NeurIPS 2023.
>
> [3] Aikaterina Manoli, Frank Van Overwalle, Charlotte Grosse Wiesmann, Sofie L. Valk. Functional recruitment and connectivity of the cerebellum is associated with the emergence of Theory of Mind in early childhood. Nature Communications 2025.
>
> [4] Qiujie Xie, Qiming Feng, Tianqi Zhang, Qingqiu Li, Linyi Yang, Yuejie Zhang, Rui Feng, Liang He, Shang Gao, Yue Zhang. Human Simulacra: Benchmarking the Personification of Large Language Models. ICLR 2025.

---

> ### Author Response · Authors · 2025-12-04
> **Response to Reviewer VpnV (2/4)**
>
> Thank you for raising this point, and we apologize for the confusion in the original submission. We have corrected Table 2 in the rebuttal version. The updated table now includes results for **all 16 models (including GPT-4o)**, as well as current state-of-the-art systems such as **GPT-5.1 Reasoning High**, **Claude Sonnet 4.5**, and **Gemini 2.5 Pro**. We also improved the visualization by **grouping models into four categories**: OpenAI Models, Other Closed-Source Models, Open-Source Models, and Memory-Augmented Baselines.
>
> Please refer to the shared response and **Appendix C (Full Results)** for the complete details. All models were evaluated across **five independent runs**, and the updated results **do not change the original claim**.

---

> ### Author Response · Authors · 2025-12-04
> **Response to Reviewer VpnV (3/4)**
>
> We appreciate the reviewer’s feedback and agree that deeper analyses of each finding would be valuable. In this work, our primary contribution is to **establish the first benchmark for individualized reasoning** and to **formulate the corresponding evaluation framework**. Accordingly, we focused on constructing the task, developing the **open-source data-collection pipeline**, and presenting **statistically supported empirical findings** that already provide meaningful insights into individualized reasoning.
>
> Given the scope of this work and **page-length constraints**, we present a focused set of foundational interpretations rather than exhaustive ablations. Several findings naturally open up directions for **future detailed analyses**, and we appreciate the reviewer’s suggestions. Please refer to the shared response for an expanded discussion (**Regarding Deeper Analyses and Guiding Principles**).

---

> ### Author Response · Authors · 2025-12-04
> **Response to Reviewer VpnV (4/4)**
>
> We thank the reviewer for pointing this out. In the rebuttal version, we have added a **clear justification and explanation for the update operator in Eq. (2)** to clarify its role within the overall framework. The revised text now makes explicit how the operator formalizes individualized reasoning dynamics and why it is necessary for the evaluation setup.

---

### Official Review · Reviewer_pX6y · 2025-10-31

**Soundness:** 3
**Presentation:** 3
**Contribution:** 2
**Rating:** 4
**Confidence:** 4

**Summary:**

This paper introduces HugAgent, a benchmark for evaluating whether LLMs can simulate individual reasoning rather than just population-level consensus. The key innovation is framing "average-to-individual reasoning adaptation" as a measurable task: predicting how a specific person would reason and update their beliefs in novel scenarios given their past views.

The benchmark features a dual-track design combining synthetic agents (for controlled testing) with human participants (for ecological validity), and evaluates two core tasks 1/ Belief-State Inference (predicting current stance and reasoning), and 2/ Belief Dynamics Update (predicting how beliefs change under counterfactual evidence). Experiments with 9 state-of-the-art LLMs reveal systematic failure modes, with models achieving 74-79% accuracy on belief-state inference but struggling more with belief dynamics.

**Strengths:**

The shift from simulating "the average" to "the individual" is critical for applications like digital twins, personalized AI assistants, and social simulation. This is well-motivated.

The paper goes beyond accuracy to include error analysis, cross-domain transfer tests, context ablations, and systematic failure mode analysis (directional bias, domain sensitivity, tail-driven errors).


Promising to release the full pipeline, including the chatbot, is valuable for the community.


The formalization using belief states, Bayesian updating, and structural causal models (Section 2) provides principled anchors. The four guiding hypotheses (H1-H4) make the evaluation partially interpretable

**Weaknesses:**

1. Missing important related work: The paper cites Agent Bank (Park et al., 2024) and discusses ToM benchmarks briefly, unless I miss it -several highly relevant recent works should be discussed more thoroughly:

PersonalLLM (ICLR 2025) - directly addresses personalization with heterogeneous preferences, very similar to HugAgent's goals

UniToMBench (2025) - unified ToM benchmark with multi-interaction tasks and evolving scenarios


2. 36 humans is quite limited for establishing generalizable conclusions about individual reasoning. The paper acknowledges this but doesn't adequately address:

 - Demographic diversity concerns

- Whether 36 participants provide sufficient coverage of reasoning patterns

- Statistical power for cross-domain transfer claims


3. I feel there is a theoretical grounding overclaim. Section 2.2 invokes Bayesian updating, PLoT, and SCMs as "anchors". But: 1) These aren't actually used in the evaluation or analysis, 2) The connection between theory and practice is unclear, and 3) H2 (cross-domain transfer) references graph similarity but admits it's "left as future work"

4. Table 2 shows that DeepSeek-R1 performs poorly (40.16% on belief-state inference, 42.25% on update). The explanation ("over-elaboration diverges from human reasoning") is speculative and not empirically validated.

**Questions:**

For the claim that "more context doesn't always help" is interesting but needs deeper analysis,
- Why does update accuracy peak at 5-10 questions, then decline?
- Is this cognitive overload or increasing noise?


Section 7 proposes "guiding principles" but admits these are "left as future work." Did you test any of these principles? If so, what were the results?

The claim is that synthetic agents "approximate the structural statistics of human belief graphs." Can you provide evidence in the main paper that synthetic agents exhibit similar error patterns to humans when evaluated by LLMs?

---

> ### Author Response · Authors · 2025-12-04
> **Response to Reviewer pX6y (1/7)**
>
> We thank Reviewer pX6y for their feedback and for recognizing the important impact of the question(average to individual). We proposed our work and the usefulness of our open-source data collection pipeline. In addition to the general response, we’d like to respond below to the reviewer’s main concerns:
>
> > **Weakness 1:** Missing important related work: The paper cites Agent Bank (Park et al., 2024) and discusses ToM benchmarks briefly, unless I miss it -several highly relevant recent works should be discussed more thoroughly:
> PersonalLLM (ICLR 2025) - directly addresses personalization with heterogeneous preferences, very similar to HugAgent's goals and UniToMBench (2025) - unified ToM benchmark with multi-interaction tasks and evolving scenarios
>
> We thank the reviewer for pointing us to these relevant works. We appreciate the suggestion, and we have added the missing related work to the rebuttal version accordingly. These references are now discussed in the **Related Work** section.
>
> Regarding **PersonaLLM**, the only overlap with our work lies in the general goal of building individualized LLMs. However, the objectives differ substantially. PersonaLLM focuses on **aligning models with users’ preferences**, whereas our work aims to **model why and how users make decisions**, emphasizing **underlying reasoning processes** rather than preference alignment.
>
> For **UniToMBench**, we acknowledge its contributions to expanding ToM evaluation through multi-interaction tasks, evolving narratives, and improved prompting strategies. However, these benchmarks still rely on **curated synthetic scenarios** and do not incorporate **real-world tasks**, **real human behavioral data**, or **individualized cognitive modeling**. While our work shares the high-level goal of assessing LLMs’ ability to understand and reason about human mental states, our contribution lies in **capturing real-world cognitive variability** and **individualized decision-making processes**, rather than relying solely on synthetic, scenario-based ToM evaluations.

---

> ### Author Response · Authors · 2025-12-04
> **Response to Reviewer pX6y (2/7)**
>
> > **Weakness 2:** 36 humans is quite limited for establishing generalizable conclusions about individual reasoning. The paper acknowledges this but doesn't adequately address: 1. Demographic diversity concerns, 2. Whether 36 participants provide sufficient coverage of reasoning patterns, 3. Statistical power for cross-domain transfer claims
>
> We appreciate the reviewer’s concern regarding sample size. As noted in our general response (Sample Size and Participant Diversity), we increased the human sample from 36 to **54 participants**, the maximum feasible under our budget constraints, and expanded the synthetic agent cohort from 50 to **500**. While **54 is a moderate sample size**, it is consistent with prior work [1,2,3,4]. Despite this limitation, our statistical analyses reveal **clear and consistent patterns** across participants, as discussed in Sections 5–7. Please refer to the shared response and **Appendix C (Human Demographic Breakdown)** for detailed justification and participant demographics.
>
> References:
> [1] Jeroen M. van Baar, Luke J. Chang, Alan G. Sanfey. The computational and neural substrates of moral strategies in social decision-making. Nature Communications 2019.
> [2] Karthik Valmeekam, Matthew Marquez, Sarath Sreedharan, Subbarao Kambhampati. On the planning abilities of large language models—a critical investigation. NeurIPS 2023.
> [3] Aikaterina Manoli, Frank Van Overwalle, Charlotte Grosse Wiesmann, Sofie L. Valk. Functional recruitment and connectivity of the cerebellum is associated with the emergence of Theory of Mind in early childhood. Nature Communications 2025.
> [4] Qiujie Xie, Qiming Feng, Tianqi Zhang, Qingqiu Li, Linyi Yang, Yuejie Zhang, Rui Feng, Liang He, Shang Gao, Yue Zhang. Human Simulacra: Benchmarking the Personification of Large Language Models. ICLR 2025.
>
> We also appreciate the reviewer’s concern about the **statistical power of our cross-domain transfer claims**. As noted in our general response, we conducted additional analyses using **four individuals**, each evaluated with **GPT-4o over five runs**. The results consistently show the expected pattern:
> **in-domain accuracy > cross-domain accuracy > no-context accuracy**,
> providing stronger statistical grounding for the claim. We additionally include a **qualitative example** from one participant to illustrate why such cross-domain transfer behavior is expected.

---

> ### Author Response · Authors · 2025-12-04
> **Response to Reviewer pX6y (3/7)**
>
> > **Weakness 3:** I feel there is a theoretical grounding overclaim. Section 2.2 invokes Bayesian updating, PLoT, and SCMs as "anchors". But: 1) These aren't actually used in the evaluation or analysis, 2) The connection between theory and practice is unclear, and 3) H2 (cross-domain transfer) references graph similarity but admits it's "left as future work"
>
> We thank the reviewer for this incisive observation. We acknowledge that the connection between the theoretical “anchors” and the empirical evaluation could be stated more precisely. We have revised Section 2.2 and Section 3 to clarify the exact role these theories play in our framework.
>
> SCMs and Bayesian Updating as Generative Engines:
> We clarify that these frameworks are **directly used in benchmark construction**. The **Synthetic Track (Section 3.2)** is not based on heuristic scripting; it is rigorously generated using **Structural Causal Models (SCMs)**. The ground-truth labels for belief updates are computed via **exact Bayesian inference** over these causal graphs. Thus, SCMs serve as the **mathematical ground-truth generator** against which LLM performance is evaluated.
>
> PLoT as the Evaluation Target:
> We clarified that **Probabilistic Language of Thought (PLoT)** is the **hypothesis being tested**, not the method used for construction. HugAgent evaluates whether LLMs can **approximate the reasoning fidelity** predicted by formal probabilistic theories (i.e., PLoT-like behavior) without having access to the underlying math.
>
> Refining H2 (Cross-domain):
> We agree that referencing “graph similarity” without quantification was premature. We revised **Hypothesis 2 (H2)** to focus on **Domain Structural Transfer**—whether models can transfer individualized reasoning patterns across domains with differing causal densities—and removed the unsubstantiated claim about graph similarity metrics to ensure alignment with our empirical findings.

---

> ### Author Response · Authors · 2025-12-04
> **Response to Reviewer pX6y (4/7)**
>
> > **Weakness 4:** Table 2 shows that DeepSeek-R1 performs poorly (40.16% on belief-state inference, 42.25% on update). The explanation ("over-elaboration diverges from human reasoning") is speculative and not empirically validated.
>
> We sincerely appreciate the reviewer for identifying this issue. As noted in our general response (DeepSeek-R1 Results and Expanded Evaluation with Frontier LLMs), we have corrected the error in the updated results: **DeepSeek-R1 achieves 77.57 on the state inference task** (instead of 40.16) and **64.88 on the update task** (instead of 42.25). We have also added the latest state-of-the-art models, including **GPT-5.1 Reasoning High**, **Claude Sonnet 4.5**, and **Gemini 2.5 Pro**. For full details, please refer to the shared response and **Appendix C (Full Results)**.

---

> ### Author Response · Authors · 2025-12-04
> **Response to Reviewer pX6y (5/7)**
>
> Q1:“More context doesn’t always help” requires deeper analysis
> We thank the reviewer for raising this important question. Although we did not perform targeted ablations to disentangle cognitive overload from noise, the asymmetric trend in Table 4 provides suggestive evidence consistent with a noise/distraction interpretation rather than pure overload:
> Belief State Inference continues to improve with longer histories.
>
>
> Belief Dynamics Update improves initially but then declines as context grows.
>
>
> If cognitive overload were the dominant factor, we would expect degradation across both tasks. Instead, models still reliably extract static beliefs from long contexts, indicating that overall capacity is sufficient. The decline in update accuracy is more consistent with a signal-to-noise ratio effect—i.e., as additional historical turns accumulate, the small, local update signal becomes harder for the model to isolate.
> We have added this clarifying analysis. We emphasize that this is an interpretation rather than a causal claim; dedicated ablations (e.g., controlled noise injection or context pruning) would be required for formal identification, and we outline these as future work.
> As noted in our general response, our primary objective is to introduce the first framework for individualized simulation, together with the data-collection pipeline and benchmark. These findings are initial observations intended to motivate the deeper analyses that our framework enables.

---

> ### Author Response · Authors · 2025-12-04
> **Response to Reviewer pX6y (6/7)**
>
> Q2: Guiding principles were proposed, but were they tested?
> We appreciate the reviewer’s question.
> As noted in our general response, the primary contribution of this work is introducing the individualized evaluation framework and establishing the first benchmark and task formulation. The guiding principles in Section 8 are derived from empirical observations and are intended to outline promising research directions rather than claims tested within this paper. However, we agree that some observations warrant deeper unpacking. In the revision, we have expanded the analysis of context-length scaling (Sec. 5) and provided a concrete example: we discuss the trade-off between information gain (which benefits belief-state inference) and accumulated noise that dilutes the local update signal (which hinders belief-dynamics tracking). This clarification illustrates how HugAgent can serve as a diagnostic tool for identifying individualized reasoning bottlenecks. Please refer to the shared response for an expanded discussion.

---

> ### Author Response · Authors · 2025-12-04
> **Response to Reviewer pX6y (7/7)**
>
> Q3:Evidence that synthetic agent error pattern are similar to humans.
>
> We appreciate the reviewer’s question. In the main paper, we focused our analysis on the human-track results, so the evidence for synthetic agents mirroring human error patterns was placed in the Appendix C(Results on Synthetic dataset). The synthetic track has been expanded from 50 to 500 agents, and we report results for 50 agents evaluated across five runs.
> The findings show that synthetic agents exhibit error patterns highly similar to humans. Across all 13 models, performance on the state inference task is consistently higher than on the dynamics update task, matching the human-track trend. Synthetic agents also reproduce key qualitative findings from the human results: (1) several models show peak performance at intermediate context length ( 5–10 questions), suggesting that more context does not always help; and (2) accuracy decreases substantially in cross-domain, cross-person, and no-context conditions, paralleling the degradation observed with human data.
> These results support our claim that synthetic agents approximate the structural statistics and error dynamics found in human belief graphs, and we highlight this more explicitly in the updated Appendix C (Full Results) for the full details.

---

### Official Review · Reviewer_imYS · 2025-11-01

**Soundness:** 3
**Presentation:** 2
**Contribution:** 3
**Rating:** 6
**Confidence:** 2

**Summary:**

The paper proposes HugAgent, a dataset for evaluating LLM agents' capabilities on simulating human-like reasoning. It features a human track for ecological validity and a synthetic track for scalable stress tests. Results reveal persistent adaptation gaps: top LLMs trail human retest ceilings, with larger deficits on belief dynamics.

**Strengths:**

- Novel problem formulation assessing LLMs’ capability to simulate *individual* human reasoning.
- Results provide interesting insights into LLM failure modes
- Open-source pipeline (chatbot, data, evaluation code) supports reproducibility and extension.

**Weaknesses:**

- Confusing use of “open-ended”: in LLM evaluation, this often implies systems auto-generating novel tasks; here it seems to refer to diverse human reasoning traces in curated domains—clarify terminology to avoid misinterpretation.
- Scalar score calibration: predicting 1–10 stance or 1–5 reason weights is known to be noisy in LLMs; no analysis of calibration error or distribution shift vs. human ratings.
- Synthetic track lacks human validation: 50 scripted agents use deterministic causal graphs. Is there a study that confirms humans perceive these as plausible or follow similar update paths?

**Questions:**

see above weakness

---

> ### Author Response · Authors · 2025-12-04
> **Response to Reviewer imYS (1/3)**
>
> We’d like to thank Reviewer imYS for their positive and encouraging feedback. We appreciate the reviewer recognizing the motivation of our work and viewing our contribution as good. In addition to the general response, we’d like to respond below to the reviewer’s main concerns:
>
> > **Weakness 1:** Confusing use of “open-ended”: in LLM evaluation, this often implies systems auto-generating novel tasks; here it seems to refer to diverse human reasoning traces in curated domains—clarify terminology to avoid misinterpretation.
>
> We thank the reviewer for highlighting this terminological ambiguity. We are fully aware that in RL and ALife communities, *open-endedness* often implies systems that auto-generate novel tasks (e.g., open-ended evolution).
>
> **Our usage is grounded in Cognitive Science and Psychometrics**, where *open-ended* describes the nature of the solution space rather than the task format.
>
> **Cognitive Source.** Unlike traditional vignettes constrained by closed-world assumptions, our data derive from unconstrained human think-aloud protocols [1]. These capture genuine reasoning traces where the original solution space was fully open and divergent [2,3].
>
> **Evaluation Design.** While we formalize these rich traces into structured formats (e.g., MCQs) to ensure rigor and scalable measurability, the content itself is anchored in the open-ended nature of human cognition. HugAgent bridges this gap: retaining the cognitive depth of open-ended reasoning while providing the rigor of structured benchmarks.
>
> To prevent confusion, we have added a footnote upon the first mention of *open-ended* in the revision. This footnote explicitly distinguishes our usage (data provenance and cognitive validity) from the *task-generation* definition common in RL literature.
>
> **Reference:**
> [1] Ericsson, K. Anders. "Protocol analysis." A companion to cognitive science (2017): 425-432.
>
> [2] Martinez, M. E. *Cognition and the question of test item format.* Educational Psychologist, 34(4), 207–218, 1999.
>
> [3] Joy Paul Guilford. *The Nature of Human Intelligence.* MacGraw–Hill, 1967.

---

> ### Author Response · Authors · 2025-12-04
> **Response to Reviewer imYS (2/3)**
>
> > **Weakness 2:** Scalar score calibration: predicting 1–10 stance or 1–5 reason weights is known to be noisy in LLMs; no analysis of calibration error or distribution shift vs. human ratings.
>
> We agree with the reviewer that **numerical prediction can be noisy in LLMs**. Following the setup of *Generative Agents: Simulating 1000 People*, we **minimized stochasticity** by using **temperature 0.1**, a **fixed random seed**, and **function calling for deterministic formatting**[1]. To assess stability, we ran all experiments **five times per model** under the same protocol. As shown in Appendix G, accuracies on both state inference and update tasks exhibit **very low variance across runs (std < 1)**, indicating that our scalar outputs are stable.
>
> We also compared **model-generated scalar distributions** to **human distributions** across tasks and scales. Across most models, **JSD values fall between 0.02 and 0.08**, a range typically interpreted as indicating **high distributional similarity**. **Pearson correlations** between model and human mean scores fall between **0.2 and 0.4**, reflecting a **modest but reliable alignment** in central tendency. For example, in the healthcare domain (1–5 scale), multiple models (GPT-5-mini, Gemini, Qwen-max) achieve **JSD < 0.03 with r > 0.35**, meaning that both the shape of the distribution and the average ratings closely follow human judgments. While certain tasks (e.g., zoning, 1–10 scale) show small deviations, **KS tests provide no evidence of statistically significant divergence**.
>
> These results together show that **scalar responses are stable, well-aligned with human patterns, and robust** to the level of noise typically observed in LLM numerical outputs. Any item-level variability does not affect our overall findings. Additional details are provided in **Appendix G (Scalar Sensitivity Analysis)**.
>
> Reference:
> [1] Joon Sung Park, Carolyn Q. Zou, Aaron Shaw, Benjamin Mako Hill, Carrie Cai, Meredith Ringel Morris, Robb Willer, Percy Liang, Michael S. Bernstein. *Generative agent simulations of 1,000 people.* arXiv 2024.

---

> ### Author Response · Authors · 2025-12-04
> **Response to Reviewer imYS (3/3)**
>
> > **Weakness 3:** Synthetic track lacks human validation: 50 scripted agents use deterministic causal graphs. Is there a study that confirms humans perceive these as plausible or follow similar update paths?
>
> We appreciate the reviewer’s concern regarding the **human validation of synthetic agents**. As summarized in our **General Response (Synthetic Track Validation)**, model performance on synthetic versus human data shows **consistent alignment across 13 LLMs**, with **significant Pearson correlations** and **positive Spearman trends**. We also provide **distributional comparisons of belief-graph structural statistics**, showing that synthetic agents closely match key human-grounded patterns. These results together support the **reliability of the synthetic track**.
>
> For full details, including numerical results, correlation analyses, and the construction pipeline, please refer to the **General Response and Appendix P** in the revised PDF.

---

### Author Response · Authors · 2025-12-04
**1 General Response to All Reviewers**

**General Response to All Reviewers**

We thank all reviewers for their detailed comments and thoughtful engagement with our work. We especially appreciate that reviewers recognized the significance of assessing LLMs’ ability to simulate individual human reasoning, as well as the potential for HugAgent to make a meaningful contribution to areas such as LLM digital twins, social simulation, and Theory of Mind research.

We would like to clarify the scope and primary contribution of this work. This paper is intended as a first step toward studying individualized reasoning in LLMs. Our central objective is to introduce the research question and to establish the first benchmark, dataset, and task formulation designed specifically to evaluate whether LLMs can simulate the reasoning patterns of a specific individual—going beyond population-level trends that dominate existing evaluation frameworks.

To evaluate this capacity, we introduce HugAgent, a benchmark that pairs structured belief measurements with rich reasoning traces collected through a controlled, open-source pipeline. Our contributions are threefold:

- A new evaluation framework that explicitly targets individualized reasoning rather than population-level averages.
- The first dataset and benchmark designed to assess whether LLMs can model a specific person’s belief states and belief updates.
- A reusable, scalable, and open-source data-collection pipeline (chatbot) that enables the community to extend the benchmark to new domains or populations with minimal effort.

In addition, our empirical results provide foundational, yet meaningful, insights into the challenges and potential of simulating individual reasoning with modern LLMs. More broadly, we believe that establishing the importance and feasibility of individualized simulation is a valuable contribution in itself, even prior to conducting deeper or fully comprehensive analyses.

Also, we note that we attempted to follow best practices by including a dataset card, open-sourcing the data and the entire data collection pipeline (chatbot), and informing users about potential concerns and responsible use. We also discuss the risks and ethical implications of our dataset release in Section 11. If there are any remaining concerns that we can address here, please let us know.

Before moving to our point-by-point responses, we’d like to address some common main points that were raised by more than one reviewer:

**Synthetic Track Validation**

We thank the reviewer for raising the question regarding the quality of the synthetic-agent data. We first clarify that the main findings and insights in our paper are derived from the real human dataset (n = 54). The synthetic-agent track is introduced only as a complementary and scalable extension, enabling controlled stress-testing and larger-scale evaluation possible. Our goal is not to replace human data, but to explore whether synthetic agents can serve as a practical proxy when scaling is required.

To assess the quality of the synthetic agents, we compared model performance ordering on synthetic versus human data across 13 LLMs. For BSI (Accuracy), Pearson shows a significant correlation (r = 0.6466, p = 0.0169), while Spearman shows a consistent positive trend (r = 0.2944, p = 0.3289). For BDU, correlations are consistently positive and statistically significant:

Accuracy: Pearson r = 0.7499 (p = 0.0032), Spearman r = 0.5604 (p = 0.0463)
MAE: Pearson r = 0.6773 (p = 0.0110), Spearman r = 0.6171 (p = 0.0246)

We report both Pearson and Spearman because they capture complementary aspects of ordering similarity: Pearson measures linear score correspondence, and Spearman measures monotonic ranking agreement. Together, the results show stable alignment in model rankings between synthetic and human data.

Beyond performance correlations, we also examined distributional similarity in the structural statistics of belief graphs. The results show that synthetic-agent graphs reproduce key human-grounded patterns across structural dimensions such as node and edge counts, edge density, confidence levels. These aligned distributions indicate that synthetic agents not only match model-level performance ordering but also approximate the structural properties of human belief graphs, supporting their use as controlled stress-test agents.

Please refer to the detailed pipeline for constructing synthetic agents in the Appendix P, along with the accompanying figures comparing the distributional statistics of human-grounded and synthetic belief graphs.

Taken together, both the ordering consistency and the structural distributional alignment suggest that our synthetic agents provide a high-quality and scalable complementary track. While human data remain central to our study, the synthetic agents offer a reliable extension that preserves key behavioral and structural properties relevant for benchmarking belief reasoning.

---

### Author Response · Authors · 2025-12-04
**2 General Response to All Reviewers**

**Sample Size and Participant Diversity**

We appreciate the reviewer’s concern and agree that the original sample size was limited. In response, we expanded the human sample from 36 to 54 participants, which is the maximum feasible within our budget constraints. We also expanded the synthetic agent track from 50 to 500 samples and reported updated results across 13 models evaluated over five runs in the rebuttal version. Because participant compensation has already exceeded 8,35071 USD, 54 participants represent the maximum number we were able to collect within our budget.

While several published studies [1,2,3,4] in this area use similarly sized samples, we acknowledge that 54 remains relatively moderate. Importantly, our goal in this work is to establish the first benchmark focused on simulating individual reasoning instead of estimating population-level coverage. This work is the first to introduce the research question, dataset, and evaluation framework needed to enable such individualized analysis. Our interview protocol was intentionally designed to be in-depth. Each participant completed more than two hours of chatbot-based interviewing and a 30-minute survey across three topics, yielding rich and detailed behavioral data at the individual level. For the purpose of modeling individualized reasoning patterns, the depth and quality of each participant’s data provide substantial value even with a moderate sample size.

While we acknowledge that a larger sample size would be preferable, our statistical analyses nonetheless show that even state-of-the-art models struggle with our task of simulating individualized reasoning, indicating that the observed patterns are robust despite the limited statistical power. A larger cohort was unfortunately not feasible under our budget constraints.

Following the reviewer’s suggestion, we have added detailed demographic information for all 54 participants in the appendix, including distributions of age, occupation, income, and education level.
All participants were recruited through Prolific, restricted to U.S.-based users, and compensated for their time. The study did not impose any race or income thresholds during recruitment, ensuring a naturally diverse participant pool.

The sample reflects substantial diversity across all attributes:

**Age.** Participants span a wide age range:
- 18–24 (7.4%)
- 25–34 (27.8%)
- 35–44 (29.6%)
- 45–54 (27.8%)
- 55–64 (5.6%)
- 65+ (1.9%)

**Education.** The sample includes:
- High school graduates (11.1%)
- Individuals with some college (18.5%)
- Associate’s degrees (13.0%)
- Bachelor’s degrees (40.7%)
- Graduate/professional degrees (16.7%)

**Income.** Participants are distributed across income levels:
- <$25k (9.3%)
- $25k–$49k (22.2%)
- $50k–$74k (22.2%)
- $75k–$124k (22.2%)
- $125k–$199k (22.2%)
- ≥$200k (1.9%)

**Occupation.** Respondents represent diverse employment categories:
- Management/business/science/arts (46.3%)
- Service occupations (31.5%)
- Sales/office (13.0%)
- Construction/maintenance (7.4%)
- Production/transportation (1.9%)

**Race.** Racial composition includes:
- White (55.6%)
- Black (16.7%)
- Asian (13.0%)
- Middle Eastern/North African (3.7%)
- Multiracial participants (11.1%)

Please refer to Appendix C (Human Demographic Breakdown) for the full details.

**References:**
[1] Jeroen M. van Baar, Luke J. Chang, Alan G. Sanfey. *The computational and neural substrates of moral strategies in social decision-making.* Nature Communications 2019.

[2] Karthik Valmeekam, Matthew Marquez, Sarath Sreedharan, Subbarao Kambhampati. *On the planning abilities of large language models—a critical investigation.* NeurIPS 2023.

[3] Aikaterina Manoli, Frank Van Overwalle, Charlotte Grosse Wiesmann, Sofie L. Valk. *Functional recruitment and connectivity of the cerebellum is associated with the emergence of Theory of Mind in early childhood.* Nature Communications 2025.

[4] Qiujie Xie, Qiming Feng, Tianqi Zhang, Qingqiu Li, Linyi Yang, Yuejie Zhang, Rui Feng, Liang He, Shang Gao, Yue Zhang. *Human Simulacra: Benchmarking the Personification of Large Language Models.* ICLR 2025.

---

### Author Response · Authors · 2025-12-04
**3 General Response to All Reviewers**

**Deepseek-R1 Results and Expanded Evaluation with Frontier LLMs**
We thank the reviewers for raising this point on the results presentation and apologize for the confusion in the original submission. We have updated Table 2 in the rebuttal version.

Regarding the DeepSeek-R1 performance, we sincerely apologize for the earlier issues. The updated results are now included in the rebuttal version: **DeepSeek-R1 achieves 77.57 (instead of 40.16) on the state inference task and 64.88 (instead of 42.25) on the update task.**

The issue occurred because DeepSeek-R1 is a reasoning-style model, and our initial extraction inadvertently captured intermediate reasoning text rather than the final answer. To address this, **we now use function calling to enforce precise and structured outputs**, and we have updated the entire pipeline so that all models rely on the same extraction mechanism. We also reran all experiments five times to verify stability and now report consistent results. In total, benchmarking all models incurred approximately **$1,430** in computational cost.

All models are evaluated across five runs, and we report standard deviations to demonstrate the reliability and consistency of our benchmark.

Furthermore, to make the evaluation clearer and more comparable, **we have added the unified Average-to-Individual (ATI) score**, along with its full definition and equation in Appendix Q.1. This metric aggregates performance across both tasks, making model comparison more straightforward and highlighting performance gaps more clearly.

We have also improved the visualization of Table 2 by grouping models into four categories:
**OpenAI Models**, **Other Closed-Source Models**, **Open-Source Models**, and **Memory-Augmented Baselines**.

In addition, we expanded our evaluation to include several state-of-the-art models, such as **Gemini 2.5 Pro**, **GPT-5.1 Reasoning (High)**, and **Claude Sonnet 4.5**, each assessed over five runs.

Please refer to the updated Table 2 and Appendix C (Full Results) in the rebuttal version for the corrected and expanded results. All updated numbers were obtained under the same evaluation protocol, and they do not change the original claims of the paper.

---

**Model Results**


| Model                  | BSI Acc (%) | BDU Acc (%) | MAE  | Dir. Acc (%) | ATI (%) |
|------------------------|------------:|------------:|-----:|-------------:|--------:|
| **Human**              | **84.84**   | **85.66**   | **0.68** | **88.92**    | **100.00** |
| **OpenAI Models**      |             |             |      |              |         |
| GPT-4o                 | 74.66±0.24  | 63.11±0.14  | 1.29±0.00 | 82.27±1.02 | 67.29±0.74 |
| GPT-5-mini             | 75.30±0.79  | 58.21±0.50  | 1.43±0.01 | 77.02±2.02 | 61.53±2.09 |
| GPT-5.1 Reasoning High | 73.36±2.16  | 67.13±0.46  | **1.17±0.01** | 80.94±3.28 | 64.66±3.79 |
| o3-mini                | 75.12±0.78  | 64.54±0.32  | 1.22±0.01 | 71.29±2.87 | 60.92±1.68 |
| **Other Closed-Source Models** |   |             |      |              |         |
| Claude Sonnet 4.5      | 76.04±0.01  | **68.61±0.09** | 1.18±0.01 | 78.73±0.10 | 67.39±0.11 |
| Gemini 2.0 Flash       | 69.95±0.12  | 60.55±0.08  | 1.35±0.00 | **83.31±0.19** | 59.76±0.18 |
| Gemini 2.5 Pro         | 75.45±1.13  | 64.87±0.39  | 1.27±0.01 | 78.65±1.17 | 64.29±1.58 |
| DeepSeek-R1            | 75.55±0.34  | 64.88±0.33  | 1.29±0.01 | 79.69±0.91 | 67.20±1.03 |
| Qwen-plus              | **77.57±0.44** | 58.93±0.22 | 1.40±0.00 | 77.17±0.61 | 65.48±0.87 |
| Qwen-max               | 77.40±0.27  | 58.86±0.25  | 1.40±0.00 | 77.17±0.48 | 65.21±0.49 |
| **Open-Source Models** |            |             |      |              |         |
| LLaMA 3.3 70B          | 76.64±0.18  | 67.57±0.20  | 1.24±0.00 | 79.56±0.36 | **69.84±0.27** |
| Qwen2.5-32B-instr.     | 77.17±0.32  | 58.96±0.05  | 1.40±0.00 | 76.88±0.29 | 64.71±0.23 |
| Qwen2.5-7B-instr.      | 77.18±0.73  | 58.82±0.20  | 1.40±0.00 | 77.12±0.80 | 64.83±1.21 |
| **Memory-Augmented Baselines** |   |             |      |              |         |
| RAG                    | 75.46±0.47  | 51.90±0.39  | 1.57±0.05 | 72.25±0.57 | 54.96±0.87 |
| RAG-FC                 | 77.56±0.38  | 59.97±0.23  | 1.39±0.00 | 76.80±0.80 | 65.65±0.61 |
| Generative Agents      | 76.19±0.36  | 58.22±0.43  | 1.40±0.01 | 76.13±2.45 | 62.43±1.74 |
| **Non-Learning Baselines** |      |             |      |              |         |
| Global Majority        | 65.77±0.00  | 58.18±0.00  | 2.54±0.00 | 17.93±0.00 | 4.44±0.00 |
| Random Guess           | 51.89±3.96  | 43.12±0.78  | 1.88±0.05 | 46.74±3.28 | 0.00±5.80 |


---

---

### Author Response · Authors · 2025-12-04
**4 General Response to All Reviewers**

**Cross-Domain Transfer Claim**
Regarding the concern about “statistical power for cross-domain transfer claims, and why personalization transfer is expected from a single conversation about a person’s beliefs,” we have added additional analyses below.

We assume that cross-domain personalization is feasible because **participants express stable value-laden cues in language**. These cues are not tied to a single topic but reflect **general underlying principles** that consistently influence preferences across domains.

Specifically, we report cross-domain transfer results for four individuals. For each person, we evaluate GPT-4o across five runs and observe a **consistent pattern**:

**in-domain accuracy > cross-domain accuracy > no-context accuracy**

This ordering holds across all four individuals and all five runs, providing a more statistically grounded indication that the observed cross-domain value transfer is reasonable and expected.

---

## **Cross-domain Transfer Results (GPT-4o)**

| User (n-BSI / n-BDU) | Experiment    | BSI Acc (%) | BDU Acc (%) | MAE  | Dir. Acc (%) | ATI (%) |
|----------------------|---------------|------------:|------------:|-----:|-------------:|--------:|
| **User 1 (9 / 48)**  | In Domain     | 88.89       | 91.67       | 0.57 | 85.00        | 87.88   |
|                      | Cross Domain  | 55.56       | 54.58       | 1.38 | 50.00        | 55.28   |
|                      | No Context    | 33.33       | 8.33        | 2.33 | 18.00        | 27.42   |
| **User 2 (13 / 41)** | In Domain     | 92.31       | 91.71       | 0.58 | 65.00        | 84.57   |
|                      | Cross Domain  | 92.31       | 83.90       | 0.97 | 73.00        | 84.37   |
|                      | No Context    | 30.77       | 39.02       | 1.68 | 65.00        | 43.78   |
| **User 3 (5 / 24)**  | In Domain     | 66.67       | 64.23       | 1.26 | 57.50        | 64.29   |
|                      | Cross Domain  | 66.67       | 28.57       | 1.87 | 65.00        | 59.80   |
|                      | No Context    | 46.67       | 19.22       | 1.95 | 50.00        | 44.63   |
| **User 4 (7 / 38)**  | In Domain     | 57.14       | 42.11       | 1.72 | 92.50        | 64.10   |
|                      | Cross Domain  | 51.43       | 30.00       | 1.79 | 85.00        | 57.63   |
|                      | No Context    | 31.43       | 30.00       | 1.79 | 77.50        | 45.73   |

---

To complement these quantitative results, we also provide a qualitative example from one individual’s transcript. The snippet illustrates how the person’s explicitly stated beliefs (e.g., **stable values, moral principles, tradeoff preferences**) apply across multiple policy domains.

This example demonstrates why **grounded individualized value extracted from a single in-depth conversation (2-hour interview)** can reasonably generalize to related domains for the same individual.

---

### **Transcript Example**

**1. A safety-first orientation that generalizes across contexts.**
- *Zoning:* “If cheaper apartments were made, it’ll cause overcrowding, and that means more low-income people moving in… which will make crime rates go up!”
- *Surveillance:* “Cameras are a tool of preventing potential crime and it’s effective.”

A safety-oriented signal extracted from zoning therefore naturally predicts support for surveillance technologies.

---

**2. A stable opposition to redistribution across domains.**
- *Healthcare:* “Nobody is entitled to other people’s money. Taxpayers shouldn’t be forced to pay for other people’s medical needs.”
- *Zoning:* “Equity means taking opportunities away from some to redistribute them to others. I don’t like redistributing what a successful person has.”

The anti-redistribution stance expressed in healthcare directly predicts resistance to equity-motivated zoning reforms.

---

These dimensions—**safety orientation**, **redistribution aversion**, **negative out-group attribution**, and **selective distrust of government**—are expressed consistently across the three domains.

This illustrates why personalization can generalize:
> the model is transferring **stable value signals**, rather than topic-specific information.

Together, the quantitative seed-level consistency and the qualitative evidence offer a clearer explanation of why cross-domain personalization transfer is expected and why the effect we observe is robust.

Please refer to Appendix N (Rationale for Individual Cross-Domain Transfer) for the full details.

---

### Author Response · Authors · 2025-12-04
**5 General Response to All Reviewers**

**Regarding Deeper Analyses and Guiding Principles**

We appreciate the reviewer’s feedback and agree that deeper analysis of each finding and further empirical validation of the guiding principles would indeed be valuable. However, conducting an exhaustive investigation of all emerging patterns is **beyond the scope of this initial benchmark release**.

As a benchmark paper, our goal is to **establish the task**, **validate feasibility**, and provide **statistically grounded observations** that demonstrate the utility of individualized reasoning evaluation. The primary contribution of this work is to introduce the **shift from simulating the average to simulating the individual**, and to provide the **first dataset**, **open-source data-collection pipeline**, and **formal evaluation framework** that enable such individualized assessment. HugAgent is intended as a **foundational first step**: we design the task, build the pipeline, construct the benchmark, and report a focused set of empirical findings aligned with this scope foundation empirical observations.

Within this scope, we already provide several analyses—including **multi-run evaluation**, **cross-domain transfer**, and **context-length sensitivity**—that substantively support our main claims and reveal meaningful insights.
Given the focus of this paper is to establish and validate the benchmark space and scope constraints, and because of these constraints, we present **foundational interpretations** of these findings rather than exhaustive ablations. The reviewer’s suggestions highlight promising avenues for deeper follow-up analyses, and we intend for future work to build upon this foundation.

Finally, we clarify that the **“guiding principles” in Section 8** are **conceptual takeaways** derived from the empirical patterns observed in our experiments. They are intentionally presented as **forward-looking hypotheses** for future exploration rather than claims that we exhaustively test within this paper.

---

### Author Response · Authors · 2025-12-04
**Summary to AC**

**[Summary of Scope & Contribution]**

HugAgent introduces a shift from **behaviorist evaluation** (mimicking population averages) to **cognitively grounded evaluation** (simulating the reasoning of a specific individual based on personal context). We are the first to **open-source a complete pipeline** for individualized reasoning evaluation.

--

**[Highlights: Strengths & Novelty]**
We appreciate the reviewers’ recognition of HugAgent’s core contributions:

• **Novelty of problem setting**: shifting evaluation from population consensus to **individual-level simulation** (R-imYS, R-pX6y).
• **Ecological validity**: using “out-loud” human reasoning traces instead of static QA, yielding a dense and naturalistic testbed (R-VpnV).
• **Significance**: addressing a critical gap in LLM-based digital twins and social simulation (R-pX6y).

--

**[Executive Summary of Major Updates]**
In response to the feedback, we have substantially revised the paper while preserving the core contributions and conclusions:

• **Expanded datasets** (R-VpnV, R-z6pz): The Human Track was increased to **N=54** (the maximum feasible within our >$8,350 budget), and the Synthetic Track was scaled to **N=500**.
• **New baselines** (R-VpnV, R-pX6y): We added results for GPT-5.1 Reasoning High, Claude Sonnet 4.5, and Gemini 2.5 Pro.
• **Robustness** (R-imYS): All models were evaluated over **five repeated runs**, showing strong stability.
• **Improved presentation** (R-z6pz): Section 3 now narrates the full pipeline, and the reasoning flow and structure have been clarified for readability. We also reorganized Sections 4–10 into a coherent, linear logic flow, moving from task construction to findings, diagnostics, error mechanisms, and mitigation strategies.

Below, we address the common concerns in detail with supporting evidence.

---

### Meta-Review · Area_Chair_bpC6 · 2026-01-02

**Summary:**

Reviewer imYS raised several methodological and framing concerns that limit confidence in the overall contribution. First, the use of the term “open-ended” was considered potentially misleading for the ML audience, as it diverges from its common interpretation in the literature and requires clearer positioning. Second, the reliance on scalar numerical predictions (e.g., 1 to 10 or 1 to 5 scales) prompted concerns regarding calibration and noise in LLM outputs, and the initial presentation did not sufficiently address robustness or distributional alignment with human judgments. Third, the synthetic agent track, based on deterministic causal graphs, was viewed as lacking direct human validation, raising questions about the ecological plausibility of the simulated reasoning trajectories.

Reviewer pX6y raised concerns about Scope and Completeness of Related Work, Limited Human Sample Size and Generalizability, Disconnect Between Theoretical Framing and Empirical Use, and Insufficient Validation of Certain Explanations and Claims, and Unvalidated Forward-Looking Claims.

Reviewer VpnV views the paper as marginally above the acceptance threshold (score: 6) and finds the core idea timely and well executed. The reviewer’s concerns are constructive rather than blocking, focusing on clarity, depth, and representativeness rather than questioning the validity or novelty of the contribution. The main concerns can be summarized as (1) human participant diversity and statistical power, clarity and consistency in experimental reporting, depth of analysis for ablation findings, and justification of the update operator in the formalism.

Reviewer z6pz’s concerns reflect a view that, while the problem setting is interesting and timely, the paper in its current form lacks the clarity, robustness, and evidentiary strength needed to support its claims. The combination of unclear methodology, noisy and weakly interpreted results, unresolved questions about benchmark validity, and limited engagement with prior work led the reviewer to recommend rejection.

**Reviewer Concerns:**

For Reviewer imYS: Concerns, such as calibration and interpretability of scalar scores and human plausibility of synthetic agents, are partially addressed or outstanding, while concerns regarding terminological ambiguity of “Open-ended” and stability of scalar predictions are adequately addressed.

For Reviewer pX6y, the rebuttal successfully addresses most factual, clarity, and overclaiming issues, and significantly improves the paper’s rigor and positioning. However, limitations related to human sample scale and the empirical strength of some higher-level claims remain intrinsic rather than fixable within rebuttal scope.

For Reviewer VpnV , the following concerns are fully addressed: Sample size / diversity justification, experimental reporting inconsistency, and formalism clarity. However, concern about depth of interpretation for ablation results is partially outstanding. Overall, the remaining issue concerns analytical richness rather than methodological validity, and does not undermine the paper’s core contribution or correctness.

For Reviewer z6pz:  The rebuttal substantially addresses the major clarity, presentation, and evaluation-protocol concerns, correcting concrete errors and strengthening empirical rigor. However, higher-level concerns about the depth, interpretability, and ecological validity of the findings remain only partially resolved. As a result, while the rebuttal significantly improves confidence in the soundness of the benchmark, it does not fully eliminate doubts about the strength and maturity of the conclusions.

**Reviewer Scores:**

Reviewer imYS would likely view the rebuttal as strengthening the paper, but not sufficiently to warrant a clear accept recommendation.

Reviewer pX6y’s original score was not driven by a negative view of the contribution, but by confidence gaps: missing context, perceived overclaims, and unclear empirical grounding. The rebuttal addresses most of these decisively. Therefore, the score would likely increase to a marginal accept.

Reviewer VpnV already rated the paper as marginally above the acceptance threshold and expressed strong support for the motivation, benchmark design, ecological validity, breadth of models evaluated, and open-source contribution. None of the weaknesses raised challenge the paper’s correctness, novelty, or relevance. Thus, a jump to score 7, reflecting clear acceptance, is the most plausible outcome.

The rebuttal significantly improves clarity and correctness, and alleviates many presentation concerns. However, the main findings still feel preliminary, and questions about ecological validity and interpretability remain. Reviewer z6pz might remain unconvinced that the paper is ready for acceptance.

---

### Decision · Program_Chairs · 2026-01-26

Reject